

# States, symmetries and correlators
# of $T\bar{T}$ and $J\bar{T}$ symmetric orbifolds

**Soumangsu Chakraborty[1,2], Silvia Georgescu[1,3] and Monica Guica[1,4,5]**

**1** Université Paris-Saclay, CNRS, CEA, Institut de Physique Théorique,
91191 Gif-sur-Yvette, France
**2** Institute for Theoretical Physics, University of Amsterdam,
PO Box 94485, 1090GL, Amsterdam, The Netherlands
**3** CPHT, CNRS, École polytechnique, Institut Polytechnique de Paris, 91120 Palaiseau, France
**4** Institute of Physics, Ecole Polytechnique Fedérale de Lausanne,
CH-1015 Lausanne, Switzerland
**5** Theoretical Physics Department, CERN, CH-1211 Geneva 23, Switzerland

## Abstract

We derive various properties of symmetric product orbifolds of $T\bar{T}$ and $J\bar{T}$ - deformed CFTs from a field-theoretical perspective. First, we generalise the known formula for the torus partition function of a symmetric orbifold theory in terms of the one of the seed to non-conformal two-dimensional QFTs; specialising this to seed $T\bar{T}$ and $J\bar{T}$ - deformed CFTs reproduces previous results in the literature. Second, we show that the single-trace $T\bar{T}$ and $J\bar{T}$ deformations preserve the Virasoro and Kac-Moody symmetries of the undeformed symmetric product orbifold CFT, including their fractional counterparts, as well as the KdV charges. Finally, we discuss correlation functions in these theories. By extending a previously-proposed basis of operators for $J\bar{T}$ - deformed CFTs to the single-trace case, we explicitly compute the correlation functions of both untwisted and twisted-sector operators and compare them to an appropriate set of holographic correlators. Our derivations are based mainly on Hilbert space techniques and completely avoid the use of conformal invariance, which is not present in these models.



# 1  Introduction

The study of symmetric product orbifolds of $T\bar{T}$ and $J\bar{T}$ - deformed CFTs is interesting for a number of reasons. First, symmetric product orbifolds of two-dimensional QFTs play an important role in holography, as their large $N$ behaviour is compatible with that of a gravitational dual where quantum-gravitational corrections are supressed [1]. When the seed theory is a CFT, they enter concrete realisations of the $\text{AdS}_3/\text{CFT}_2$ correspondence [2–7]. According to the proposals of [8–10], symmetric product orbifolds of $T\bar{T}$ [11,12] and $J\bar{T}$ - deformed CFTs [13] - a set of non-local, yet UV-complete and solvable two-dimensional QFTs - should provide tractable models of three-dimensional non-AdS holography. More precisely, the $T\bar{T}$ symmetric orbifold should be related to a spacetime that is asymptotically flat with a linear dilaton, whereas the $J\bar{T}$ one should correspond to a warped $\text{AdS}_3$ background, which is relevant to understanding the Kerr/CFT correspondence [14,15].

The study of symmetric product orbifolds of $T\bar{T}$ and $J\bar{T}$ - deformed CFTs is also interesting from the point of view of the original motivation of [11,12] - namely, to understand the space of integrable two-dimensional QFTs. The existence of exactly solvable irrelevant deformations of two-dimensional QFTs whose UV behaviour is not governed by a standard UV CFT fixed point, yet is entirely under control [16], is quite remarkable. The orbifold construction provides a simple way to enlarge the set of tractable examples of such QFTs. The properties of the resulting theories are similar - though not exactly the same - as those of the seed QFTs. It is a useful exercise to work them out explicitly from first principles, which is the main goal of this article.

Another motivation for studying this problem is that neither $T\bar{T}$, nor $J\bar{T}$ - deformed CFTs possess (full) conformal invariance, which is nevertheless omnipresent in the symmetric product orbifold literature. We would therefore like to use these examples to illustrate the fact that many observables in symmetric orbifold QFTs *can* be obtained without the conformality assumption. Depending on the specifics of the system under study, these observables can even include twisted-sector correlation functions, as we show explicitly for the case of single-trace $J\bar{T}$ - deformed CFTs.

The analysis presented in this article is purely field-theoretical, and the QFTs under study are *exact* symmetric product orbifolds of $T\bar{T}$ or $J\bar{T}$ - deformed CFTs, obtained via a single-trace $T\bar{T}/J\bar{T}$ deformation of an *exact* symmetric orbifold of two-dimensional CFTs. As a result, the large $N$ holographic duals of these theories are highly stringy. Our setup is thus different from that used in the holographic proposals [8–10], who deformed an *approximate* symmetric product orbifold of CFTs - namely, the CFT dual to the near horizon of several NS5 branes and a large number of F1 strings[1] - by an operator whose action resembles that of the single-trace $T\bar{T}$ or $J\bar{T}$ operator.[2] These deformations were argued to correspond to exactly marginal

---

[1]See [17] for a proposed concrete realisation of this CFT.

[2]Throughout this article, the single-trace $T\bar{T}/J\bar{T}$ operator will simply denote the sum over copies, in a sym-

deformations of the worldsheet string theory, which can be studied with a variety of techniques [18–29]. Most of the results obtained so far in the "single-trace $T\bar{T}$" and $J\bar{T}$ literature were in fact derived using worldsheet methods that, given the only approximate identification of the boundary deformation with single-trace $T\bar{T}/J\bar{T}$, may or may not agree with the exact symmetric product orbifold calculations. Thus, yet another motivation for this work is to provide an *independent* derivation of various properties of these theories that were previously predicted via holography.

The first observable we study is the finite-size spectrum of the orbifolded theories. This has been first computed using worldsheet methods, by studying the effect of the exactly marginal deformations on the spectrum of long strings in the massless BTZ background [9,10,18]. More precisely, it was shown that the spectrum of singly-wound long strings in the deformed backgrounds precisely coincides with the $T\bar{T}$ and, respectively, $J\bar{T}$ - deformed spectrum, which provided a non-trivial check of the proposed duality; the string theory prediction for the spectrum of multiply-wound strings was then naturally conjectured to represent the contribution of the twisted sectors of the symmetric product orbifold in this specific example. The $T\bar{T}$ result has been recently confirmed by the field-theoretical analysis of [30], who fixed the partition function by requiring it to be modular invariant in a generalised $T\bar{T}$ sense [31].

As already noted in [31], this modular invariance is an automatic property of the partition function of any (UV - complete) QFT with a single dimensionful scale, assuming its path integral on the torus is well-defined; the generalization to several parameters, including non Lorentz-invariant ones, is straightforward [32]. In this article, we provide a general expression for the partition function of the symmetric orbifold of such theories, based on a slight generalisation of Bantay's formula [33–35] for the case of CFTs; its modular invariance follows automatically from that of the seed QFT. When applied to the case of $T\bar{T}$ and $J\bar{T}$ - deformed CFTs, this partition function precisely reproduces or generalises previous results in the literature.

Given the partition function, one may analyse the thermodynamic properties of the symmetric product orbifold of $T\bar{T}/J\bar{T}$ - deformed CFTs. The $T\bar{T}$ case has been analysed in detail in [30]. We use these results to compare the entropy of a single-trace to that of a double-trace $T\bar{T}$ deformation [36] of a symmetric orbifold CFT and note that while they agree - as they should - in the universal high-energy regime discussed in [30], they disagree outside it. We also discuss the entropy of single/double-trace $J\bar{T}$ - deformed CFTs, showing there exists a regime of real high energies where the behaviour of the entropy is either Cardy-like or Hagedorn, depending on the chirality properties of the $U(1)$ current.

Next, we study the extended symmetries of single-trace $T\bar{T}$ and $J\bar{T}$ - deformed CFTs. A non-trivial property of the standard $T\bar{T}$ and $J\bar{T}$ deformations is that they preserve the Virasoro and, if present, the Kac-Moody symmetries of the undeformed CFT [37–39]. That the same is true of the single-trace deformation is strongly suggested by the results of the asymptotic symmetry group analysis of the linear dilaton spacetime [40], which uncovered an infinite set of symmetries, whose algebra closely resembles the $T\bar{T}$ symmetry algebra.

In this article, we provide a purely field-theoretical proof that these symmetries are indeed preserved, closely following the argument used in the double-trace case [38, 39]. This argument requires understanding the operator that drives the flow of the energy eigenstates under the single-trace $T\bar{T}/J\bar{T}$ deformation, which is technically more complicated than the corresponding double-trace flow in that many of the initial CFT degeneracies are broken when the deformation is first turned on. We also discuss other bases of symmetry generators, which are non-linearly related to the Virasoro one, and argue that they may be preferred at a global

---

metric orbifold QFT, of the corresponding Smirnov-Zamolodchikov operator. By contrast, in [8–10] "single-trace $T\bar{T}/J\bar{T}$" is a nickname given to a certain operator of dimension $(2,2)/(1,2)$ that is single-trace (in the sense of corresponding to a single-particle bulk excitation) and some of whose properties resemble those of $T\bar{T}/J\bar{T}$.

level in the single-trace $T\bar{T}$ and $J\bar{T}$ - deformed CFT. Working out the corresponding non-linear symmetry algebra in single-trace $T\bar{T}$ - deformed CFTs, we show the result agrees precisely with the holographic calculation [40]. In addition, we show that the KdV charges and the fractional Virasoro and Kac-Moody modes are preserved by the deformation; the fate of the higher spin symmetries such as those discussed in [41] is less clear.

Finally, we turn our attention to correlation functions. For standard $J\bar{T}$ - deformed CFTs, these have been understood in [42] (see also [43]), and recently have also been computed in $T\bar{T}$ - deformed CFTs [44] (see also [45]), using rather different methods. In addition, several holographic calculations of two-point functions - using either worldsheet or supergravity techniques - were performed in [20–25]. We provide explicit expressions for the correlation functions of a proposed set of both untwisted and twisted-sector operators in single-trace $J\bar{T}$ - deformed CFTs, which we then compare with a holographic computation of the two-point functions of long string vertex operators - the only worldsheet operators that are described by a symmetric product orbifold - performed using the methods of [20, 25]. The two results are found to slightly differ, and we comment on possible reasons for this.

This article is organised as follows. In section 2, we study the torus partition function of symmetric product orbifolds of general two-dimensional QFTs and show that it can be obtained via a slight generalisation of Bantay's formula; we work out the $T\bar{T}$ and $J\bar{T}$ case as an example. We also comment on the thermodynamics of single-trace $T\bar{T}$ and $J\bar{T}$ - deformed CFTs. In section 3, we study the flow of the states and of the Virasoro (- Kac-Moody) generators, including their fractional counterparts, in single-trace $T\bar{T}$ and $J\bar{T}$ deformed CFTs and show that they are still conserved, as are the KdV charges. We also discuss other possible bases of symmetry generators. Finally, in section 4 we compute correlation functions in single-trace $J\bar{T}$ - deformed CFTs and compare them with an appropriate holographic result. We end with a summary in section 5. For completeness, each section contains an introductory subsection that summarizes the relevant results from the double-trace case.

# 2 The spectrum and the entropy

In this section, we explain in a simple fashion how to obtain the finite-size spectrum of a symmetric product orbifold of any two-dimensional QFT whose partition function is modular invariant in an appropriately generalised sense. Our results are exemplified by symmetric orbifolds of $T\bar{T}$ - deformed CFTs in the Lorentz-invariant case, and symmetric orbifolds of $J\bar{T}$ - deformed CFTs in the non-Lorentz-invariant one. For completeness, we start this section with a brief review of the spectrum and partition function of standard (double-trace) $T\bar{T}$ and $J\bar{T}$ - deformed QFTs.

## 2.1 Review of the $T\bar{T}$ and $J\bar{T}$-deformed spectrum and partition function

One remarkable feature of $T\bar{T}, J\bar{T}$ deformations and their generalisations is that the spectrum of the deformed QFT on a cylinder of circumference $R$ is entirely determined by the finite-size spectrum of the undeformed QFT, as we now review.

### $T\bar{T}$ - deformed QFTs

The $T\bar{T}$ deformation is a universal irrelevant deformation of a two-dimensional QFT by an operator constructed from the components of the stress tensor

$$\partial_\mu S = \frac{1}{2} \int d^2x\, \mathcal{O}_{T\bar{T}}^{[\mu]}, \qquad \mathcal{O}_{T\bar{T}} = \epsilon^{\alpha\beta}\epsilon^{\gamma\delta}T_{\alpha\gamma}T_{\beta\delta}, \tag{1}$$

which enjoys nice factorization properties in energy eigenstates [11, 46]. These properties imply that the energies $E_n^{[\mu]}(R)$ of the eigenstates of the deformed theory on a cylinder of circumference $R$ obey Burger's equation

$$\partial_\mu E_n^{[\mu]}(R) = E_n^{[\mu]}(R)\frac{\partial E_n^{[\mu]}(R)}{\partial R} + \frac{P_n^2(R)}{R}\,. \tag{2}$$

This equation can be solved via the method of characteristics. For $P = 0$, the solution is simply given by $E_n^{[\mu]}(R) = E_n^{[0]}(R + \mu E_n^{[\mu]})$, where $E_n^{[0]}$ are the undeformed energies; the solution for $P \neq 0$ is a slight generalisation of this result [12]. Thus, if the spectrum of the undeformed QFT is known explicitly as a function of $R$, then so is the spectrum of the corresponding $T\bar{T}$ - deformed QFT. A well-studied example where this is the case is that of $T\bar{T}$ - deformed CFTs, where the undeformed energies are inversely proportional to $R$, and the solution for the deformed spectrum is

$$E_n^{[\mu]}(R) = \frac{R}{2\mu}\left(-1 + \sqrt{1 + \frac{4\mu E_n^{[0]}(R)}{R} + \frac{4\mu^2 P_n^2}{R^2}}\right). \tag{3}$$

This solution can also be written in terms of the conformal dimension $\Delta$ and spin $s$ of the corresponding operator by plugging in the expressions for $E_n^{[0]}, P_n$ as a function of $\Delta$ and $s$.

The torus partition function of the deformed QFT is defined as usual via the Hilbert space trace

$$Z^{[\mu]}(\tau, \bar{\tau}, R) = \sum_n e^{-\tau_2 R E_n^{[\mu]}(R) + i\,\tau_1 R P_n}\,, \tag{4}$$

where $\tau = \tau_1 + i\tau_2$ is the complex structure modular parameter and $R$ is the length of the $a$-cycle of the torus, here designated as the spatial one. For a $T\bar{T}$ - deformed CFT, $Z^{[\mu]}$ only depends on $R$ via the dimensionless combination $\mu/R^2$, since $\mu$ is the only dimensionful parameter in the theory.

Let us now discuss the modular transformation properties of this partition function. The flat metric on the torus can be written as

$$ds^2 = R^2|dx + \tau dy|^2 = R^2 dz d\bar{z}\,, \tag{5}$$

where $x, y$ are real coordinates of unit periodicity and the complex coordinates $z, \bar{z}$ are defined as $z = x + \tau y, \bar{z} = x + \bar{\tau} y$. This metric is invariant under large $PSL(2, \mathbb{Z})$ diffeomorphisms of the torus

$$\begin{pmatrix} x \\ y \end{pmatrix} \mapsto \begin{pmatrix} a & -b \\ -c & d \end{pmatrix}\begin{pmatrix} x \\ y \end{pmatrix}, \qquad \tau \mapsto \frac{a\tau + b}{c\tau + d}, \qquad ad - bc = 1\,, \tag{6}$$

which leave the coordinate periodicities intact, provided we also transform

$$R \rightarrow |c\tau + d|R\,. \tag{7}$$

Note this ensures that the area of the torus, $R^2\tau_2$, is invariant. Under (6), the complex coordinates change as

$$z \mapsto \frac{z}{c\tau + d}, \qquad \bar{z} \mapsto \frac{\bar{z}}{c\bar{\tau} + d}\,. \tag{8}$$

Assuming the partition function (4) can also be computed via an Euclidean path integral over the torus, which is naturally invariant under the diffeomorphisms discussed above, we conclude that

$$Z^{[\mu]}\left(\frac{a\tau + b}{c\tau + d}, \frac{a\bar{\tau} + b}{c\bar{\tau} + d}, R|c\tau + d|\right) = Z^{[\mu]}(\tau, \bar{\tau}, R)\,. \tag{9}$$

While we wrote this relation with $T\bar{T}$ - deformed CFTs in mind, whose partition function depends on a single dimensionful parameter $\mu$, it should hold in any UV-complete two-dimensional QFT with dimensionful scalar couplings - collectively denoted as '$[\mu]$' - whose partition function can be computed via a path integral over the euclidean torus. In a CFT, the radius dependence drops out by scale invariance, resulting in the usual modular invariance requirement; (9) may then be referred to as "generalised modular invariance". It simply states the invariance of the partition function under a relabeling of the torus coordinates, and as such it is natural that these transformations relate theories defined on tori with different sizes of the $a$-cycle, where the scalar couplings (which may be dimensionful) are held fixed.

In a $T\bar{T}$ - deformed CFT, the partition function $Z^{[\mu]}(\tau,\bar{\tau},R) = Z_{T\bar{T}}(\tau,\bar{\tau},\mu/R^2)$, and so the above relation reads

$$Z_{T\bar{T}}\left(\frac{a\tau+b}{c\tau+d}, \frac{a\bar{\tau}+b}{c\bar{\tau}+d}, \frac{\mu}{R^2|c\tau+d|^2}\right) = Z_{T\bar{T}}\left(\tau,\bar{\tau},\frac{\mu}{R^2}\right). \tag{10}$$

Thus, in this case one may reinterpret (9) as relating theories on a circle of the same radius, but with different dimensionless couplings. The above relation was checked explicitly in [47].

The density of states of a $T\bar{T}$ - deformed CFT follows from the adiabaticity of the deformation, which implies that the number of states is unchanged along the flow. We thus have

$$S_{T\bar{T}}(E) = S_{\text{Cardy}}(E^{[0]}(E)) = 2\pi\sqrt{\frac{cE_L(R+2\mu E_R)}{12\pi}} + 2\pi\sqrt{\frac{cE_R(R+2\mu E_L)}{12\pi}}, \tag{11}$$

where the relation between $E^{[0]}$ and $E, P$ was obtained by inverting (3), $E_{L,R} \equiv (E \pm P)/2$ and, for simplicity, we have dropped the $\mu$ label for the deformed energies. Note that the high-energy behaviour is Hagedorn.

Finally, let us remind the reader that reality of the deformed ground state energy (3) implies that $T\bar{T}$ - deformed CFTs can only be defined on cylinders whose circumference satisfies

$$R \geq R_{min} = \sqrt{\frac{2\pi\mu c}{3}}. \tag{12}$$

The high-energy behaviour of the entropy implies in turn that the thermal partition function only makes sense below the Hagedorn temperature $T_H = R_{min}^{-1}$, which amounts to the same constraint. More generally, we have

$$\lim_{E_{L,R}\to\infty} Z_{T\bar{T}} \approx e^{-\beta_L E_L - \beta_R E_R} e^{2\sqrt{\frac{2\pi\mu c}{3}E_L E_R}} \leq e^{2\sqrt{E_L E_R}(R_{min}-\sqrt{\beta_L \beta_R})}, \tag{13}$$

where $\beta_{L,R}$ are the left/right-moving temperatures, which satisfy $\beta_L \beta_R = R^2|\tau|^2$. The partition function is thus well defined provided also $R|\tau| > R_{min}$. One may check - by appropriately choosing the integer part of $\tau_1$ - that modular transformations do not take us out of this regime.

## $J\bar{T}$ - deformed QFTs

The $J\bar{T}$ deformation, as well as all Smirnov-Zamolodchikov deformations involving $U(1)$ currents and the stress tensor, can be treated in an entirely analogous manner. The only new element is that now the coupling has non-trivial transformation properties under diffeomorphisms, which need to be taken into account when discussing the modular invariance properties of the partition function.

We start our discussion with a general $JT^a$ deformation of a two-dimensional QFT, defined via the flow equation

$$\partial_{\lambda^a} S = \int d^2x\, T^{\alpha}{}_a \epsilon_{\alpha\beta} J^{\beta}. \tag{14}$$

The coupling parameters $\lambda^a$ are vectors with dimensions of length; these deformations thus break Lorentz invariance. The spectrum of the deformed QFT coupled to certain background fields is again simply related to the undeformed spectrum in a shifted background [48, 49]

$$E_n^{[\lambda^a]}(R, \nu, a^\sigma) = E_n^{[0]}\left(R - \lambda^\sigma q_n^{[0]}, \frac{\nu R - \lambda^t q_n^{[0]}}{R - \lambda^\sigma q_n^{[0]}}, a^\sigma + \frac{\lambda^\sigma(P_n + \nu E_n^{[\lambda^a]}) - \lambda^t E_n^{[\lambda^a]}}{R - \lambda^\sigma q_n^{[0]}}\right), \quad (15)$$

where $\nu$ is a background vielbein, $q_n^{[0]}$ is the undeformed $U(1)$ charge of the state, and $a^\sigma$ is a background gauge field, which may be set to zero at the end of the computation. The $J\bar{T}$ deformation corresponds to the case when $\lambda^a$ is a null vector with $\lambda^t = \lambda^\sigma = \lambda$ (or, equivalently, $\lambda^{\bar{z}} = 2\lambda, \lambda^z = 0$). If the seed theory is a CFT, then the $J\bar{T}$ deformation has the special property of preserving locality and conformal invariance on the left-moving side, which leads to great simplifications in the study of the deformed theory. The deformed spectrum is obtained by applying (15) to a seed CFT, and is best expressed in terms of the deformed right-moving energy

$$E_R^{[\lambda]}(R) = \frac{4\pi}{\lambda^2 k}\left(R - \lambda q^{[0]} - \sqrt{\left(R - \lambda q^{[0]}\right)^2 - \frac{\lambda^2 k E_R^{[0]} R}{2\pi}}\right), \quad (16)$$

where $E_R^{[0]}, q^{[0]}$ are the right-moving energies and $U(1)$ charges of the corresponding state in the undeformed CFT, and we have dropped the label '$n$' on the eigenstates. The left-moving $U(1)$ charge also changes non-trivially with $\lambda$, and is given by

$$q^{[\lambda]} = q^{[0]} + \frac{\lambda k}{4\pi} E_R^{[\lambda]} = \frac{1}{\lambda}\left(R - \sqrt{\left(R - \lambda q^{[0]}\right)^2 - \frac{\lambda^2 k R E_R^{[0]}}{2\pi}}\right). \quad (17)$$

Note that the deformed spectrum will become imaginary if the states in the undeformed CFT have large right-moving energy at fixed $q^{[0]}$, a behaviour that resembles that of $T\bar{T}$-deformed CFTs with $\mu < 0$. At the same time, reality of the deformed energy results into an upper bound[3] on $q^{[\lambda]}$, suggesting it is the latter that should be held fixed as $E_R^{[0]}$ is taken to be large. The relationship between the deformed and undeformed right-moving energies at fixed $q^{[\lambda]}$ is given by

$$E_R^{[\lambda]} = \frac{4\pi}{\lambda^2 k}\left(\sqrt{\left(R - \lambda q^{[\lambda]}\right)^2 + \frac{\lambda^2 k E_R^{[0]} R}{2\pi}} - \left(R - \lambda q^{[\lambda]}\right)\right). \quad (18)$$

From (17), $q^{[\lambda]}$ will be real provided the undeformed dimensionless energies $R E_R^{[0]}/2\pi$ lie below the parabola $(R/\lambda - q^{[0]})^2/k$ depicted in figure 1, which still allows access to infinite energies. In addition, there are lower bounds on the allowed energy. For example, if the seed CFT also posesses a right-moving $U(1)$ symmetry, the cosmic censorship bound on the right-moving side $R E_R^{[0]}/2\pi \geq (\bar{q}^{[0]})^2/k$ indicates it should lie above the parabola $(q^{[0]} - \mathrm{w})^2/k$, where $\mathrm{w} \equiv q^{[0]} - \bar{q}^{[0]}$ is the winding of the state, which is to be held fixed as we vary $q^{[0]}$. As long as $R > \lambda \mathrm{w}$ (for $\lambda > 0$), there is always a sliver in the $E_R^{[0]}, q^{[0]}$ plane so that both conditions are satisfied.

The allowed values of $E_R^{[0]}$ are further restricted by the cosmic censorship bound on the left-moving energy and charge, which requires that $R E_R^{[0]}/2\pi \geq (q^{[0]})^2/k - PR/2\pi$. It is easy

---

[3]One may increase $q^{[\lambda]}$ beyond the limiting value $R/\lambda$ by choosing a different branch of the square root. A similar behaviour was found for $T\bar{T}$ - deformed CFTs with $\mu < 0$ [50].

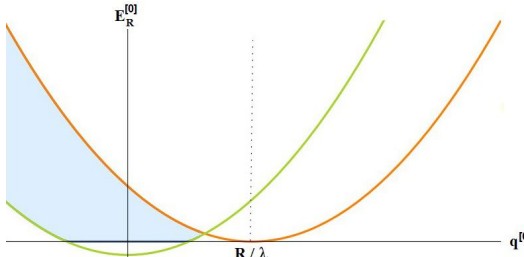

(a) The allowed region in the chiral case.

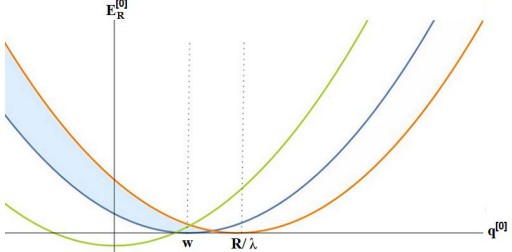

(b) The allowed region for non-chiral $U(1)$ current.

Figure 1: Range of undeformed right-moving energies (shaded region) that lead to real energies in $J\bar{T}$ - deformed CFTs and are allowed by the cosmic censorship bounds (green/blue parabolae). Note this range extends to infinite energies.

to check that for w $< R/\lambda$, there is always a region, depicted in figure 1b, that extends to infinite energies and obeys all three constraints. If the $U(1)$ current is chiral, then the second constraint is replaced by positivity of the energy, and we are in the situation of figure 1a. Within the allowed region, we will be interested in the regime where $E_R^{[0]}$ is large and $q^{[0]}$ is large and negative, which corresponds via (18) to a large deformed right-moving energy.

The full deformed spectrum may be understood as a spectral flow by the right-moving Hamiltonian, as discussed in [51]. This observation also extends to the spectrum of $SL(2,\mathbb{R})_L$ conformal dimensions on the plane, which in $J\bar{T}$ - deformed CFTs are well-defined thanks to the fact that the theory enjoys full left conformal invariance. This spectrum may be obtained by applying an infinite boost to (16), and reads, as a function of the right-moving energy, now denoted $\bar{p}$ [43]

$$h^{[\lambda]}(\bar{p}) = h^{[0]} + \frac{\lambda}{2\pi}q^{[0]}\bar{p} + \frac{\lambda^2 k}{16\pi^2}\bar{p}^2 , \qquad q^{[\lambda]}(\bar{p}) = q^{[0]} + \frac{\lambda k}{4\pi}\bar{p} , \qquad (19)$$

where $h^{[0]}, q^{[0]}$ are the left-moving conformal dimension and charge in the undeformed CFT. These dimensions can also be obtained via conformal perturbation theory [43]. Note this spectrum is manifestly real, indicating that the problems associated with the imaginary energy states disappear in infinite volume, in agreement with their physical interpretation put forth in [52].

The torus partition function of a $J\bar{T}$ - deformed CFT is given by

$$Z_{J\bar{T}}\left(\tau, \bar{\tau}, \nu, \frac{\lambda}{R}\right) = \sum_n e^{-\tau_2 R E_n^{[\lambda]}(R) + i\tau_1 R P_n + 2\pi i \nu q_n^{[\lambda]}(R)} , \qquad (20)$$

where $\nu$ is the chemical potential that couples to the chiral $U(1)$ current. We wrote the coupling $\lambda$ as an argument - rather than a label - because it changes under diffeomorphisms, due to its vectorial nature. Since imaginary energy modes are present for any value of the radius, it is currently not well understood to what extent this partition function is well defined; however, the fact that the theory admits a non-perturbative definition [49] yields hope that its study is meaningful.

The modular transformation properties of this partition function were discussed in [32]. Since $\lambda^a$ transforms as a vector, $\lambda^{\bar{z}}$, under modular transformations, which has the same transformation properties as $R\bar{z}$, it follows that the dimensionless combination $\lambda/R$ transforms exactly as $\bar{z}$

$$\frac{\lambda}{R} \mapsto \frac{\lambda}{R(c\bar{\tau}+d)} . \qquad (21)$$

Consequently, the dimensionful deformation parameter changes $\lambda \mapsto \frac{\lambda|c\tau+d|}{(c\bar{\tau}+d)}$. A similar argument can be used to derive the well-known transformation properties of the chemical potential[4] $\nu$

$$\nu \mapsto \frac{\nu}{c\tau+d}. \tag{22}$$

With this in mind, the partition function has the standard anomalous transformation under diffeomorphisms of the torus

$$Z_{J\bar{T}}\left(\frac{a\tau+b}{c\tau+d}, \frac{a\bar{\tau}+b}{c\bar{\tau}+d}, \frac{\nu}{c\tau+d}, \frac{\lambda}{R(c\bar{\tau}+d)}\right) = \exp\left(\frac{2i\pi kc\,\nu^2}{c\tau+d}\right) Z_{J\bar{T}}\left(\tau, \bar{\tau}, \nu, \frac{\lambda}{R}\right). \tag{23}$$

One may also consider a slightly redefined partition function, which is invariant under these transformations [53]

$$Z_{J\bar{T}}^{inv}\left(\tau, \bar{\tau}, \nu, \frac{\lambda}{R}\right) \equiv e^{\frac{\pi k\nu^2}{\tau_2}} Z_{J\bar{T}}\left(\tau, \bar{\tau}, \nu, \frac{\lambda}{R}\right). \tag{24}$$

This transformation law can be readily extended to QFTs that may have various couplings that transform non-trivially under Lorentz transformations.

Finally, let us discuss the density of states. The entropy is again estimated by using the fact that the number of states does not change in fixed units. One may distinguish two cases: if the current in the seed CFT is not chiral, then

$$S_{J\bar{T}}(E, q) = S_{\text{Cardy}}(E^{[0]}, q^{[0]}) = 2\pi\sqrt{\frac{c}{6}\left(\frac{RE_L^{[0]}}{2\pi} - \frac{q^{[0]2}}{k}\right)} + 2\pi\sqrt{\frac{c}{6}\left(\frac{RE_R^{[0]}}{2\pi} - \frac{\bar{q}^{[0]2}}{k}\right)}$$

$$= 2\pi\sqrt{\frac{c}{6}\left(\frac{RE_L}{2\pi} - \frac{q^2}{k}\right)} + 2\pi\sqrt{\frac{c}{6}\left(\frac{(R-\lambda\text{w})E_R}{2\pi} - \frac{\bar{q}^2}{k}\right)}, \tag{25}$$

where we assumed, as explained above, that $q \equiv q^{[\lambda]}$ and $\bar{q} \equiv \bar{q}^{[\lambda]}$ are to be fixed in the deformed theory, as well as their difference, w. Note that the difference from the standard Cardy formula in presence of $U(1)$ charge is rather minimal. If, on the other hand, the current $J$ is chiral, then effectively $\bar{q}^{[0]} = 0$, and we obtain instead

$$S_{J\bar{T}} = 2\pi\sqrt{\frac{c}{6}\left(\frac{RE_L}{2\pi} - \frac{q^2}{k}\right)} + 2\pi\sqrt{\frac{c}{12\pi}\left(E_R(R-\lambda q) + \frac{\lambda^2 kE_R^2}{8\pi}\right)}. \tag{26}$$

Taking $E_R$ large with $q$ fixed, one finds Hagedorn behaviour at large energies. The above formula can be alternatively rewritten in terms of $q^{[0]}$ using (17), but then the limit of large $E_R$ with $q^{[0]}$ fixed is problematic because the square roots become imaginary.

## 2.2 Torus partition function of general symmetric product orbifold QFTs

In this subsection, we review and slightly generalize the well-known group-theoretical derivation [33] of the torus partition function of a symmetric product orbifold of two-dimensional QFTs. While this discussion is usually particularised to symmetric orbifolds of *CFTs*, we point out that the derivation only mildly depends on the conformal property of the seed. We can thus apply this method to general two-dimensional QFTs, and in particular to $T\bar{T}$ and $J\bar{T}$ - deformed CFTs.

---

[4]The chemical potential is related to a background gauge field, $a^z$, that couples to the left current as $\nu = \beta a^z = \tau_2 Ra^z$. Invariance of the action under diffeomorphisms implies that $a^z$ transforms in the opposite way from $R\bar{z}$ which, using (8), leads to $Ra^z \mapsto (c\bar{\tau}+d)Ra^z$. Taking into account the transformation of $\tau_2$, we find the above result.

We thus consider a two-dimensional QFT on a cylinder of circumference $R$. This will be referred to as the *seed* QFT and will be denoted as $\mathcal{M}$. The theory obtained by taking a $N$-fold tensor product $\mathcal{M}^N$ admits a natural action of the permutation group, $S_N$. Quotienting it by the permutation group, one obtains the symmetric product orbifold theory, denoted as $\mathcal{M}^N/S_N$.

The Hilbert space of $\mathcal{M}^N/S_N$ is organized into twisted sectors [54], labeled by the conjugacy classes, denoted $[g]$, of $S_N$

$$\mathcal{H}(\mathcal{M}^N/S_N) = \oplus_{[g]} \mathcal{H}^{[g]}. \tag{27}$$

Each $S_N$ conjugacy class is entirely specified by the lengths ($n$) and multiplicities ($N_n$) of the cycles of the permutation, with $\sum_n n N_n = N$. Within each conjugacy class, one keeps the states invariant under the centralizer (a.k.a commutant) of $g$, which does not depend on the chosen representative. The resulting structure of the factors $\mathcal{H}^{[g]}$ is

$$\mathcal{H}^{[g]} = \otimes_{n>0} \left( \mathcal{H}^{\mathbb{Z}_n} \right)^{N_n}/S_{N_n}, \tag{28}$$

where $\mathcal{H}^{\mathbb{Z}_n}$ is the Hilbert space assciated with a $\mathbb{Z}_n$ cyclic orbifold of the seed QFT, and the symmetrization is performed with respect to all cycles of the same length $n$. The untwisted sector of this Hilbert space, which corresponds to the conjugacy class of the identity, is simply $(\mathcal{H}_{seed})^N/S_N$. The twisted sectors are characterized by the basic fields having twisted boundary conditions around the spatial cycle of the cylinder. States belonging to different twisted sectors are orthogonal, as is clear from the direct sum structure (27).

The twisted sectors can be understood by mapping to corresponding covering spaces. This is particularly simple to implement for the torus partition function, as the relevant covering spaces are again tori, allowing one to express the partition function of the orbifold QFT solely in terms of the seed partition function. The first results on the torus partition function (or, rather, elliptic genus) of a symmetric product orbifold were obtained in [54] using string-theoretical methods. In a series of articles [33–35] that built upon this work, an explicit group-theoretical construction of the torus partition function of a symmetric product orbifold of a generic CFT in terms of that of the seed was given. Importantly, this derivation - detailed below - does not involve conformal invariance, but only relies on the modular invariance of the seed partition function.

**Review and slight generalisation of Bantay's formula**

The basic idea is the following: the partition function of the symmetric product orbifold QFT receives contributions from the different twisted sectors of the theory. Rather than considering fields with twisted boundary conditions on the original torus - denoted $\mathcal{T}^2$ - one can equivalently work with fields with standard boundary conditions on a covering space of the torus. The latter are unramified coverings with $N$ sheets - not necessarily connected - for which the monodromy group - which encodes how the various sheets permute as one goes around a loop in base space - is a subgroup of the permutation group, $S_N$ [55]. Permuting the sheets of the covering space under the monodromy action of the fundamental group corresponds to permuting the copies in the symmetric product orbifold, thus implementing the action of $S_N$ in the QFT in a geometrical fashion.

Connected components of such covering spaces are associated to orbits of elements of the set $\{1,2,...,N\}$ under the action of $S_N$. We generically denote these orbits by[5] $\xi$. By the Riemann-Hurwitz theorem, each such connected component is a torus, denoted $\mathcal{T}^2_\xi$, on which the seed theory lives and which covers the base $\mathcal{T}^2$ $|\xi|$ times.

---

[5]For example, for $N = 5$ and the monodromy group generated by the permutations $(12)$ and $(345)$, namely $\langle (12),(345) \rangle = \{e,(12),(345),(354),(12)(345),(12)(354)\} \subset S_5$, the orbit of e.g. the element 3 is $\xi = \{3,4,5\}$.

Each covering torus $\mathcal{T}_\xi^2$ can be written as the quotient of the complex plane by its fundamental group $\pi_1(\mathcal{T}_\xi^2) \cong \mathbb{Z} \oplus \mathbb{Z}$, which is a subgroup of index[6] $|\xi|$ of the fundamental group of the base torus $\pi_1(\mathcal{T}^2) \cong \mathbb{Z} \oplus \mathbb{Z}$ or, equivalently, a sublattice of index $|\xi|$ of the lattice associated to the base torus. These subgroups are labeled by three integers $m_\xi, r_\xi, \ell_\xi$, with $\ell_\xi > r_\xi \geq 0$ and $m_\xi \ell_\xi = |\xi|$. Given a basis of generators for $\pi_1(\mathcal{T}^2)$, usually denoted as the $a$ and $b$ cycles, these integers determine the generators of $\pi_1(\mathcal{T}_\xi^2)$ - namely, the $a_\xi, b_\xi$ cycles - as

$$a_\xi = \ell_\xi a, \qquad\qquad b_\xi = r_\xi a + m_\xi b. \qquad\qquad (29)$$

Hence, the modular parameters of the covering tori can be written as

$$\tau_\xi = \frac{m_\xi \tau + r_\xi}{\ell_\xi}. \qquad\qquad (30)$$

In addition, if the length of the $a$-cycle on the base torus is $R$, it follows that the length of the $a_\xi$-cycle on the covering torus $\mathcal{T}_\xi^2$ is

$$R_\xi = \ell_\xi R. \qquad\qquad (31)$$

An explicit example of the covering tori can be found in the pedagogical exposition of [56]. The area of the covering torus is $R_\xi^2 (\tau_\xi)_2 = \ell_\xi m_\xi R^2 \tau_2 = |\xi| R^2 \tau_2$, in agreement with the fact that it covers the base torus $|\xi|$ times. This size information will be important when discussing the partition function of a non-conformal QFT, such as $T\bar{T}$-deformed CFTs, whose partition function depends explicitly on the length, $R$, of the $a$-cycle.

Note that in the above, we have made a specific choice of parametrization, i.e. choice of basis of the generators of the fundamental group of the base and the covering tori. However, this choice should be immaterial as long as the quantities we compute are modular invariant.

Let us now reformulate these geometric data in group-theoretical language. The covering spaces discussed above are in one-to-one relation with the homomorphisms $\phi : \pi_1(\mathcal{T}^2) \to S_N$. Using this correspondence, one can rewrite the covering space data in terms of permutations. Any such homomorphism is fully specified by two commuting permutations, $\phi(a), \phi(b) \in S_N$ that generate $\phi(\pi_1(\mathcal{T}^2)) \subset S_N$, corresponding to the choice of the two loops that generate $\pi_1(\mathcal{T}^2)$. From the perspective of the QFT on the base $\mathcal{T}^2$, $\phi(a)$ and $\phi(b)$ correspond to the monodromies acquired by the fields as they circle around the $a$ and $b$-cycle. The covering tori $\mathcal{T}_\xi^2$ correspond to the orbits $\xi$ under the action of $\phi(\pi_1(\mathcal{T}^2)) \subset S_N$. Each covering torus is determined by its fundamental group which, as explained in [33], is isomorphic to the stabilizer associated to the orbit

$$S_\xi = \{\phi(x) \in \phi(\pi_1(\mathcal{T}^2)) \mid \phi(x)\xi^* = \xi^*, \forall \xi^* \in \xi\} \cong \pi_1(\mathcal{T}_\xi^2). \qquad (32)$$

Intuitively, the elements of the stabilizer act by definition as identity on $\xi$, thus mapping each sheet of the associated covering space into itself. Under this trivial monodromy action, all loops in the base space are lifted to loops in the covering space, i.e. elements of the fundamental group of the covering. In particular, for the choice (29), the generators of $\pi_1(\mathcal{T}_\xi^2)$ are mapped by this isomorphism into

$$\phi(a)^{\ell_\xi} \quad \text{and} \quad \phi(a)^{r_\xi}\phi(b)^{m_\xi}, \qquad\qquad (33)$$

providing a group-theoretical interpretation of the integers $m_\xi, r_\xi, \ell_\xi$ that determine the complex structure of the covering tori: $m_\xi$ is the number of $\phi(a)$ orbits in $\xi$, $\ell_\xi$ is their common

---

[6]The index of a subgroup $H \subset G$ is the number, $|G/H|$, of cosets of $H$ in G.

length and $r_\xi$ is the smallest nonnegative integer such that $\phi(a)^{r_\xi}\phi(b)^{m_\xi}$ belongs to the stabilizer.[7] See e.g. [56] for more examples.

Putting everything together, one can express the partition function of the symmetric product orbifold on a torus with modular parameter $\tau$ and length of the $a$-cycle $R$ in terms of the seed partition function on tori of different modular parameters (30) and radii (31) as [33]

$$Z^{S_N}(\tau, \bar\tau, R) = \frac{1}{N!} \sum_{\phi:\pi_1(\mathcal{T}^2)\to S_N} \prod_{\xi \text{ orbit}} Z^{seed}(\tau_\xi, \bar\tau_\xi, R_\xi), \tag{34}$$

where we suppressed for now the possible dependence of the partition function on other parameters. This should be sufficient for constructing the partition function of a symmetric product orbifold of arbitrary Lorentz-invariant QFTs. The Lorentz-breaking case will be discussed when treating symmetric orbifolds of $J\bar T$ - deformed CFTs.

This formula can be further massaged by considering the generating function

$$\sum_{N=0}^{\infty} p^N Z^{S_N}(\tau, \bar\tau, R) = \exp \sum_{n=1}^{\infty} p^n \mathcal{Z}^{(n)}, \qquad \mathcal{Z}^{(n)} = \frac{1}{n} \sum_{\mathcal{T}^2_\xi\big|_{|\xi|=n}} Z^{seed}(\mathcal{T}^2_\xi), \tag{35}$$

where the sum in $\mathcal{Z}^{(n)}$ runs over connected covering space of $\mathcal{T}^2$ with $n$ sheets, which are the tori $\mathcal{T}^2_\xi$ discussed previously, with $|\xi| = n$. Collecting the coefficient of $p^N$ on the right-hand-side of the first sum in (35), the formula (34) for the partition function of the $\mathcal{M}^N/S_N$ orbifold theory can be written more compactly as

$$Z^{S_N}(\tau, \bar\tau, R) = \frac{1}{N!} \sum_{x\in S_N} \prod_{\xi \text{ orbit}} |\xi| \mathcal{Z}^{(|\xi|)}. \tag{36}$$

As noted in [35], in the above formula the sum runs over all permutations in $S_N$, which is much simpler to handle than the previous sum (34) over homomorphisms, namely over pairs of commuting permutations in $S_N$. Since twisted sectors correspond to permutations up to conjugation, (36) provides an easy way to read off the contributions of the different sectors.

The individual contributions $\mathcal{Z}^{(n)}$ are given explicitly by

$$\mathcal{Z}^{(n)} = \frac{1}{n} \sum_{\ell|n} \sum_{0\leq r<\ell} Z^{seed}\left(\frac{n\tau}{\ell^2}+\frac{r}{\ell}, \frac{n\bar\tau}{\ell^2}+\frac{r}{\ell}, R\ell\right). \tag{37}$$

The above formula gives the contribution of all sets of equal-length cycles whose lengths sum to $n$, where $\ell$ is the length of the cycles and $n/\ell$ gives the number of cycles of that length. Choosing $\ell = 1$, we obtain the contribution to this term of the states from the untwisted sector (in the form of $n$ identical copies of the same state in the seed QFT), while choosing $\ell = n$ we obtain the contribution of the twisted sector of a single cycle of length $n$.

---

[7]Let us give an example: in $S_5$, we consider again the covering space associated to the homomorphism $a \mapsto \phi(a) = (345), b \mapsto \phi(b) = (12)$. Clearly, it has two connected components. We first consider the one associated to the orbit $\xi = \{3,4,5\}$, that should give a covering space with $|\xi| = 3$ sheets. The corresponding stabilizer is $S_\xi = \{e, (12)\} \cong \mathbb{Z}_2$. The number of $\phi(a)$ orbits in $\xi$ is 1 and its length is 3, which means $m_\xi = 1, \ell_\xi = 3$ and $r_\xi = 0$, leading to a modular parameter $\tau_\xi = \tau/3$ for the covering torus. Note that $\phi(a)^{\ell_\xi} = (345)^3 = e, \phi(a)^{r_\xi}\phi(b)^{m_\xi} = (12)$ are the elements of $S_\xi$. Similarly, for the orbit $\xi = \{1,2\}$, the stabilizer is $\{e, (345), (354)\}$. The number of $\phi(a)$ orbits is $m_\xi=2$ and their common length is $\ell_\xi = 1$ because $\phi(a)$ acts as identity on 1 and 2; again $r_\xi = 0$, implying $\tau_\xi = 2\tau$.

**Comments on modular invariance**

As we already discussed, well-definiteness of the torus partition function of the seed QFT requires it to be modular invariant in the generalised sense we reviewed. We would now like to show that modular invariance of the symmetric orbifold partition function (36) automatically follows from that of the seed.

In the simplest case of a CFT seed, the partition function depends only on the modular parameter of the torus. It is then not hard to recognise the 'connected' $\mathcal{Z}^{(n)}$ contribution as the action of the $n^{th}$ Hecke operator, $T_n$, on the seed partition function

$$\mathcal{Z}^{(n)}_{CFT} = T_n Z^{seed}_{CFT}, \tag{38}$$

where, by definition, the action of a Hecke operator $T_n$ on a modular form of weight $(\kappa, \bar{\kappa})$ is

$$T_n f(\tau, \bar{\tau}) = \frac{1}{n} \sum_{\substack{r, \ell \in \mathbb{Z}, \ \ell | n \\ 0 \le r < \ell}} \frac{1}{\ell^{\kappa + \bar{\kappa}}} f\left(\frac{n\tau}{\ell^2} + \frac{r}{\ell}, \frac{n\bar{\tau}}{\ell^2} + \frac{r}{\ell}\right), \tag{39}$$

and produces another modular form of the same weight. The seed partition function is modular invariant, i.e. it is simply a modular form of weight zero. Then, (38) implies that $\mathcal{Z}^{(n)}_{CFT}$, and thus the full symmetric orbifold partition function, is also modular invariant.

More generally, if the theory only possesses scalar dimensionful couplings, the partition function would depend on the dimensionless combinations built from these couplings and $R$. If this partition function allows for a Taylor expansion in terms of this dimensionless coupling, as is the case for e.g. $T\bar{T}$-deformed CFTs

$$Z^{seed}_{T\bar{T}}\left(\tau, \bar{\tau}, \frac{\mu}{R^2}\right) = \sum_{\kappa=0}^{\infty} \left(\frac{\mu}{R^2}\right)^\kappa Z_\kappa(\tau, \bar{\tau}), \tag{40}$$

then, as already discussed in [31], the coefficients of this expansion are all modular forms of weight $(\kappa, \kappa)$. Acting with the $n^{th}$ Hecke operator on each term and then resumming yields precisely (37)

$$T_n Z^{seed}_{T\bar{T}}\left(\tau, \bar{\tau}, \frac{\mu}{R^2}\right) := \frac{1}{n} \sum_\kappa \left(\frac{\mu}{R^2}\right)^\kappa \sum_{\substack{r, \ell \in \mathbb{Z}, \ell | n \\ 0 \le r < \ell}} \frac{1}{\ell^{2\kappa}} Z_\kappa\left(\frac{n\tau}{\ell^2} + \frac{r}{\ell}, \frac{n\bar{\tau}}{\ell^2} + \frac{r}{\ell}\right)$$

$$= \frac{1}{n} \sum_{\substack{r, \ell \in \mathbb{Z}, \ell | n \\ 0 \le r < \ell}} Z^{seed}_{T\bar{T}}\left(\frac{n\tau}{\ell^2} + \frac{r}{\ell}, \frac{n\bar{\tau}}{\ell^2} + \frac{r}{\ell}, \frac{\mu}{R^2 \ell^2}\right). \tag{41}$$

Hence, we can again write $\mathcal{Z}^{(n)} = T_n Z^{seed}(\tau, \bar{\tau}, R)$, where the action of the Hecke operator on the full partition function is defined as the action on each coefficient in the series expansion in $\mu/R^2$. The modular invariance of the seed then implies the modular invariance of the symmetric product. The same reasoning applies when the couplings have non-trivial transformation properties; examples will be given in the following section, where we will be discussing in detail the case of $J\bar{T}$-deformed CFTs with a chemical potential.

$T\bar{T}$ - deformed CFTs are, in a certain sense, the next simplest case to consider beyond just CFT, since the fact that the coupling $\mu$ has a negative mass dimension allows for an expansion in terms of standard modular forms of positive weight. More generally, there is no reason to expect that the partition function would be analytic in the given coupling, and therefore the above argument using the Taylor expansion would not hold. It is, nevertheless, possible to argue for modular invariance of $\mathcal{Z}^{(n)}$ directly from the modular properties of the seed partition function: the $T$ transformation $\tau \to \tau + 1$ of the base QFT simply reshuffles the terms in the sum

(40), whereas the $S$ transformation $\tau \to -1/\tau$ can be undone by a modular transformation of the covering tori, together with a reshuffling of the terms in the sum, as argued in [57] for the CFT case. Including the radius dependence is straightforward.[8]

**General features of the spectrum of symmetric product orbifolds**

The spectrum of the symmetric product orbifold can be readily extracted from the partition function (36), by giving it a Hilbert space interpretation

$$Z^{S_N}(\tau, \bar{\tau}, R) = \sum_n d_n e^{-\beta E_n + i P_n \theta}, \quad \beta = R\tau_2, \ \theta = R\tau_1, \tag{42}$$

where we have now explicitly included the ventual degeneracies, $d_n$, of the energy levels. We would like to express the finite-size energies and momenta $E_n, P_n$ of the symmetric product orbifold, as well as $d_n$, in terms of those of the seed QFT, denoted $E_n^{(s)}, P_n^{(s)}, d_n^{(s)}$

$$Z^{seed}(\tau, \bar{\tau}, R) = \sum_n d_n^{(s)} e^{-\beta E_n^{(s)} + i P_n^{(s)} \theta}. \tag{43}$$

As a warm-up, it is useful to first work out the contribution to the partition function of the twisted sector associated to a single cycle of length $w$, which we will refer to as the $w$-twisted sector. It corresponds to the $\ell = w$ contribution to $\mathcal{Z}^{(w)}$ and will be denoted $Z^{(w)}$

$$Z^{(w)} \equiv \mathcal{Z}^{(w)}(\tau, \bar{\tau}, R)\big|_{\ell=w} = \frac{1}{w} \sum_{0 \le r < w} Z_{seed}\left(\frac{\tau + r}{w}, \frac{\bar{\tau} + r}{w}, Rw\right). \tag{44}$$

One immediately notes that in this sector, the contributions of the seed partition are evaluated at the same inverse temperature, $\beta = R\tau_2$, as that of the full orbifold, even though the length of the spatial circle is $w$ times larger. This implies that

$$E_n^{(w)}(R) = E_n^{(s)}(Rw), \tag{45}$$

where $E_n^{(w)}(R)$ represent the finite-size energy levels in the $w$ - twisted sector. Note this result follows without any use of conformal invariance, but only of the modular invariance properties of the partition function we have been assuming throughout this section. For the case of a CFT, the energies on the cylinder can be related to the conformal dimensions of the corresponding operators on the plane via the usual conformal map, which yields

$$E_{n, CFT}^{(w)}(R) = \frac{2\pi(\Delta_n^{(w)} - \frac{cw}{12})}{R}, \qquad E_{n, CFT}^{(s)}(wR) = \frac{2\pi(\Delta_n^{(s)} - \frac{c}{12})}{Rw}, \tag{46}$$

where $c$ is the central charge of the seed CFT. Note that the gap above the ground state ($\Delta^{(s)} = 0$) is, as is well-known, $w$ times smaller in the twisted sector than in the untwisted one. The $cw/12$ shift between the energy and the dimension in the twisted sector follows from the fact that the effective central charge of the latter is $cw$. Combining (45) and (46), we obtain

$$\Delta_n^{(w)} = \frac{\Delta_n^{(s)}}{w} + \frac{c}{12}\left(w - \frac{1}{w}\right), \tag{47}$$

---

[8]The $S$ transformation on the base torus maps $\tau_\xi \mapsto \tau'_\xi = (r_\xi \tau - m_\xi)/(\ell_\xi \tau)$ and $R_\xi \mapsto R'_\xi = |\tau|\ell_\xi R$. Using equation (12) of [57], one can show that $\tau'_\xi, R'_\xi$ are related by a modular transformation to $\tilde{\tau}_\xi = (m_\xi^* \tau - r_\xi^*)/\ell_\xi^*, \tilde{R}_\xi = \ell_\xi^* R$, where $\{\ell_\xi^*, m_\xi^*, r_\xi^*\}$ are integers with $\ell_\xi^* m_\xi^* = |\xi|$ and $0 \le r_\xi^* < \ell_\xi^*$, that parametrize the orbit $\xi$, just like $\{\ell_\xi, m_\xi, r_\xi\}$. The relation between the two parametrization is explained in [57]. Using the modular invariance of the seed partition function, it follows that the sum in (37) is invariant under the $S$ transformation of the base torus.

which reproduces the known result for the twisted sector operator dimensions [58]. Note that in the above, conformal invariance was only used to translate the cylinder energies into operator conformal dimensions, but is otherwise not needed to derive (45), which holds equally well in a non-conformal theory.

Let us now also take into account the momentum dependence of the $w$-twisted sector partition function (44). The quantity $\theta = \tau_1 R$ being the same as in the seed, we again have

$$P_n^{(w)}(R) = P_n^{(s)}(wR) = \frac{2\pi p_n}{wR}, \quad p_n \in \mathbb{Z}, \tag{48}$$

where $p_n$ is the integer-quantized momentum of the corresponding state in the seed QFT. The above appears to imply that the twisted-sector momentum may be a fractional multiple, $p_n/w$, of the inverse radius, which would be inconsistent with modular invariance. This is resolved by the sum present in (44), since for every energy-momentum eigenstate in the seed, the full contribution to $Z^{(w)}$ is

$$\frac{1}{w} \sum_{r=0}^{w-1} e^{-\beta E_n + i(\theta + rR)P_n} = \frac{1}{w} e^{-\beta E_n + i\theta P_n} \sum_{r=0}^{w-1} e^{2\pi i r \{\frac{p_n}{w}\}} = e^{-\beta E_n + i\theta P_n} \, \delta(p_n = 0 \bmod w), \tag{49}$$

where in the second step we noted that only the fractional part of $p_n/w$ contributes, and in the third we trivially summed the geometric series. Thus, the momentum in the twisted sector is an integer, as expected, and only seed momenta that are multiples of $w$ will end up contributing to the $w$-twisted sector partition sum.

To summarize, each state in the seed QFT gives rise to a state in the $w$ - twisted sector, whose energy and momentum are

$$E^{(w)}(R) = E^{(s)}(wR), \quad P^{(w)}(R) = P^{(s)}(wR) \quad \text{iff} \quad P^{(s)}(R) \in \frac{2\pi}{R} w \mathbb{Z}, \tag{50}$$

In particular, the degeneracies of these states are the same, provided the constraint on the momentum is satisfied.

The contributions of the terms with $\ell \neq w$ to $\mathcal{Z}^{(w)}$ - namely, of sectors with $w/\ell$ cycles of length $\ell$ - can be analysed in an analogous manner. We find

$$E_n^{(w)}(R) = \frac{w}{\ell} E_n^{(s)}(\ell R), \quad P_n^{(w)}(R) = \frac{w}{\ell} P_n^{(s)}(\ell R) \quad \text{iff} \quad P^{(s)}(R) \in \frac{2\pi}{R} \ell \mathbb{Z}. \tag{51}$$

These states correspond to $w/\ell$ identical copies of the same state from the $\ell$ - twisted sector, in agreement with the selection rule on the momentum.

The full spectrum of the symmetric orbifold is given by putting together these elements inside the partition function. It it useful to work out explicity the full partition function (36) for the the simplest example $N = 2$, as higher $N$ work qualitatively similarly. In this case, there are only two sectors, one untwisted and one 2-twisted. Applying Bantay's formula (36), we have

$$Z^{S_2}(\tau, \bar{\tau}, R) = \frac{1}{2}(\mathcal{Z}^{(1)})^2 + \mathcal{Z}^{(2)} \tag{52}$$

$$= \frac{1}{2} Z_{seed}^2(\tau, \bar{\tau}, R) + \frac{1}{2} Z_{seed}(2\tau, 2\bar{\tau}, R) + \frac{1}{2} Z_{seed}\left(\frac{\tau}{2}, \frac{\bar{\tau}}{2}, 2R\right) + \frac{1}{2} Z_{seed}\left(\frac{\tau+1}{2}, \frac{\bar{\tau}+1}{2}, 2R\right)$$

$$= \frac{1}{2} \sum_{m,n} d_m^{(s)} d_n^{(s)} e^{-\beta(E_m^{(s)} + E_n^{(s)}) + i\theta(P_m^{(s)} + P_n^{(s)})} + \frac{1}{2} \sum_m d_m^{(s)} e^{-2\beta E_m^{(s)} + 2i\theta P_m^{(s)}} + \sum_m d_m^{(s)} e^{-\beta E_m^{(2)} + i\theta P_m^{(2)}},$$

where in the last term we have used our previous result on the twisted sector spectrum and allowed momenta. The degeneracies of the various states simply follow from the seed degeneracies, with the given restriction. The first two terms contribute to the untwisted sector

partition function, as can be seen by further massaging them into

$$Z^{S_2}\big|_{untw} = \sum_{m<n} d_m^{(s)} d_n^{(s)} e^{-\beta(E_m^{(s)}+E_n^{(s)})+i\theta(P_m^{(s)}+P_n^{(s)})} + \sum_m \frac{d_m^{(s)}(d_m^{(s)}+1)}{2} e^{-2\beta E_m^{(s)}+2i\theta P_m^{(s)}}. \tag{53}$$

The contributing states belong to the symmetrized tensor product $(\mathcal{H}_{seed})^2/S_2$ and they take the form $(|E_n\rangle|E_m\rangle + |E_m\rangle|E_n\rangle)/\sqrt{2}$ for the first term, and $|E_m\rangle|E_m\rangle$ for the second. The degeneracies precisely correspond to those in the symmetrized tensor product of seed Hilbert spaces. Note that integer degeneracies are obtained only after including all contributions to the partition function. In the twisted sector, the degeneracies are the same as in the seed, subject to the projection (48).

All higher $N$ cases work similarly. The full energy spectrum is given by sums of the form

$$\sum_{cycles} E^{(w_i)}(R), \qquad \sum_{cycles} P^{(w_i)}(R), \tag{54}$$

which run over all the cycles in the various conjugacy classes $[g]$. The ground state energy in each sector varies from $-\pi cN/(6R)$ in the untwisted sector to $-\pi c/(6NR)$ in the maximally twisted one.

## 2.3 Spectrum of $T\bar{T}$ and $J\bar{T}$ symmetric product orbifolds

We would now like to apply these considerations to the specific examples of interest, namely $T\bar{T}$ and $J\bar{T}$ - deformed CFTs.

### $T\bar{T}$ - deformed CFTs

As reviewed in section 2.1, the partition function of a $T\bar{T}$ - deformed CFT - a Lorentzian QFT with a single dimensionful coupling, $\mu$ - depends not only on $\tau, \bar{\tau}$, but also on $R$ through the dimensionless combination $\mu/R^2$. This partition function is modular invariant in the generalised sense (10).

The partition function of the symmetric product orbifold of $T\bar{T}$ - deformed CFTs is obtained via a trivial application of Bantay's formula (36) to this seed, where the 'connected' contributions $\mathcal{Z}^{(n)}$ are given by particularizing (37) to the specific dependence on $R$ of the $T\bar{T}$ seed partiton function. Explicitly,

$$\mathcal{Z}_{T\bar{T}}^{(n)} = \frac{1}{n} \sum_{\ell|n} \sum_{0 \le r < \ell} Z_{T\bar{T}}^{seed}\left(\frac{n\tau}{\ell^2} + \frac{r}{\ell}, \frac{n\bar{\tau}}{\ell^2} + \frac{r}{\ell}, \frac{\mu}{\ell^2 R^2}\right). \tag{55}$$

As explained in the previous section, the modular invariance of $\mathcal{Z}^{(n)}$ follows from that of the seed partition function. It can be made particularly evident by rewriting $\mathcal{Z}^{(n)}$ in terms of Hecke operators. This result is in full agreement with the previous worldsheet computations [26,27] and the recent derivation [30].

As explained in our general analysis, this allows us to obtain the spectrum in the various twisted sectors. In particular, the energies in the $w$ - twisted sector are given by

$$E_{T\bar{T}}^{(w)}(R) = E_{T\bar{T}}^{(s)}(Rw) = \frac{Rw}{2\mu}\left(\sqrt{1 + \frac{4\mu E_{CFT}^{(s)}(Rw)}{wR} + \frac{4\mu^2\left(P^{(s)}(wR)\right)^2}{R^2 w^2}} - 1\right), \tag{56}$$

where we have opted to sometimes use the subscript 'CFT' to denote the undeformed fields, either in the seed or in the symmetric orbifold. The momenta are given by (50), which includes

the projection. One may further plug in the expression (46) for $E^{(s)}_{CFT}(wR)$ in terms of the conformal dimensions in the seed CFT, obtaining perfect agreement with the spectrum previously worked out in the literature [24, 26, 27]; note this brings additional powers of $w$ to the denominators. Alternatively, we may use (45) to replace $E^{(s)}_{CFT}(wR)$ by $E^{(w)}_{CFT}(R)$, and interpret (56) instead as the solution to a universal flow equation in the twisted sector with an effective parameter $\mu/w$, as was previously observed in [30]. The full spectrum of the symmetric orbifold of $T\bar{T}$ - deformed CFTs is given by sums over this kind of terms, as in (54), and is thus entirely determined by the spectrum of the seed undeformed CFT. Note that since the twisted sectors are equivalent to the seed theory on a cylinder of radius $Rw$, the torus partition function of the symmetric orbifold is well-defined provided the seed is, namely if the circumference of the torus satisfies (12).

Note the deformed spectrum may also be obtained directly from the flow equation. As usual, first order perturbation theory implies that

$$\partial_\mu E = \langle n| \sum_{I=1}^{N} T_I \bar{T}_I |n\rangle \,. \tag{57}$$

In the untwisted sector, $E = \sum_I E_I$, where each $E_I$ obeys the $T\bar{T}$ flow equation in the given copy. In the twisted sectors, one may uplift the flow equation to the covering space, which is a cylinder of circumference $Rw$. Since the right-hand-side of the flow equation is inversely proportional to the radius, it will pick up an overall factor of $1/w$

$$\partial_\mu E^{(w)} = \frac{1}{w}(E^{(w)}\partial_R E^{(w)} - P^{(w)2}/R) \,. \tag{58}$$

The solution will be given by the usual $T\bar{T}$ solution, but with $R \to Rw$ or, equivalently, $\mu \to \mu/w$. This agrees with (56), provided one takes into account the fact that the undeformed energies and momenta are already in the twisted sector, and thus are related via (50) to the ones of the seed.

## $J\bar{T}$-deformed CFTs

The case of $J\bar{T}$-deformed CFTs is more interesting, since the dependence on the couplings must be explicitly included in the partition function, as they transform non-trivially under modular transformations. This concerns both the $J\bar{T}$ coupling, $\lambda$, and the external chemical potential, $\nu$, for the left-moving charge. In addition, the partition function of the seed is not modular invariant, but instead transforms (23) as a Jacobi form of weight $(0,0)$ and index $(k,0)$, where $k$ is the level of the $U(1)$ Kac-Moody algebra.

Our goal is to understand the dependence on the parameters $\lambda/R$ and $\nu$ of the seed theories on the covering tori. Let us first treat the case of the left-moving chemical potential $\nu$. As explained, $\nu = \beta a^z = \tau_2 R a^z$, where $a^z$ is the gauge field that couples to the chiral left current. This coupling is held fixed when placing the seed theory on a covering torus; as a result, the chemical potential $\nu_\xi$ on the covering tori is given by

$$\nu_\xi = (\tau_\xi)_2 R_\xi a^z_\xi = \frac{n}{\ell^2} \tau_2 R \ell a^z = \frac{n}{\ell} \nu \,. \tag{59}$$

On the other hand, the dimensionless coupling $\lambda/R$ simply picks up the factor of $\ell$ that follows from dimensional analysis, $\lambda$ itself being the same.

The partition function of the symmetric product orbifold of $J\bar{T}$ - deformed CFTs is again given by Bantay's formula (36), where the individual contributions read

$$\mathcal{Z}^{(n)}_{J\bar{T}}\left(\tau, \bar{\tau}, \nu, \frac{\lambda}{R}\right) = \frac{1}{n} \sum_{\ell|n} \sum_{0 \le r < \ell} Z^{seed}_{J\bar{T}}\left(\frac{n\tau}{\ell^2} + \frac{r}{\ell}, \frac{n\bar{\tau}}{\ell^2} + \frac{r}{\ell}, \frac{n\nu}{\ell}, \frac{\lambda}{\ell R}\right) \,. \tag{60}$$

Given the modular transformation properties (23) of the seed partition function, we would now like to show that the partition function of the symmetric orbifold transforms in the same manner, but with $k \to Nk$, as follows from the fact that the level of the $U(1)$ current in the symmetric product is $N$ times larger than that of the seed. Remember from (24) that the seed partition function differs from a modular-invariant one by a factor of $\exp(\frac{\pi k \nu^2}{\tau_2})$. On the covering tori, we have

$$\frac{k \nu_\xi^2}{(\tau_\xi)_2} = \frac{nk\nu^2}{\tau_2}, \tag{61}$$

which is $\ell$ - independent. Thus, each $\mathcal{Z}^{(n)}$ will differ from a modular-invariant contribution by the exponential of such a factor. Since the symmetric orbifold partition function is a sum of products $\prod_n (\mathcal{Z}^{(n)})^{N_n}$ and $\sum_n nN_n = N$, we immediately note that the lack of modular invariance of each term in the sum in (36) is $kN\nu^2/\tau_2$, which is the same for every possible partition of the integer $N$. Thus, the transformation properties of the seed partition function under modular transformations determine those of the symmetric orbifold one, which transforms as in (23), but with $k \to Nk$. This connection can be made explicit by rewriting the result using Hecke operators, whose action can also be defined on the Jacobi forms of weight $(0, \kappa)$ and index $(k, 0)$ relevant to $J\bar{T}$ as [59]

$$T_n \phi(\tau, \bar{\tau}, \nu) = \frac{1}{n} \sum_{\substack{r, \ell \in \mathbb{Z}, \ell | n \\ 0 \leq r < \ell}} \frac{1}{\ell^\kappa} \phi\left(\frac{n\tau}{\ell^2} + \frac{r}{\ell}, \frac{n\bar{\tau}}{\ell^2} + \frac{r}{\ell}, \frac{n\nu}{\ell}\right), \tag{62}$$

and yields a Jacobi form of the same weight and index $(nk, 0)$. Expanding the $J\bar{T}$-deformed CFT partition function in a Taylor series in $\lambda$, the coefficient of $\lambda^\kappa$ is $R^{-\kappa}$ times a $(0, \kappa)$ Jacobi form of index $(k, 0)$. Thus, (60) can be written as

$$\mathcal{Z}_{J\bar{T}}^{(n)} = T_n Z_{J\bar{T}}^{seed}\left(\tau, \bar{\tau}, \nu, \frac{\lambda}{R}\right), \tag{63}$$

while the whole partition function is given by the right-hand side of (36).

Let us now understand the consequences of this formula for the spectrum of single-trace $J\bar{T}$ - deformed CFTs. We focus first on the $w$-twisted sector, for which $\ell = w$ and thus $\nu_\xi = \nu$, implying that the spectrum of left-moving charges is the same as in the seed. According to our general formula (45), the right-moving energies in the $w$-twisted sector read

$$E_{R,J\bar{T}}^{(w)}(R) = E_{R,J\bar{T}}^{(s)}(Rw) = \frac{4\pi}{k\lambda^2}\left(Rw - \lambda q^{[0]} - \sqrt{\left(Rw - \lambda q^{[0]}\right)^2 - \frac{\lambda^2 k}{2\pi} Rw\, E_{R,CFT}^{(s)}(Rw)}\right),$$

$$q^{(w)} = q^{[0]} + \frac{\lambda k}{4\pi} E_{R,J\bar{T}}^{(w)}(R), \tag{64}$$

where $q^{[0]}$ is the charge in the undeformed seed and $P^{(w)}(R)$ is given as before by (50), which entails a selection rule on the seed momenta. One can rewrite (64) in terms of the seed conformal dimensions by plugging in the explicit expressions (46) for the CFT finite-size energies. Alternatively, one can reinterpret $E_{R,CFT}^{(s)}(Rw)$ as $E_{R,CFT}^{(w)}(R)$ and view this expression as the solution to the $J\bar{T}$ flow equation in the $w$ - twisted sector, where the flow parameter is effectively $\lambda/w$ and the effective $U(1)$ level is $kw$. The above formula matches the worldsheet analysis of [9, 28, 60].[9]

For the contributions to $\mathcal{Z}^{(w)}$ that have $\ell \neq w$, note that the chemical potential on the covering torus is $w/\ell$ times that of the full symmetric orbifold. This implies that the chages

---

[9]In [9, 60] a different convention for the winding is used, such that $w_{here} = -w_{there}$. The charges in [9] are also related to ours by $\bar{q} = -q^{[0]}, \bar{Q} = -q^{(w)}$ and the definitions of left and right are exchanged.

in this sector are $w/\ell$ times the seed ones. This is in agreement with the fact that the states contributing to these terms take the form $(\otimes|E^{(\ell)}, q^{(\ell)}\rangle)^{w/\ell}$.

Finally, since the single-trace $J\bar{T}$ deformation also preserves the left conformal symmetry on the plane, we may again compute the corresponding left conformal dimensions, following the same steps as in the double-trace analysis of [43]. We obtain

$$h^{(w)}_{J\bar{T}}(\bar{p}) = h^{(w)}_{CFT} + \frac{\lambda \bar{p}\, q^{[0]}}{2\pi w} + \frac{k\lambda^2 \bar{p}^2}{16\pi^2 w} = \frac{h^{(s)}_{J\bar{T}}(\bar{p})}{w} + \frac{c}{24}\left(w - \frac{1}{w}\right), \tag{65}$$

where $h^{(w)}_{CFT}$ is the undeformed conformal dimension in the $w$-twisted sector, related to that in the seed CFT via (47), $h^{(s)}_{J\bar{T}}(\bar{p})$ is the momentum-dependent conformal dimension (19) in the $J\bar{T}$ seed QFT, and $\bar{p} \leftrightarrow E^{(w)}_R$ is the right-moving energy on the boosted cylinder. Overall, we obtain the standard CFT formula (47) for the orbifolded conformal dimensions, now taking into account the fact that the seed left-moving conformal dimension has been modified to (19). Another possible interpretation of this formula is as fractional spectral flow [61, 62] with parameter $\lambda\bar{p}/w$ in the $w$ - twisted sector, where the level is $k^{(w)} = kw$. The left-moving charge is simply given by (64) with $E^{(w)}_R \to \bar{p}$. This observation will be important for constructing the single-trace $J\bar{T}$ correlation functions in section 4.2.

## 2.4 Comments on the entropy

Given the partition function of the symmetric orbifold, the density of states can be readily extracted from it. In this section, we comment upon the entropy of both single-trace $T\bar{T}$ and $J\bar{T}$ - deformed CFTs, as well as its relation to that of the respective double-trace deformations.

### $T\bar{T}$ - deformed CFTs

In the $T\bar{T}$ case, the entropy of both the single-trace and double-trace deformation has been discussed in detail in the recent work [30], so we will be brief. For simplicity, we will set $P = 0$.

The analysis of [30] closely follows that of [63] for the case of two-dimensional CFTs. One of their results is that the entropy of a large $N$ symmetric product orbifold of $T\bar{T}$ - deformed CFTs presents two regimes[10]

$$S_{SPO\,of\,T\bar{T}}(E) = \begin{cases} R(E - E_{vac}), & \text{for} \quad E_{vac} \lesssim E < E_c, \\ 2\pi\sqrt{\frac{c^{(s)}E}{6\pi}(RN + \mu E)}, & \text{for} \qquad E > E_c, \end{cases} \tag{66}$$

with a sharp transition between them. Here

$$E_c = -\frac{E_{vac}}{1 + 2\mu E_{vac}/NR}, \quad \text{with} \quad E_{vac} = \frac{NR}{2\mu}\left(\sqrt{1 - \frac{2\pi\mu\, c^{(s)}}{3R^2}} - 1\right), \tag{67}$$

and, in this subsection only, $c^{(s)}$ is the central charge of the seed CFT.

Thus, the behaviour of the entropy is Hagedorn in an intermediate range of energies and then transitions to the universal $T\bar{T}$ behaviour (Cardy $\to$ Hagedorn) at high energies. Note that, since the partition function of single-trace $T\bar{T}$ - deformed CFTs only makes sense on a circle of circumference $R > R_{min}$, with $R_{min}$ given in (12), the slope of the high-energy Hagedorn regime is always less than the slope of the intermediate Hagedorn one. It is interesting to ask whether the two Hagedorn regimes need to be separated by a Cardy one. The crossover

---

[10]Note that our conventions differ from those in [30] by $\mu_{here} = 2\pi\mu_{there}$ and also $R_{here} = 2\pi R_{there}$.

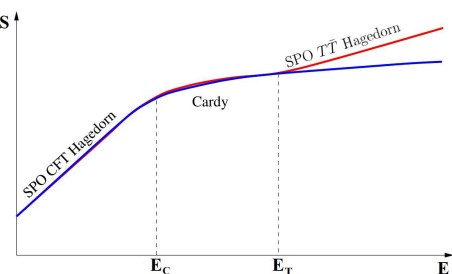

(a)   The deformation parameter $\mu$ is small enough so that there is a well-separated Cardy regime.

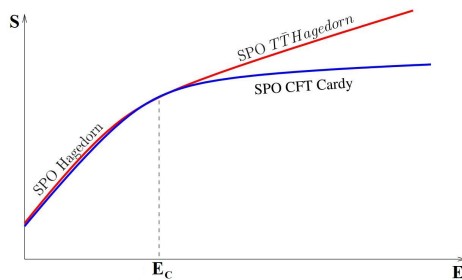

(b)   When $\frac{4R^2}{3\pi c} < \mu \le \frac{3R^2}{2\pi c}$ , there is a direct transition between the two Hagedorn regimes.

Figure 2: Plot of the entropy as a function of energy for a large $N$ symmetric orbifold of $T\bar{T}$ - deformed CFTs (red), as compared to the entropy of a symmetric orbifold of CFTs with the same central charge (blue).

between Cardy and Hagedorn behaviour in the universal regime $E > E_c$ occurs at an energy scale $E_T \sim NR/\mu$. The ratio of this scale[11] to $E_c$ is

$$\frac{E_T}{E_c} = \frac{RN}{\mu E_c} = \frac{2\sqrt{1-x}}{1-\sqrt{1-x}}, \qquad x \equiv \frac{2\pi\mu c^{(s)}}{3R^2} \le 1, \tag{68}$$

which is a monotonously decreasing function of $x \propto \mu$, from infinity at $\mu = 0$ to zero at the maximum allowed $\mu$ for that compactification radius ($x = 1$). For $x < 8/9$, the transition from the Cardy to the second Hagedorn regime occurs after the high-energy universal regime sets in (see figure 2a). However, when $8/9 < x \le 1$, then the Hagedorn term dominates from the beginning, and we have a Hagedorn to Hagedorn transition (see figure 2b). This regime is possible precisely because the value of $E_c$ at which the universal regime kicks in depends on $\mu$; otherwise, the above ratio would be $4/x$, which never becomes less than one in the given range.

As shown in [63] (henceforth HKS) using modular invariance, in a two-dimensional CFT with a large central charge and a sparse light spectrum, the entropy is universally given by Cardy's formula for energies $E > \pi c/6R$, and satisfies a Hagedorn upper bound for smaller energies, which is saturated by symmetric product orbifold CFTs. Closely following this analysis, [30] (henceforth ASY) showed that a similar statement holds in double-trace $T\bar{T}$-deformed CFTs with a large central charge and an appropriately sparse light spectrum: the high-energy density of states is given by the universal formula (11) for $E > E_c$, where $E_c$ is given by (67) with $N = 1$ and $c^{(s)}$ replaced by $c$. Below $E_c$, the entropy satisfies an upper bound, given by

$$S_{ASY\ T\bar{T}\ bnd.}(E) = R(E - E_{vac}), \qquad E < E_c, \tag{69}$$

where $E_{vac}$ is given by (67) with $N = 1, c^{(s)} \to c$. From (66), one can see that the symmetric product orbifold of $T\bar{T}$ - deformed CFTs can be thought of precisely saturating the bound for $c = Nc^{(s)}$, provided we replace $\mu_{s.tr} \to N\mu_{d.tr}$, as follows from the matching of the entropies in the high-energy universal regime $E > E_c$. Thus, the red curves in the plots above can also be interpreted as upper bounds on the entropy of double-trace $T\bar{T}$ - deformed CFTs with central charge $Nc$, coupling $\mu/N$ and a certain sparseness condition on their light states.

Another way to obtain an upper bound on the density of states of a $T\bar{T}$ - deformed CFT is to use the fact that in such a theory the degeneracies are identical, at leading order, to those in the seed CFT, but they are measured in a different variables. The degeneracies of a CFT

---

[11]In units of $N\pi c/3R$, the energies are $E_T = 2/x$, $E_c = (1/\sqrt{1-x} - 1)/x$ and $E_{vac} = (\sqrt{1-x} - 1)/x$.

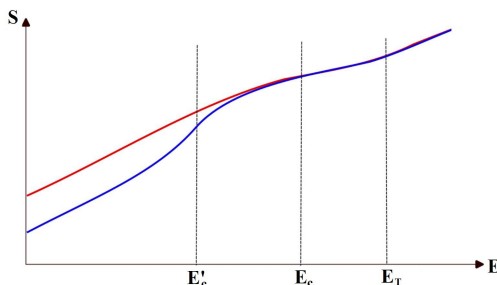

Figure 3: Comparison of the ASY (red) and the $T\bar{T}$ deformation of the HKS (blue) bounds on the entropy of $T\bar{T}$ - deformed CFTs with a large central charge. The latter bound is stronger due to the associated stricter sparseness condition on the light states. These plots can also be interpreted as the entropy of a single-trace (red) and double-trace (blue) $T\bar{T}$ - deformed entropy of a symmetric product orbifold CFT.

with a sparse light spectrum satisfy, as discussed, the HKS bound [63], which is saturated by a symmetric orbifold CFT. Simply plugging in the relationship between the undeformed and deformed energies into this universal bound, one obtains

$$S_{T\bar{T}\,of\,HKS\,bnd.}(E) = \begin{cases} E(R+\mu E) + \frac{\pi c}{6}, & \text{for} \quad E < E'_c, \\ 2\pi\sqrt{\frac{cE}{6\pi}(R+\mu E)}, & \text{for} \quad E > E'_c, \end{cases} \tag{70}$$

where $E'_c$ is given by

$$E'_c = \frac{R}{2\mu}\left(\sqrt{1 + \frac{2\pi\mu c}{3R^2}} - 1\right). \tag{71}$$

For $c = Nc^{(s)}$, this represents the entropy of a double-trace $T\bar{T}$ - deformed symmetric product orbifold CFT, which was studied in [36]. From (70) it follows that this exhibits an intermediate super-Hagedorn regime in the microcanonical ensemble, smoothly crossing over to the usual $T\bar{T}$ Hagedorn behaviour at high energies. The specific heat in the intermediate super-Hagedorn regime is negative, and the system exhibits a first-order phase transition when coupled to a heat bath.

Let us now check whether this super-Hagedorn intermediate regime is consistent with the bound derived in [30]. From (71) and (67) with $N = 1$, $c^{(s)} \to c$, it is easy to check that $E'_c < E_c$, so the universal high-energy regime is reached by the $T\bar{T}$ - deformed HKS bound before the prediction (66) of the ASY bound. By evaluating (66) and (70) at $E'_c$, we obtain:

$$S_{ASY\,T\bar{T}\,bnd}(E'_c) > S_{T\bar{T}\,of\,HKS\,bnd}(E'_c). \tag{72}$$

One can check that this is also the case at $E = 0$. Since the double-trace $T\bar{T}$ entropy is monotonously increasing in this interval, and at the end of the super-Hagedorn regime the double-trace $T\bar{T}$ entropy is smaller than the AS bound, we can conclude that

$$S_{ASY\,T\bar{T}\,bnd}(E) > S_{T\bar{T}\,of\,HKS}(E), \quad \forall E \le E_c, \tag{73}$$

as depicted in figure 3. To obtain this result, it was important that the transition between the two regimes in (66) is given by (67), as opposed to simply replacing the CFT energies by the $T\bar{T}$ - deformed ones.

Thus, the entropy bound obtained by $T\bar{T}$ - deforming the HKS bound for a CFT with a large central charge and a sparse light spectrum is tighter than that obtained in [30] directly from modular invariance in the $T\bar{T}$ - deformed CFT. This can be explained by the fact that the

sparseness condition imposed in [30] on the light spectrum of $T\bar{T}$ - deformed CFTs is less restrictive than the $T\bar{T}$ deformation of the HKS sparseness condition on the seed CFT, which results in a less strong bound at intermediate energies. Given that these bounds can also be interpreted as the entropy of a single-trace and, respectively, double-trace $T\bar{T}$ - deformed SPO CFT at appropriately scaled parameters, the above relation can also be written as

$$S_{SPO\,of\,T\bar{T}}(E) > S_{T\bar{T}\,of\,SPO}(E), \quad \forall E \leq E_c. \tag{74}$$

We explicitly note that, while the entropies of the deformed theories agree at large energy, they differ in the intermediate regime.

## $J\bar{T}$ - deformed CFTs

Let us now discuss the single-trace $J\bar{T}$ deformation, concentrating on the case of a purely chiral $U(1)$ current, for concreteness. As discussed in section 2.1 for the double-trace case, there exists a sliver of energies in the undeformed CFT, given in figure 1, for which the deformed energies are real and can become arbitrarily high; in the chiral case, the high-energy behaviour of the entropy in the right-moving sector is Hagedorn. The starting point of the Hagedorn regime may be determined, as in our previous discussion, by translating the onset of the universal regime in CFT to $J\bar{T}$ variables. We will henceforth assume that the energies in the deformed theory are sufficiently high, so that the Hagedorn formula (26) applies, and deduce the high-energy behaviour of the entropy in single-trace $J\bar{T}$ - deformed CFTs from the fact that the twisted-sector degeneracies are directly determined from the seed $J\bar{T}$ degeneracies via (60).

Let us concentrate on the contribution of a single twisted sector of length $n$, which is given by the $\ell = n$ term in (60). We will moreover only write the right-moving piece of the entropy, as the left-moving one is identical to that of a CFT with a fixed charge. Since in this sector, the degeneracy is the same as that of the seed at the same right-moving energy and charge, but on a cylinder $n$ times larger, we find

$$S_R^{(n)}(E_R, q) = 2\pi\sqrt{\frac{cE_R}{12\pi}\left((nR - \lambda q) + \frac{\lambda^2 k E_R}{8\pi}\right)}. \tag{75}$$

It is interesting to compare the degeneracy of the maximally-twisted sector, $S_R^{(N)}(E_R, q)$, to that of the untwisted one with the same total energy and charge

$$S_R^{untw}(E_R, q) = 2\pi N\sqrt{\frac{c}{12\pi}\left(\frac{E_R}{N}(R - \frac{\lambda q}{N}) + \frac{\lambda^2 k E_R^2}{8\pi N^2}\right)} = 2\pi\sqrt{\frac{cE_R}{12\pi}\left((NR - \lambda q) + \frac{\lambda^2 k E_R}{8\pi}\right)}, \tag{76}$$

where we have assumed that, on average, each copy posesses an equal share of the energy and charge, as this distribution maximizes the entropy. Thus, we find the same leading behaviour across sectors, similarly to the case of CFTs and $T\bar{T}$ - deformed CFTs.

In order to establish which sector dominates, one would need to consider subleading corrections to the entropy, which can be analysed by translating the CFT results [63] to deformed $J\bar{T}$ variables. In addition, one would need to ascertain that all sectors may in principle compete in the given energy range: for example, if the energies in the maximally twisted sector are real, then one should check that so are the contributing energies from the untwisted one, at least on average. In the maximally twisted sector, the reality constraint on the undeformed (twisted) energy and charge is

$$\frac{(q^{[0]})^2}{k} - \frac{PRN}{2\pi} \leq \frac{RNE_R^{[0]}}{2\pi} \leq \frac{1}{k}\left(\frac{RN}{\lambda} - q^{[0]}\right)^2, \tag{77}$$

where we have simply translated the double-trace constraints discussed in section 2.1 to a cylinder of circumference $RN$. This should be compared with the reality condition for a state in the seed CFT that has, on average, energy $E_R^{[0]}/N$, momentum $P/N$ and charge $q^{[0]}/N$; it is straightforward to check that the two conditions are the same. We conclude that if we concentrate on a range of energies and charges in the undeformed symmetric product orbifold CFT such that the deformed energies in the untwisted sector are real, the ones in the maximally twisted sector will also be real, and viceversa.

It would be interesting to understand the validity of these formulae at lower energies, as well as to establish alternate universal bounds based on modular methods, analogous to that of [30] for $T\bar{T}$. Given the relation (63) between the full $\mathcal{Z}_{J\bar{T}}^{(n)}$ and generalised Hecke operators, one would expect its behaviour to be universal and fixed by modular invariance. However, such arguments are harder to invoke in $J\bar{T}$ - deformed CFTs, despite the formal modular covariance (23) of the partition function, due to the generic presence of imaginary energy states. Should one be able to apply such methods, it would be interesting to understand the energy at which different twisted sectors enter the universal regime. Note also that, as can be seen from (60), $\mathcal{Z}_{J\bar{T}}^{(n)}$ not only captures the contributions from the $n$-twisted sector, but also special contributions from sectors of lower twist; however, these other contributions with $\ell < N$ are exponentially subleading, since their degeneracy is given by $d(E\ell/N, R\ell)$ i.e. the degeneracy of tensor products of *identical* states of lower energy.

Finally, if the $U(1)$ current $J$ is non-chiral, then the results are very similar to those in a CFT with both left- and right-moving $U(1)$ charges, one simply needs to ensure that the undeformed energies lie in the correct regions, given in figure 1b, to yield real deformed energies and charges.

# 3 Flow of the states and symmetries

One of the most remarkable properties of the $T\bar{T}/J\bar{T}$ deformation of a two-dimensional CFT is that it preserves the Virasoro - and, if present, Kac-Moody - symmetries of the undeformed theory, despite rendering it non-local. These symmetries were studied from both a holographic and field-theoretic perspective in [37–40, 50, 51, 64]. The most rigorous way to date to ascertain their presence is by transporting the symmetry generators of the undeformed CFT along the irrelevant flow.[12]

In this section we show, as intuited in [38], that an analogous analysis predicts the existence of Virasoro ($\ltimes$ Kac-Moody) symmetries in single-trace $T\bar{T}$ and $J\bar{T}$ - deformed CFTs. For completeness, we start this section with a brief review of these symmetries in double-trace $T\bar{T}$ and $J\bar{T}$ - deformed CFTs, then discuss the flow operator and the symmetries of the single-trace variant. In addition, we show that the fractional Virasoro and Kac-Moody generators present in the undeformed symmetric orbifold CFT can also be defined in the deformed theories and are conserved.

## 3.1 Brief review of the symmetries of $T\bar{T}$ and $J\bar{T}$ - deformed CFTs

The simplest way to show that $T\bar{T}$ and $J\bar{T}$ - deformed CFTs possess Virasoro (and, if initially present, also Kac-Moody) symmetries is by transporting the original Virasoro and Kac-Moody generators along the $T\bar{T}/J\bar{T}$ flow. The operator used to define the transport is precisely the one that controls the flow of the energy eigenstates under the $T\bar{T}/J\bar{T}$ deformation.

Concretely, since the $T\bar{T}/J\bar{T}$ deformations are adiabatic, one can define for each a flow

---

[12]For $T\bar{T}$ - deformed CFTs, this perspective clears up a number of ambiguities present in the earlier analysis [37].

operator

$$\partial_\lambda |n_\lambda\rangle = \mathcal{X}|n_\lambda\rangle \,, \tag{78}$$

where $|n_\lambda\rangle$ are the energy eigenstates in the deformed theory with generic coupling $\lambda$, which in this section can represent either the coupling $\mu$ of $T\bar{T}$, or $\lambda$ of $J\bar{T}$. The flow operator is determined via first order perturbation theory of the states $|n_\lambda\rangle$, and can be shown to satisfy

$$[H, \mathcal{X}] = \int d\sigma \, \mathcal{O}_{T\bar{T}/J\bar{T}} - \text{diag} \,, \tag{79}$$

where "diag" stands for the diagonal elements of the integrated $T\bar{T}/J\bar{T}$ operator in the energy eigenbasis. If the energy levels are degenerate, as is always the case if we start from a CFT, then it represents the matrix elements of the $T\bar{T}/J\bar{T}$ operator on the union of the degenerate subspaces. These matrix elements are only non-zero on the diagonal, as follows from the fact that the $T\bar{T}$ and the $J\bar{T}$ deformation do not break any existing degeneracies. The operators $\mathcal{X}_{T\bar{T}}, \mathcal{X}_{J\bar{T}}$ are known explicitly, at least in the classical limit [38,39,51,65], though their specific expression is not needed for the argument that follows.

The flow operator can be used to define the generators $\widetilde{L}_m^\lambda, \widetilde{\bar{L}}_m^\lambda$ (and their Kac-Moody counterparts $\widetilde{J}_m^\lambda, \widetilde{\bar{J}}_m^\lambda$) as solutions of the flow equations [38,66]

$$\partial_\lambda \widetilde{L}_m^\lambda = [\mathcal{X}, \widetilde{L}_m^\lambda], \qquad \partial_\lambda \widetilde{\bar{L}}_m^\lambda = [\mathcal{X}, \widetilde{\bar{L}}_m^\lambda], \tag{80}$$

with the initial condition $\widetilde{L}_m^0 = L_m^{CFT}, \widetilde{\bar{L}}_m^0 = \bar{L}_m^{CFT}$, etc. From this definition it follows that at any point along the flow, the algebra of these generators will consist of two commuting copies of the Virasoro ($\ltimes$ Kac-Moody) algebra, with the same central charge as that of the undeformed CFT.

So far, this is just a definition. In order for these operators to generate symmetries of $T\bar{T}$ and $J\bar{T}$- deformed CFTs, one needs to show that they are conserved. The conservation equation for the Schrödinger picture operators reads

$$\frac{\partial \widetilde{L}_m^\lambda}{\partial t} + \frac{i}{\hbar}[H, \widetilde{L}_m^\lambda] = 0 \,. \tag{81}$$

The commutator of the flowed generators with the Hamiltonian can be computed from first principles using the universality of the spectrum of $T\bar{T}/J\bar{T}$- deformed CFTs, and takes the simple form [38]

$$[\widetilde{L}_m^\lambda, H] = \alpha_m \widetilde{L}_m^\lambda, \qquad [\widetilde{\bar{L}}_m^\lambda, H] = \bar{\alpha}_m \widetilde{\bar{L}}_m^\lambda \,, \tag{82}$$

where $\alpha_m, \bar{\alpha}_m$ are operator-valued functions that depend on the Hamiltonian and other conserved charges. Their expressions for the two types of theories we consider are

$$T\bar{T} \; : \; \alpha_m(H,P) = \bar{\alpha}_m(H,-P) = \frac{1}{2\mu}\left(\sqrt{(R+2\mu H)^2 + \frac{4\mu m\hbar(R+2\mu P)}{R} + \frac{4\mu^2 m^2 \hbar^2}{R^2}} - (R+2\mu H)\right),$$

$$J\bar{T} \; : \; \alpha_m = \frac{m\hbar}{R}, \qquad \bar{\alpha}_m(Q) = 2\frac{R - \lambda Q - \sqrt{(R-\lambda Q)^2 - \hbar k m \lambda^2}}{k\lambda^2}, \qquad Q \equiv \tilde{J}_0 + \frac{\lambda k}{2} H_R \,, \tag{83}$$

where $\hbar$ is Planck's constant. Note that in $J\bar{T}$-deformed CFTs, $\alpha_m$ is a $c$-number, which is related to the fact that these theories are local on the left-moving side. The conservation equation (81) is immediately satisfed if we assign the following explicit time dependence to the Schrödinger picture generators

$$\widetilde{L}_m^\lambda(t) = e^{i\alpha_m t}\widetilde{L}_m^\lambda(0), \qquad \widetilde{\bar{L}}_m^\lambda(t) = e^{i\bar{\alpha}_m t}\widetilde{\bar{L}}_m^\lambda(0) \,, \tag{84}$$

where $\widetilde{L}^\lambda_m(0)$ are the solutions to the flow equation (82). The Kac-Moody generators are treated in an entirely analogous manner. Thus, the operators we constructed correspond to conserved charges of the theory, and the Virasoro($\ltimes$ Kac-Moody) algebra they obey represents its symmetry algebra. Note this conservation argument is the same as the one used in standard two-dimensional CFTs; the only difference is that now $\alpha_m, \bar{\alpha}_m$ are operators, whereas in the CFT case they were $c$-numbers.

The above argument, while valid at the full quantum-mechanical level, is somewhat abstract, and does not lead to an intuitive picture of the action of these symmetries. It is also not clear whether this basis of generators is the one that acts most naturally on the fields in the theory. For example, in the case of $J^1 \wedge J^2$ deformation of two-dimensional CFTs - an exactly marginal deformation of the Smirnov-Zamolodchikov type - the Virasoro generators obtained via an analogous flow argument can be explicitly shown to *differ* from the Virasoro generators of standard conformal symmetries [42].

In both $T\bar{T}$ and $J\bar{T}$ - deformed CFTs, this question can be addressed very concretely at the classical level, by explicitly constructing the flow operators. In $J\bar{T}$ - deformed CFTs, which are conformal in the standard sense on the left-moving side, one again finds that the flowed left-moving Virasoro and Kac-Moody generators $\widetilde{L}_n, \widetilde{J}_n$ *differ* from the Virasoro - Kac-Moody generators $L_n, J_n$ of left conformal and affine transformations. This difference may be characterised as a "spectral flow by $\lambda H_R$", where $H_R$ is the right-moving Hamiltonian

$$L_n = \widetilde{L}_n + \frac{\lambda H_R \widetilde{J}_n}{R} + \frac{\lambda^2 k H_R^2}{8\pi R} \delta_{n,0}\,, \qquad J_n = \widetilde{J}_n + \frac{\lambda k}{4\pi} H_R \,\delta_{n,0}\,, \qquad (85)$$

where we have dropped the $\lambda$ index on the generators and assumed that the explicit classical relation can be extended to the full quantum level. Note that in our conventions, the Virasoro generators are dimensionful; in particular $L_0 = H_L, \bar{L}_0 = H_R R_\nu / R$. A similar relation holds for the right-movers

$$\bar{L}_n = \widetilde{\bar{L}}_n + \frac{\lambda : H_R \widetilde{\bar{J}}_n :}{R} + \frac{\lambda^2 k H_R^2}{8\pi R} \delta_{n,0}\,, \qquad \bar{J}_n = \widetilde{\bar{J}}_n + \frac{\lambda k}{4\pi} H_R \,\delta_{n,0}\,. \qquad (86)$$

At least classically, the right-moving generators $\bar{L}_m$ implement infinitesimal field-dependent coordinate transformations, as one may note from their expression

$$\bar{L}_m = \frac{R_\nu}{R} \int d\sigma \, e^{-2\pi i m \hat{v}} \mathcal{H}_R\,, \quad v \sim \sigma - t - \lambda \phi\,, \qquad (87)$$

where $\phi$ is roughly the bosonisation of the $U(1)$ current with its zero mode removed - see [38] for the full expression - $R_\nu$ is the (field-dependent) circumference of the above field-dependent coordinate and $\hat{v} = v/R_\nu$. The $\bar{J}_m$ implement similar field-dependent affine $U(1)$ transformations. As discussed at length in [42], it is the $L_n, \bar{L}_n$ - rather than their tilded counterparts - that act naturally on the operators in the theory, and which should therefore be considered as the 'physical' symmetry generators in the theory; in particular, it is the $L_m, \bar{L}_m$ that have a simple integral expression in terms of the conserved currents in the theory, at least at the classical level. The algebra of these generators follows from their definition (85) - (86) and the fact that the tilded generators satisfy two copies of the Virasoro $\ltimes$ Kac-Moody commutation relations. One finds that the algebra of the left-moving generators $L_n, J_n$ is again Virasoro $\ltimes$ Kac-Moody, while that of $\bar{L}_n, \bar{J}_n$ is a non-linear modification of the right Virasoro $\ltimes$ Kac-Moody algebra that does not commute with the left generators. It has been worked out explicitly in [38] using identities such as

$$[\bar{L}_m, \bar{\alpha}_n] = (\bar{\alpha}_{m+n} - \bar{\alpha}_m - \bar{\alpha}_n)\bar{L}_m\,, \qquad (88)$$

which follow from the special properties of the functions $\bar{\alpha}_m$ defined in (83).

The $T\bar{T}$ case appears to work similarly, though to date is less understood. In the classical limit, the flowed generators take the form

$$\widetilde{L}_m^{cls} = \frac{R_u}{R} \int d\sigma \, e^{2\pi i m \hat{u}} \mathcal{H}_L \,, \qquad \widetilde{\bar{L}}_m^{cls} = \frac{R_v}{R} \int d\sigma \, e^{-2\pi i m \hat{v}} \mathcal{H}_R \,, \tag{89}$$

where $\mathcal{H}_{L,R}$ are the left/right-moving Hamiltonian densities, $R_{u,v} \equiv R + 2\mu H_{R/L}$, $\hat{u} \equiv \frac{u}{R_u}$, $\hat{v} \equiv \frac{v}{R_v}$ and $u, v$ are the $T\bar{T}$ field-dependent coordinates that emerge[13] from the solution to the flow equation

$$u \sim \sigma + t + 2\mu \int^\sigma \mathcal{H}_R \,, \qquad v \sim \sigma - t + 2\mu \int^\sigma \mathcal{H}_L \,, \tag{90}$$

whose full definition can be found in [39]. Classically, (89) generate field-dependent coordinate transformations. Note, however, that the generators of the symmetries in the natural Fourier basis are, rather, the so-called "unrescaled" generators [39]

$$Q_m = \frac{R \widetilde{L}_m}{R_u} \,, \qquad \bar{Q}_m = \frac{R \widetilde{\bar{L}}_m}{R_v} \,. \tag{91}$$

Their algebra is given by a non-linear modification of the Virasoro algebra

$$[Q_m, Q_n] = \frac{2\pi\hbar}{R_u}(m-n)Q_{m+n} + \frac{8\pi\hbar\mu^2 H_R}{R R_H R_u}(m-n)Q_m Q_n + \frac{\pi^2 c \hbar m^3}{3 R_u^2}\delta_{m+n} + \mathcal{O}(\hbar^2) \,, \tag{92}$$

where $R_H = R + 2\mu H$ and the $\mathcal{O}(\hbar^2)$ terms and higher can be computed once the full quantum relation between the $Q_m$ and $\widetilde{L}_m$ is given.[14] This result was also confirmed holographically by the analysis of [40].

We end our review of the extended symmetries of $T\bar{T}$ and $J\bar{T}$ - deformed CFTs by discussing the charges associated to integrability, which have been known to be preserved by these deformations since the original work of [11]. In a $\text{CFT}_2$, these charges correspond to the so-called KdV charges [67], which are associated to currents constructed from higher powers of the (anti)holomorphic stress tensor. For example, the first two non-trivial KdV charges are given by (for $R = 2\pi$)

$$I_3 = 2 \sum_{n=1}^\infty L_{-n} L_n + f(L_0) \,, \tag{93}$$

$$I_5 = \sum_{n_1+n_2+n_3=0} : L_{n_1} L_{n_2} L_{n_3} : + \sum_{n=1}^\infty \left( \frac{c+11}{6} n^2 - 1 - \frac{c}{4} \right) L_{-n} L_n + \frac{3}{2} \sum_{n=1}^\infty L_{1-2n} L_{2n-1} + g(L_0) \,, \tag{94}$$

where $f(L_0), g(L_0)$ are quadratic functions of $L_0$, whose explicit expressions are given e.g. in [68], but are not relevant for our purposes. As discussed in [39], one can easily define the flowed KdV charges $\widetilde{I}_s$ by requiring that they satisfy a homogenous flow equation analogous

---

[13]More precisely, one can view $\widetilde{L}_m^{cls}$ as the Fourier modes of a current $\mathcal{H}_L(\sigma)$ that satisfies the flow equation $\partial_\mu \mathcal{H}_L = [\mathcal{X}_{T\bar{T}}, \mathcal{H}_L]$ and is chiral. The full solution is given in [39]. Upon integrations by parts, the Fourier modes can be put in the form (89), where the expressions for $\hat{u}, \hat{v}$ simply follow from the flow equation. In this sense, the field-dependent coordinates are "emergent".

[14]An example of a fully quantum relation between the two that takes into account operator ordering is given by the symmetrization of (91), by letting $\widetilde{L}_m = Q_m + \mu H_R Q_m + \mu Q_m H_R$. Note that the algebra of these $Q_m$ generators is entirely determined by their definition and the fact that the $\widetilde{L}_m$ satisfy a Virasoro algebra. It can be worked out using identities of the form (88), which also apply to the $T\bar{T}$ generators for the appropriate choice of $\alpha_m$.

to (80); its solution is simply given by replacing $L_n \to \widetilde{L}_n$ in the expressions above. A rescaled version of these charges satisfies the flow equations discussed in [69].

For this prescription to make sense, we should also show that the flowed KdV charges are conserved, i.e. they commute with the Hamiltonian. This commutator can be computed using (82) and the commutation relation (88), which turns out to also hold in $T\bar{T}$ - deformed CFTs for the corresponding $\alpha_m$. One can check explicitly that the commutator of $H$ with any product of $\widetilde{L}_m$ whose indices sum to zero is proportional to $\alpha_0$, which vanishes. Since this is true term by term, we conclude that all the KdV charges constructed via the flow remain conserved.

## 3.2 Brief review of the symmetries of symmetric orbifold CFTs

Let us now review a few facts about the extended symmetries of symmetric product orbifolds of two-dimensional CFTs. These include the standard Virasoro symmetries, the fractional Virasoro generators that act in the twisted sectors, as well as higher spin symmetries [41], whose structure has yet to be fully understood.

In this subsection, we concentrate mostly on the Virasoro symmetries and their fractional counterparts, which we will subsequently generalise to single-trace $T\bar{T}$ and $J\bar{T}$ - deformed CFTs. Given that the action of the symmetric product orbifold CFT is simply the sum over individual CFT copies, the (e.g., holomorphic) stress tensor of the theory (classically denoted $\mathcal{H}_L^{[0]}$) is a sum over copies

$$T(\sigma) = \sum_{I=1}^{N} T^I(\sigma), \tag{95}$$

where we are working on a fixed time slice, say $t = 0$. The Virasoro generators correspond to the Fourier modes of the stress tensor, whose quantisation depends on the boundary conditions the latter obeys. In the untwisted sector, the stress tensor in each copy obeys periodic boundary conditions, so we can write

$$L_m = \sum_{I=1}^{N} L_m^I. \tag{96}$$

Note that in our conventions, the Virasoro generators are dimensionful (with dimensions of mass) for reasons that will become clear in the sequel.

In the twisted sectors, the boundary conditions relate operators from the different copies. We will focus on $\mathbb{Z}_w$ cyclic orbifolds, which are the building blocks of the twisted-sector operators. Since we will only be interested in single-trace operators, we will consider only one cycle of length $w$; for simplicity, we will assume that the copies of the symmetric orbifold involved are $1, ..., w$, in this order. The final expressions need of course to be symmetrized over all possible choices of the $w$ copies, to ensure $S_N$ - invariance. Since we will be working on a fixed time slice, all dependence on the time coordinate will be dropped.

We thus consider $w$ copies, $\phi_I(\sigma)$, of a (bosonic) field $\phi$ - now taken to be generic - with the boundary condition $\phi_I(\sigma + R) = \phi_{I+1}(\sigma)$, $\forall I$ defined mod $N$. This twisted boundary condition can be thought of as being due to the insertion of a $w$-twist operator at Euclidean time $\tau = -\infty$ on the cylinder. It is natural to consider the combinations

$$\phi^{(k)}(\sigma) = \sum_{I=1}^{w} \phi_I(\sigma) e^{-2\pi i k(I-1)/w}, \quad \phi^{(k)}(\sigma + R) = e^{2\pi i k/w} \phi^{(k)}(\sigma), \quad k \in [0, w-1], \tag{97}$$

which transform diagonally under $\mathbb{Z}_w$. Thus, each field in the theory will give rise to $w$ fields with twisted boundary conditions. The moding of each of them will be $n - k/w$ for $k \in [0, w-1]$ and $n \in \mathbb{Z}$; ultimately, this will result in a field with all possible fractional modes. The fractional

modes can be expressed via a Fourier transform as

$$\phi_{m/w} = \int_0^R d\sigma\, e^{2\pi i \sigma m/Rw} \phi^{(k)}(\sigma)\big|_{k=-m\,mod\,w} = \int_0^R d\sigma \sum_{I=1}^w \phi_I(\sigma)\, e^{2\pi i m(\sigma+(I-1)R)/Rw}. \tag{98}$$

Conversely, the modes of the field in a single copy can be obtained by inverting the relation above

$$\phi_I(\sigma) = \frac{1}{wR} \sum_{m=-\infty}^{+\infty} \phi_{m/w}\, e^{-2\pi i m \sigma/Rw}\, e^{-2\pi i m(I-1)/w}. \tag{99}$$

This is not quite an operatorial relation, for two reasons: first, it can only be used when acting on states from the $w$ - twisted sector; second, the left-hand-side is not an operator in the symmetric orbifold, because it is not gauge-invariant. This relation can nevertheless be used to construct operators that act on the twisted sector. For example, the action of the single-trace untwisted operator $\sum_I \phi_I(\sigma)$ on this sector is given by the Fourier sum where only integer modes appear, as all $m$ in (99) that are not multiples of $w$ will be projected out by the sum over $I$. Even so, note that the integer modes in this sector are not associated to any particular copy, but rather correspond to Fourier modes of the entire sum over operators, which satisfies periodic boundary conditions by construction.

The modes (98) can be related to the integer modes on a cylinder of circumference $Rw$. This is simply achieved by letting the coordinate $\tilde{\sigma}$ on this larger cylinder equal $\tilde{\sigma} = \sigma + (I-1)R$ in the $I^{th}$ patch given by $\tilde{\sigma} \in [R(I-1), RI)$, and defining a field on this covering space via $\phi^{cov}(\tilde{\sigma}) = \phi_I(\sigma)$ on the given patch. One can easily see that on the $I^{th}$ patch, $\phi^{cov}(\tilde{\sigma}) = \phi_I(\sigma) = \phi_1(\sigma + (I-1)R)$, $\forall I$, and thus $\phi^{cov}(\tilde{\sigma})$ is simply the field of the seed QFT, defined on this larger cylinder with periodic boundary conditions. The fractional Fourier modes discussed above become simply the Fourier modes of the field of the seed QFT on this larger cylinder

$$\phi_{m/w} = \int_0^{Rw} d\tilde{\sigma}\, \phi^{cov}(\tilde{\sigma})\, e^{2\pi i m \tilde{\sigma}/wR} = \phi_m^{cov}. \tag{100}$$

Note that so far no use of conformal symmetry was made, but rather we simply sewed the various copies of the fields into a single copy on the covering space.[15]

This procedure may be applied to any field in the theory; in particular, it may be applied to the stress tensor, which results in an infinite set of fractional Virasoro modes $L_{m/w}$. The algebra of these modes defined on the $t=0$ circle of the cylinder can be obtained using (100)

$$[L_{\frac{m}{w}}, L_{\frac{n}{w}}] = [L_m^{cov}, L_n^{cov}] = \frac{2\pi(m-n)}{wR} L_m^{cov} + \frac{4\pi^2 c\, m^3}{12R^2w^2}\delta_{m+n}$$
$$= \frac{2\pi}{R} \cdot \frac{m-n}{w} L_{\frac{m+n}{w}} + \frac{4\pi^2}{R^2} \cdot \frac{c\, m^3}{12w^2}\delta_{m+n}, \tag{101}$$

where we have used the fact that in our conventions, the generators are dimensionful, and thus explicit factors of $R$ appear in their algebra ($Rw$ if we are on the covering space). The

---

[15]The standard discussions of the covering map consider the CFT on the plane, and the map is of the form $z_{base} = t_{cov}^w$. The Fourier modes in radial quantization are integrated over a circle of circumference $2\pi$. To relate these to the Fourier modes on the RHS of (100), which are defined on a circle $w$ times larger, one needs to perform a conformal rescaling $\tilde{\sigma} \to \sigma = \tilde{\sigma}/w$. Under it, if $\phi$ is a primary field of weight $h$, then $\phi^{\{wR\}}(\tilde{\sigma}) = w^{-h}\phi^{\{R\}}(\sigma)$, where the superscript indicates the size of the cylinder on which the theory is defined. It follows that $\phi_{m/w} = \phi_m^{\{wR\}} = w^{1-h}\phi_m^{\{R\}}$ where for $R = 2\pi$, $\phi_m^{\{R\}}$ are the standard Fourier modes of the field; this reproduces the expressions in the literature. We emphasize the fact that the conformal transformation is not needed to relate the fractional modes of the twisted field to those of the seed on the covering cylinder, but only in resizing that covering cylinder back to the initial length $R$. For the non-conformal models we are interested in, we will prefer to work directly with the cylinder of size $wR$.

above is known as the fractional Virasoro algebra and, by construction, it is isomorphic to the Virasoro algebra of the seed CFT. This algebra may be brought to a more standard form by rescaling the fractional generators by $R/2\pi$, or simply setting $R = 2\pi$. The reason that the central term simply involves $m^3$ is that we work on the cylinder, where the eigenvalue of $L_0$ is shifted with respect to that on the plane by an amount proportional to the central charge in that sector.

The global Virasoro symmetry generators correspond to choosing $m \in w\mathbb{Z}$ which, as we already explained, are the only modes that survive in the action of the gauge-invariant operator (95) on this sector. The fractional Virasoro modes can be used to build fractional descendants, which may be Virasoro primaries under certain conditions [70]. Relatedly, if in the seed CFT on the covering space, two fields are related via the action of $L_{-n}$, then their images in the twisted sector will be related via $L_{-n/w}$.

A similar discussion holds for the case of other symmetries of the seed CFT, such as an $U(1)$ affine symmetry. The commutation relations of the fractional current modes are found to be

$$[J_{m/w}, J_{n/w}] = \frac{k}{2} m\, \delta_{n+m,0}\,, \quad [L_{m/w}, J_{n/w}] = -\frac{2\pi n}{wR} J_{\frac{m+n}{w}}\,, \tag{102}$$

where, as before, $k$ represents the $U(1)$ level of the seed CFT. Note that since the mode number is $m/w$, the level of this algebra, as it appears in the position space OPE, is $k^{(w)} = kw$, in agreement with the fact that $w$ copies of the seed CFT are involved in the computation. More generally, for a primary operator $\mathcal{O}$ from the seed CFT with left conformal dimension $h^{(s)}$ and $U(1)$ charge $q^{(s)}$, we find that the seed commutation relations on the covering cylinder descend to the following commutation relations with the fractional Virasoro and Kac-Moody generators on the base

$$[L_{m/w}, \mathcal{O}_{n/w}] = \frac{2\pi}{Rw}\left((h^{(s)} - 1)m - n\right)\mathcal{O}_{(m+n)/w}\,, \tag{103}$$

$$[J_{m/w}, \mathcal{O}_{n/w}] = q^{(s)}\mathcal{O}_{(m+n)/w}\,, \tag{104}$$

where we have used the relation (100) between the fractional modes on the base and Fourier modes on the covering cylinder. These are nothing but the momentum-space commutation relations on the cylinder, particularized to fractional momenta.

Note that the conformal dimension of the operator is $h^{(s)}$, corresponding to a standard untwisted-sector operator acting on a sector with twisted boundary conditions; in particular, its short-distance behaviour is governed by $h^{(s)}$. Just as for the energy-momentum modes, the twisted boundary conditions can be thought of as generated by the insertion of a $w$-twist operator at $\tau = -\infty$, which allows the modes of any operator acting on the cylinder to be fractional.

We would like to draw a distinction - which will become important in section 4.2 - between these operators, which act in the presence of a twist inserted at a different location, and genuine twisted operators, denoted $\mathcal{O}^{(w)}$, which contain a twist at their own location. To obtain the Ward identities of the latter with the Virasoro generators one may simply start on the plane, and then lift the result to the covering space using the standard map $z = z_0 + (t - t_0)^w$, where $z_0$ is the insertion point of the operator. Following the steps in [70] and suppressing, for simplicity, the right-moving labels, we find

$$[L_n, \mathcal{O}^{(w)}(z_0)] = \oint_{t_0} \frac{dt}{2\pi i} \frac{dz}{dt} z^{n+1} \frac{1}{z'(t)^2}\left(T(t) - \frac{c}{12}\{z, t\}\right)\mathcal{O}(t_0)\,. \tag{105}$$

Using the OPE on the covering space and the fact that the Schwarzian derivative is $\{z, t\} = \frac{1-w^2}{2(t-t_0)^2}$,

the above reduces to

$$[L_n, \mathcal{O}^{(w)}(z_0)] = \oint_{t_0} \frac{dt}{2\pi i} (z_0 + (t-t_0)^w)^{n+1} \left[ \frac{h^{(w)}\mathcal{O}(t_0)}{(t-t_0)^{w+1}} + \frac{\partial_z\mathcal{O}(t_0)}{t-t_0} \right], \qquad (106)$$

where $h^{(w)}$ is given in (47). Integrating and redescending the result to the base space, we obtain

$$[L_n, \mathcal{O}^{(w)}(z)] = h^{(w)}(n+1)z^n\mathcal{O}^{(w)}(z) + z^{n+1}\partial_z\mathcal{O}^{(w)}(z), \qquad (107)$$

where we have dropped the index '0' from the base space coordinate. In contrast with our previous computation, the conformal dimension which appears in the Ward identity is now $h^{(w)}$, due to the presence of a twist at the location of $\mathcal{O}^{(w)}$. The covering space considered is also different from the larger cylinder previously used and, in particular, in this case the Schwarzian derivative does yield a non-trivial contribution. One may then use the plane to cylinder map on the base, $z = e^{2\pi\zeta/R}$ with $\zeta = \tau + i\sigma$, to obtain the standard cylinder Ward identities

$$[L_n, \mathcal{O}^{(w)}(\zeta,\bar{\zeta})] = e^{\frac{2\pi n\zeta}{R}} \left( \frac{2\pi n h^{(w)}}{R} \mathcal{O}^{(w)}(\zeta,\bar{\zeta}) + \partial_\zeta\mathcal{O}^{(w)}(\zeta,\bar{\zeta}) \right), \qquad (108)$$

$$[J_n, \mathcal{O}^{(w)}(\zeta,\bar{\zeta})] = q^{(s)} e^{\frac{2\pi n\zeta}{R}} \mathcal{O}^{(w)}(\zeta,\bar{\zeta}). \qquad (109)$$

These commutation relations correspond to the cylinder Ward identities for a (properly periodic) operator of conformal dimension $h^{(w)}$ and charge $q^{(w)} = q^{(s)}$. Note that we could have also considered the commutation relations of this twisted operator with the fractional Virasoro modes, which would have simply amounted to replacing $n \to n/w$ - or, equivalently, $R \to Rw$ - in he expressions above. However, since the operator creating the twist is located at $(\zeta,\bar{\zeta})$, and not at $-\infty$, these Ward identities would have only held provided no other operator was inserted between these two points, which is a very restrictve requirement. On the other hand, the commutation relations with the globally-defined Virasoro generators are entirely general. The right-moving commutation relations take an analogous form.

## 3.3 Flow of the states in single-trace $T\bar{T}$ and $J\bar{T}$-deformed CFTs

To uncover the Virasoro and Kac-Moody symmetries of single-trace $T\bar{T}$ and $J\bar{T}$ - deformed CFTs, we follow exactly the same strategy as in the double-trace case. For this, we first need to understand the structure of the operator that drives the flow of the energy eigenstates via the single-trace analogue of (78), where now $|n_\lambda\rangle$ represent the energy eigenstates of the $T\bar{T}/J\bar{T}$ symmetric product orbifold. For simplicity, we restrict our discussion to the single-trace $T\bar{T}$ deformation; a virtually identical analysis holds in the $J\bar{T}$ case.

To determine the flow operator, it is sufficient to consider first order degenerate quantum-mechanical perturbation theory about the instantaneous energy eigenstates $|n_\lambda\rangle$. Ultimately, we will show that the flow operator in single-trace $T\bar{T}$ - deformed CFTs simply corresponds to the standard $T\bar{T}$ flow operator on the covering space, namely on the cylinder of circumference $Rw$. However, to arrive at this result, we first need to understand the degeneracies that are broken by the deformation, as these play a role in determining the flow operator. To make the discussion self-contained, we start by reminding the reader a few facts about degenerate perturbation theory, following e.g. [71].

**Brief review of degenerate perturbation theory**

Assume we have a quantum-mechanical system where a subspace - denoted $\mathcal{H}_n$ - of the Hilbert space is degenerate. In this subspace, the undeformed energy levels will be denoted as $|n^{(0)}, k\rangle$,

where $k \in \{1, \ldots, dim_{\mathcal{H}_n}\}$, and their energy as $E_n^{(0)}$. The energy levels in the orthogonal part of the Hilbert space will be denoted as $|p^{(0)}\rangle$. The $\mathcal{O}(\lambda)$ change in the energy eigenstates and eigenvalues under a perturbation $\delta H = \lambda H^{(1)} + \lambda^2 H^{(2)} + \ldots$ is given by

$$|n,k\rangle = |n^{(0)}, k\rangle + \lambda \left( \sum_{p \neq n} |p^{(0)}\rangle \frac{\langle p^{(0)}|H^{(1)}|n^{(0)}, k\rangle}{E_n^{(0)} - E_p^{(0)}} + \sum_{l \neq k} a_{k,l} |n^{(0)}, l\rangle \right) + \mathcal{O}(\lambda^2), \qquad (110)$$

$$E_{n,k}(\lambda) = E_n^{(0)} + \lambda \langle n^{(0)}, k|H^{(1)}|n^{(0)}, k\rangle + \mathcal{O}(\lambda^2). \qquad (111)$$

We note that the first order correction, denoted $|n^{(1)}, k\rangle$, to the state corresponds simply to the action of the flow operator $\mathcal{X}$ defined as in (78) on $|n^{(0)}, k\rangle$ at leading order in perturbation theory. We can then easily show that, at this order

$$[H, \mathcal{X}]|n^{(0)}, k\rangle = H|n^{(1)}, k\rangle - E_n^{(0)}|n^{(1)}, k\rangle = -\sum_{p \neq n} |p^{(0)}\rangle \langle p^{(0)}|H^{(1)}|n^{(0)}, k\rangle. \qquad (112)$$

Since the state $|n^{(0)}, k\rangle$ is arbitrary, we conclude that, to this order

$$[H, \mathcal{X}] = -(H^{(1)} - \text{diag}), \qquad (113)$$

where 'diag' represents the diagonal matrix elements of the deforming operator on the union, denoted $\mathcal{H}_D$, of all the degenerate subspaces. Thus, the commutator of $\mathcal{X}$ with $H$ is determined by the fully off-diagonal pieces of the perturbation, whether the degeneracy is lifted or not.

The basis $|n^{(0)}, k\rangle$ and the coefficients $a_{k,l}$ depend on whether the perturbation breaks or not the degeneracy of the undeformed theory. If it does, then the basis $|n^{(0)}, k\rangle$ must be *chosen* so that $\delta H$ is diagonal, and the $a_{k,l}$ - which correspond to the matrix elements of $\mathcal{X}$ that lie inside the initially degenerate subspace - are determined by the $\mathcal{O}(\lambda^2)$ analysis to be

$$a_{k,l} = \frac{1}{E_{n,k}^{(1)} - E_{n,l}^{(1)}} \left( \sum_p \frac{H_{nl,p}^{(1)} H_{p,nk}^{(1)}}{E_n^{(0)} - E_p^{(0)}} + \langle n^{(0)}, l|H^{(2)}|n^{(0)}, k\rangle \right), \qquad (114)$$

where $H^{(2)}$ is an eventual $\mathcal{O}(\lambda^2)$ correction to the Hamiltonian (not considered in the analysis of [71]) and $H_{p,nk}^{(1)} \equiv \langle p^{(0)}|H^{(1)}|n^{(0)}, k\rangle$, $H_{nk,p}^{(1)} = (H_{p,nk}^{(1)})^\star$ and we have assumed, for simplicity, that the breaking of the degeneracy is complete. If, on the other hand, the perturbation does not break the degeneracy at $\mathcal{O}(\lambda)$, but rather at some higher order $b$, then the coefficients $a_{k,l}$ are determined by the analysis at $\mathcal{O}(\lambda^{b+1})$. If the degeneracy is never broken, then one is free to choose any basis on the degenerate subspace. Note also that the matrix elements of the perturbing operator $\delta H$ are diagonal on $\mathcal{H}_D$, either because the basis had to be chosen so that this holds, or because $\delta H$ is proportional to the identity on this subspace, as is the case when the degeneracies are not broken.

**Degeneracies of single-trace $T\bar{T}$ - deformed CFTs**

The main lesson of the discussion above is that, whenever a deformation breaks an existing degeneracy, the elements of the flow operator in the broken subspace of the Hilbert space are fixed. Thus, in order to understand the structure of the single-trace $T\bar{T}$ flow operator, we need to check whether the deformation breaks - or not - the existing degeneracies. This can be determined by the exact knowledge of the spectrum of the deformed theory. For example, in the case of the double-trace $T\bar{T}$ deformation, the fact that the deformed energies are only functions of the undeformed ones implies that any degeneracy initially present in the CFT will not be lifted by the deformation. Consequently, the elements of $\mathcal{X}_{T\bar{T}}$ on the degenerate

subspace of the corresponding double-trace $T\bar{T}$ - deformed CFT are not fixed. We can in particular choose, as in [39], the same expression for $\mathcal{X}_{T\bar{T}}$ that is given by the assumption of non-degenerate eigenstates; this amounts to a particular way to continue the arbitrarily-chosen basis of degenerate eigenstates from the undeformed CFT to the deformed one.

The case of the single-trace $T\bar{T}$ deformation is different from the double-trace one in that - as remarked in [30] - the degeneracy is partially lifted when we turn on the deformation. This breaking of the degeneracies can be easily seen from the energy formula (54)

$$E = \sum_{cycles} E^{(w_i)}\left(\mu, R, E_{CFT}^{(w_i)}, P_{CFT}^{(w_i)}\right), \tag{115}$$

where $E_{CFT}^{(w_i)}$ represents the undeformed CFT energy in that sector.

Let us first discuss the untwisted sector, where the the undeformed energies are of the schematic form $\sum_{I=1}^{N}\Delta_I + n_I$, with $\Delta_I$ the primary operator dimensions (we omit writing the shift by $c/12$, and set the radius to one) and $n_I$ the total level of the descendants. One type of degeneracies are those within a single copy (fixed $n_I$), which are the same as in the double-trace case, and thus are not lifted. Another type of degeneracies are those among different copies (fixed $\sum_I n_I$) of the seed CFT. These degeneracies are generically broken when the deformation is first turned on, as the first order correction to the energy is $\mu\sum_I(\Delta_I + n_I)^2$. For generic operator dimensions of the seed CFT, the $\sum\Delta_I n_I$ term completely breaks any initial degeneracy; if some of the $\Delta_I$ happen to coincide, then the $\sum n_I^2$ term will break the degeneracy. A similar discussion holds for the twisted sectors, where the energy at the CFT point is given by $E = \sum_{i \in cycles}(\Delta_i + n_i)/w_i$. Degeneracies within a single cycle will not be broken at any order in perturbation theory, because the flow equation (58) within a single cycle is nothing but the standard $T\bar{T}$ flow equation with an effective parameter $\mu/w_i$. However, the degeneracies corresponding to different ways of distributing the energy among the different cycles, with $\sum_i n_i/w_i$ fixed, will be broken once we turn on the single-trace $T\bar{T}$ perturbation. We will ignore the possibility of level crossing at finite $\mu$.

Thus, the degeneracies are only broken when the deformation is first turned on. Denoting the degnerate subspace at $\mu = 0$ by $\mathcal{H}_{D^{(0)}}$, and the smaller degenerate subspace at $\mu \neq 0$ by $\mathcal{H}_D$, then the matrix elements of the single-trace flow operator on $\mathcal{H}_{D^{(0)}} \setminus \mathcal{H}_D$ are fixed and are given by (114) ; the matrix elements on $\mathcal{H}_D$ can be chosen at will, since further tuning $\mu$ away from zero is not expected to break any additional degeneracies. For this statement to be true, we assume that the spectrum of the seed CFT is generic, i.e. the primary dimensions are arbitrary real numbers subject to the unitarity and all other consistency constraints.

Note that the above result should be consistent with that obtained from *instantaneous* perturbation theory $\lambda \to \lambda + \delta\lambda$, in the limit $\lambda \to 0$. The expression for the instantaneous correction to the state vector is given by (110), with $|n^{(0)}\rangle$ and $E_n^{(0)}$ now representing the state and energy at a given finite $\lambda$, and the sum in the first term running over the states outside $\mathcal{H}_D$. As $\lambda \to 0$, the energy difference between some of these states (namely, those who have inter-cycle degeneracies at $\lambda = 0$) becomes $\mathcal{O}(\lambda)$, which implies that the numerator of that expression will receive contributions from different orders in the $\lambda$ expansion. The structure of these terms is similar to the expected contribution (114), and we expect that the matrix elements that get fixed on $\mathcal{H}_{D^{(0)}} \setminus \mathcal{H}_D$ precisely coincide with those predicted by general non-degenerate perturbation theory outside $\mathcal{H}_D$, and we need not worry about them. The elements inside $\mathcal{H}_D$ are not fixed, and can be chosen conveniently.

The case of the single-trace $J\bar{T}$ deformation can be treated in an exactly analogous manner. Since the first order correction to the energy is $\lambda\sum_I q_I(\bar{h}_I + n_I)$, with $\bar{h}_I$ the undeformed right-moving dimensions, it follows that the degeneracy among the different cycles will generically be broken as soon as the deformation is turned on. The rest of our conclusions follow straightforwardly.

**The single-trace $T\bar{T}$ flow operator**

Let us now apply this to the single-trace $T\bar{T}$ deformation. Using (113), the matrix elements of the single-trace flow operator, schematically denoted as $\mathcal{X}_{T_I\bar{T}_I}$, satisfy

$$[H, \mathcal{X}_{T_I\bar{T}_I}] = \sum_I \int d\sigma\, \mathcal{O}_{T_I\bar{T}_I} - \text{diag}, \tag{116}$$

where 'diag' now refers to the diagonal matrix elements of the integrated single-trace $T\bar{T}$ operator on $\mathcal{H}_D$. As discussed in the previous subsection, even though the single-trace $T\bar{T}$ operator itself is a sum over copies, its integrated version is not, except in the untwisted sector. Since then also $H = \sum_I H_I$, it is natural that

$$\mathcal{X}_{T_I\bar{T}_I} = \sum_I \mathcal{X}^I_{T\bar{T}} \qquad \text{in the untwisted sector, outside } \mathcal{H}_D, \tag{117}$$

where $\mathcal{X}^I_{T\bar{T}}$ is the flow operator associated with a single copy of the $T\bar{T}$ - deformed CFT, which is worked explicitly in [39], at least at the classical level. This relation only need to hold outside $\mathcal{H}_D$, which consists of only in-cycle degeneracies; however, as discussed, the elements of $\mathcal{X}_{T_I\bar{T}_I}$ can be chosen at will inside it.

The fact that the flow operator takes this form in the untwisted sector can also be seen very explicitly by studying the flow equation for states in this sector, which are built as sums over all possible permutations $|n_\lambda^{\sigma(1)}\rangle \otimes \ldots |n_\lambda^{\sigma(N)}\rangle$ of $N$ fixed energy eigenstates, $|n_\lambda^I\rangle$, in the seed theory. Plugging this into the right-hand-side of (110) and using the fact that $\delta H$ consists of a sum over copies in this sector, we find that the instantaneous energy eigenstates $|m_\lambda\rangle$ (corresponding to $|p^{(0)}\rangle$ in (110)), which in this sector take the form $|m_\lambda^1\rangle \otimes \ldots |m_\lambda^N\rangle$, fully symmetrized, should "click" with $|n_\lambda\rangle$ in all its entries but one, where the particular copy of the $T\bar{T}$ operator acts. More precisely

$$\partial_\lambda |n_\lambda\rangle = \sum_{m\neq n} |m_\lambda\rangle \frac{\langle m_\lambda| \int \mathcal{O}_{T_I\bar{T}_I}|n_\lambda\rangle}{E_n^\lambda - E_m^\lambda} = \sum_{perm}\sum_I |n_\lambda^1\rangle \otimes \ldots \sum_{m^I\neq n^I} |m_\lambda^I\rangle \frac{\langle m_\lambda^I| \int \mathcal{O}_{T_I\bar{T}_I}|n_\lambda^I\rangle}{E_n^{I,\lambda} - E_m^{I,\lambda}} \otimes \ldots |n_\lambda^N\rangle, \tag{118}$$

where in the second step we used the fact that the energies in the untwisted sector are just sums of the energies of the individual copies. Both sums run only over non-degenerate eigenstates; the sum over permutations should be properly normalised. We have also set all ambiguities in the state due to (intra-cycle) degeneracies to zero; including them would shift the individual copy contributions, without affecting the general structure. Near $\lambda = 0$, one could also check that the fixed matrix elements of $\mathcal{X}_{T_I\bar{T}_I}$ do not break the symmetric product orbifold structure, as would be implied by having $a_{k,l} \neq 0$ between different cycles; however, this is ensured by the fact that the $\lambda \to 0$ case can be embedded in the generic $\lambda \neq 0$ one, which does respect the symmetric orbifold structure. Since the terms in the intermediate sums may be written as $\partial_\lambda |n_\lambda^I\rangle = \mathcal{X}^I_{T\bar{T}}|n_\lambda^I\rangle$, we find that

$$\partial_\lambda |n_\lambda\rangle = \sum_{perm}\sum_I |n_\lambda^1\rangle \otimes \ldots \partial_\lambda |n_\lambda^I\rangle \otimes \ldots |n_\lambda^N\rangle, \tag{119}$$

and so the deformed state is just the tensor product of $T\bar{T}$ - deformed states - thus confirming the fact that single-trace $T\bar{T}$ preserves the symmetric orbifold structure[16] - and that the flow

---

[16]By contrast, the action of the double-trace $T\bar{T}$ deformation on the symmetric orbifold is given by

$$\partial_\lambda |n_\lambda\rangle = \sum_{perm}\sum_{I,J} |n_\lambda^1\rangle \otimes \ldots \sum_{m^I\neq n^I}|m_\lambda^I\rangle \frac{\langle m_\lambda^I|T_I|n_\lambda^I\rangle}{E_m^I - E_n^I} \otimes \ldots \otimes \sum_{m^J\neq n^J}|m_\lambda^J\rangle \frac{\langle m_\lambda^J|\bar{T}_J|n_\lambda^J\rangle}{E_m^J - E_n^J} \otimes \ldots |n_\lambda^N\rangle, \tag{120}$$

which implies that $\mathcal{X}_{T\bar{T}\,on\,SPO}$ has a bilocal structure, which does not respect the symmetric product form of the state.

operator in this sector takes the form (117).

In the twisted sector, it is no longer true that the deforming operator is a sum over copies; nevertheless, an explicit expression can still be easily obtained for $\mathcal{X}_{T_I \bar{T}_I}$, at least at the (semi)classical level, by following the steps of [39]. More precisely, we have

$$\int d\sigma\, \mathcal{O}_{T_I \bar{T}_I} = \left[ H, i \int d\sigma d\tilde{\sigma}\, G(\sigma - \tilde{\sigma}) \mathcal{H}^I(\sigma) \mathcal{P}^I(\tilde{\sigma}) \right] + \frac{1}{R} \int d\sigma d\tilde{\sigma} \left( \mathcal{H}^I(\sigma) T^I_{\sigma\sigma}(\tilde{\sigma}) - \mathcal{P}^I(\sigma) \mathcal{P}^I(\tilde{\sigma}) \right). \tag{121}$$

Upon summing over the various copies, the left-hand-side of this equation, minus its diagonal piece, corresponds precisely to $[H, \mathcal{X}_{T_I \bar{T}_I}]$. To obtain a closed-form expression for it, one may use the $T\bar{T}$ trace relation in each copy

$$T^I_{\sigma\sigma}(\sigma) = \mathcal{H}^I(\sigma) - 2\mu \mathcal{O}_{T_I \bar{T}_I}(\sigma), \tag{122}$$

which holds classically as is, and quantum-mechanically up to a total derivative. Plugging this into (121) and acting with the whole equation on a twisted sector, in which the expansion of the fields takes the form (99) (we are ignoring for now the copies that do not participate in the twist), the integral over $\mathcal{H}^I(\sigma)$ that multiplies $\mathcal{O}_{T_I \bar{T}_I}$ in the second term yields a factor of $H^{(w)}/w$, where $H^{(w)}$ is the Hamiltonian restricted to the $w$-twisted sector and the index '$w$' is now a shorthand for the individual $w$ copies that enter the cycle. Ultimately we find, at the classical level

$$\mathcal{X}^{(w),cls}_{T_I \bar{T}_I} = \frac{1}{1 + 2\mu H^{(w)}/(Rw)} \int d\sigma d\tilde{\sigma}\, G(\sigma - \tilde{\sigma}) \mathcal{H}^I(\sigma) \mathcal{P}^I(\tilde{\sigma}). \tag{123}$$

This expression corresponds to nothing but the seed $\mathcal{X}_{T\bar{T}}$ on the covering space. One way of showing this is by expanding $\mathcal{H}^I(\sigma)$ above as a function of the undeformed CFT generators $\mathcal{H}^{(0)}_I(\sigma)$ and $\mathcal{P}^I(\sigma)$, using the closed-form expression given in [39]. When acting upon a twisted-sector state, which we choose again to be a single cycle of length $w$, each of the fields can be expanded in a sum over fractional modes as in (99). We then note that the integral over the sum of $\sigma$, $\tilde{\sigma}$ ensures that the $I$ dependence will drop out from the integral, leading to an overall $w$ factor. The dependence on the Fourier modes of $\mathcal{H}^{(0)}$ and $\mathcal{P}$ will be the same as in the double-trace expression, but with $m \to m/w$. Since each factor of $\mu$ in the expansion is accompanied by a factor of $\mathcal{H}^{(0)}_I$ or $\mathcal{P}^I$, the overall factor of $w$ that multiplies the $\mu^p$ term in the expansion is $w^{-p} \times w \times w$, where the last factor comes from the Fourier transform of the Green's function. The Fourier integrals also bring a factor of $R^{-(p-2)}$, by dimensional analysis. We therefore notice that the $w$ and $R$ dependence is such that it combines into precisely a dependence on $Rw$, which coincides, upon lifting to the covering space, with the flow operator of the double-trace $T\bar{T}$ - deformed CFT. Including quantum corrections amounts to replacing factors of $L_{m/w}$ in the expansion by factors of $n/wR$ that would result from commuting two such fractional Virasoro modes, in perfect agreement with the general counting. Thus, in the $w$-twisted sector, $\mathcal{X}_{T_I \bar{T}_I}$ acts just as the descent of $\mathcal{X}_{T\bar{T}}$ on the covering space to the base cylinder.

More generally, if there are several twisted and untwisted sectors present - as determined by the conjugacy class, $[g]$, of the permutation group - then (121) reduces to a sum over sectors which, using the orthogonality of the various subsectors of the Hilbert space, leads to the following expression for the total flow operator $\mathcal{X}_{T_I \bar{T}_I}$ when acting on $\mathcal{H}^{[g]}$

$$\mathcal{X}^{[g]}_{T_I \bar{T}_I} = \sum_{\sigma \in S_N} \mathcal{X}^{(\sigma(1)...\sigma(w_1))}_{T_I \bar{T}_I} + \mathcal{X}^{(\sigma(w_1+1)...\sigma(w_1+w_2))}_{T_I \bar{T}_I} + \ldots \tag{124}$$

Here $w_i$ are the lengths of the cycles that appear in $[g]$, each contributing with a flow operator constructed along the lines of the previous paragraph, which lifts to the flow operator of the

seed $T\bar{T}$ - deformed CFT on the cylinder of circumference $Rw_i$. The sum over permutations considers all combinations in which the copies enter the cycles of $[g]$, making the full operator well-defined on the Hilbert subspace $\mathcal{H}^{[g]}$. Note that when all cycles have length one, this reduces to the expression (117) for $\mathcal{X}$ in the untwisted sector.

Being obtained from the commutator (116), the above expression for $\mathcal{X}_{T_I\bar{T}_I}$ holds except possibly on the diagonal degenerate subspaces. However, as we have just argued, we shouldn't expect the matrix elements on $\mathcal{H}_D$ to be fixed; our expression (124) corresponds to a particular choice.

## 3.4 Symmetries of single-trace $T\bar{T}$ and $J\bar{T}$-deformed CFTs

Having discussed the flow operator, we would now like to study the extended symmetries of single-trace $T\bar{T}$ and $J\bar{T}$ - deformed CFTs, following the steps of the double-trace analysis. We therefore define a set of Virasoro generators as[17]

$$\partial_\lambda \widetilde{L}_m^\lambda = [\mathcal{X}_{T_I\bar{T}_I}, \widetilde{L}_m^\lambda], \quad \widetilde{L}_m^{\lambda=0} = L_m^{CFT}. \tag{125}$$

**Untwisted sector analysis**

This equation is very easy to solve in the untwisted sector: since $\mathcal{X}_{T_I\bar{T}_I}$ is a single-trace operator and so is the initial $L_m^{CFT}$, it follows that the flowed $\widetilde{L}_m^\lambda$ will simply be

$$\widetilde{L}_m^\lambda = \sum_I \widetilde{L}_m^{I,\lambda}, \tag{126}$$

where the $\widetilde{L}_m^{I,\lambda}$ represent the solution to the flow equation in a single copy of a $T\bar{T}$ / $J\bar{T}$ - deformed CFT. The same definition can be extended to all the other Virasoro and Kac-Moody generators of the seed symmetric product orbifold. Their algebra consists of two-commuting copies of the Virasoro ($\ltimes$ Kac-Moody) algebra by construction. As explained in the introductory subsection, the non-trivial check that these generators represent symmetries of the theory is to prove that they are conserved. For this, we compute their commutator with the Hamiltonian

$$[H, \widetilde{L}_m^\lambda] = \left[\sum_I H_I, \sum_J \widetilde{L}_m^{J,\lambda}\right] = \sum_I \alpha_m^I \widetilde{L}_m^{I,\lambda}, \quad \alpha_m^I \equiv \alpha_m(H_I, P_I), \tag{127}$$

where the $\alpha_m$ for the cases of interest are given in (83). This immediately implies that $\sum_I e^{i\alpha_m^I t} \widetilde{L}_m^{I,\lambda}$ satisfies the conservation equation (81). While one may worry that different operators $\alpha_m^I$ appear in the time-dependent factors above, their expectation value in any state of the symmetric orbifold is the same, as it cannot depend on the particular copy.

**Twisted sector analysis**

In the twisted sectors on the cylinder, $\widetilde{L}_m^\lambda$ and $\mathcal{X}_{T_I\bar{T}_I}$ no longer take the form of a sum over copies. As explained in the introductory subsection, in this sector we will generically obtain fractionally-moded generators, of which the integer-moded generators are a particular case. We will therefore discuss the preservation of the fractional Virasoro and Kac-Moody symmetries generically. We again define them via the flow equation

$$\partial_\lambda \widetilde{L}_{m/w}^\lambda = [\mathcal{X}_{T_I\bar{T}_I}, \widetilde{L}_{m/w}^\lambda], \quad \widetilde{L}_{m/w}^{\lambda=0} = \widetilde{L}_{m/w}^{CFT}. \tag{128}$$

---

[17]Even though we use exactly the same notation as in the double-trace case, it should be clear from the context that these are the Virasoro generators in single-trace $T\bar{T}/J\bar{T}$ deformed CFTs.

This is a perfectly well-defined equation in the Hilbert space of the $T\bar{T}/J\bar{T}$ symmetric orbifold. Since $\mathcal{X}_{T_I\bar{T}_I}$ belongs to the untwisted sector, it infinitesimally takes a $w$-twisted sector operator to another one. The standard Virasoro generators are obtained when $m$ is a multiple of $w$.

We would now like to show that the solution to this flow equation corresponds precisely to the $T\bar{T}$ solution for the $\widetilde{L}_m^\lambda$ on the covering space - which is a cylinder of size $Rw$ - reinterpreted on the base via (100). As discussed at the beginning of this section, this map makes no use of conformal invariance, but only sews together the different copies. For simplicity, we assume again that the twist acts on the first $w$ copies of the seed, and as identity on the remaining $N-w$ ones. A summation over $S_N$ will render the final result gauge-invariant.

As we have already discussed, in the $w$-twisted sector, the flow operator on the base simply corresponds to the flow operator $\mathcal{X}_{T\bar{T}}$ of the seed theory on the covering cylinder. Also, in the undeformed CFT, the fractional Virasoro modes correspond to integer modes on the covering. It follows that the solution to (128) will be the same as the solution to (80) for the seed theory, but on a cylinder of radius $Rw$. Since we particularized our discussion to a given number of copies on which the single-cycle twist acts, only one of the $\mathcal{X}^{(w)}$ factors in (124) will act non-trivially on the generator, but the final expressions will be symmetrized with respect to $S_N$. For example, in the trivial case $w=1$, we obtain $\widetilde{L}_m^\lambda$ in the seed for the first copy and the symmetrization will yield the untwisted sector result (126). The same procedure can be applied to the right-moving Virasoro generators, as well as to the Kac-Moody currents.

In order for the solution to (128) to define a set of conserved operators, we first need to compute its commutator with the Hamiltonian. This may be simply evaluated on the cover, where it yields $\alpha_m(H^{cov},...,wR)\widetilde{L}_m^{cov}$, where the dots stand for the other conserved quantities that enter the definition (83) of $\alpha_m$ (i.e., momentum for $T\bar{T}$, and also $U(1)$ charge for $J\bar{T}$) and we have dropped the index '$\lambda$' from the generator, to lighten the notation. Translating this to the base cylinder, we find $\alpha_m(H^{(w)},...,wR)\widetilde{L}_{m/w}$. One may also derive these expressions from the non-linear relation (56) between the undeformed and deformed spectrum in the twisted sector, which implies a (non-linear and $w$ - dependent) relation between $H^{(w)}$ and $\widetilde{L}_0^{(w)}$. That this relation depends on the particular sector is not surprising, given that the definition of $\widetilde{L}_0$ is sector-dependent.

Adding the appropriate time-dependence to ensure conservation, and taking into account all the possible choices of copies that can enter the cycle, the full answer for the fractional Virasoro generators then takes the form

$$\widetilde{L}_{m/w}(t) = \sum_{\sigma \in S_N} e^{i\alpha_m(H^{(\sigma(1)...\sigma(w))},...,wR)t} \widetilde{L}_{m/w}^{(\sigma(1)...\sigma(w))}, \tag{129}$$

where the superscripts indicate the particular copies entering the non-trivial cycle. Note this reduces to the correct answer in the untwisted sector ($w=1$). The same holds for the right-moving Virasoro and Kac-Moody generators. The algebra of the fractional Virasoro generators above is the same as in the undeformed CFT, as follows from the definition (128). In particular, for $m$ a multiple of $w$ we obtain the integer Virasoro modes, which are present in any sector of the theory.

Note that in principle, these modes could also be obtained by flowing $\mathcal{H}_{L,R}(\sigma)$ and then performing the Fourier transform, as in [39]. Even though the operators we start from are single-trace operators, $\sum_I \mathcal{H}_{L,R}^I(\sigma)$, the solution to the flow equation looks differently in the different sectors because $\mathcal{X}_{T_I\bar{T}_I}$ does. The result of the flow equation will lead to a notion of emergent field-dependent coordinates, which will also be defined in the twisted sectors; from the structure of the flow we see they will correspond precisely to the field-dependent coordinates $u, v$ on the covering space.

To conclude this section, we have explicitly shown that single-trace $T\bar{T}$ and $J\bar{T}$ - deformed CFTs contain operators that are conserved and satisfy two commuting copies of the Virasoro ($\ltimes$

Kac-Moody) algebra, showing they possess the corresponding symmetry. The central charge and $U(1)$ level of the algebra are simply $N$ times those of the seed theories. In twisted sectors, one finds also fractional Virasoro ($\ltimes$ Kac-Moody) conserved modes.

**Other bases of generators**

As discussed in section 3.1, the flowed Virasoro generators may not provide the most natural basis of generators of the extended symmetries of these theories. In single-trace $J\bar{T}$ - deformed CFTs, the most natural basis of left-moving generators is given by the standard generators of conformal and affine $U(1)_L$ symmetries. In the untwisted sector, they take the form

$$L_m = \sum_I L_m^I = \sum_I \widetilde{L}_m^I + \frac{\lambda}{R} H_R^I \tilde{J}_m^I + \frac{\lambda^2 k}{8\pi R} H_{R,I}^2 \, \delta_{m,0} \,, \qquad J_m = \sum_I J_m^I = \sum_I \widetilde{J}_m^I + \frac{\lambda k}{4\pi} H_R^I \delta_{m,0} \,, \quad (130)$$

and similarly for the right-moving generators, where we are now working in the convention in which the $L_m$ are dimensionful. In the $w$-twisted sector, one can write similar relations by considering the seed on the cylinder of circumference $Rw$, as instructed by the solution of the flow equations

$$L_{m/w} = \widetilde{L}_{m/w} + \frac{\lambda}{Rw} H_R^{(w)} \tilde{J}_{m/w} + \frac{\lambda^2 k}{8\pi Rw} H_R^{(w)2} \delta_{m,0} \,, \qquad J_{m/w} = \widetilde{J}_{m/w} + \frac{\lambda k}{4\pi} H_R^{(w)} \delta_{m,0} \,, \quad (131)$$

where $H_R^{(w)}$ is the (globally-defined) right-moving Hamiltonian, restricted to the $w$ - twisted sector, with eigenvalues the $w$-twisted sector energies (64). One may easily check that for $m = 0$, these yield the correct expression (64) for the deformed energies in the $w$ - twisted sector, taking into account the fact that the eigenvalue of $\widetilde{L}_0$ is identical to that in the undeformed CFT. Note that, throughout this section, the twisted generators will be built from the first $w$ copies of the seed, and symmetrization is assumed only for the final expressions.

Rewriting the relations above in terms of the effective Kac-Moody level in the $w$-twisted sector $k^{(w)} = wk$, we obtain that the relation between the two sets of generators is given by spectral flow with $\lambda H_R^{(w)}/w$. Since the relation is non-linear, the Poisson algebra spanned by the untilded generators is non-linear in these generators.

Note that for $m$ a multiple of $w$, we obtain the global generators of extended symmetries, which correspond to the Fourier integrals of state (and thus sector) - independent quantities over the base cylinder. This can be established, at least classically, as follows: the solution for the generators $L_m, \bar{L}_m$ is the same as the double-trace solution, uplifted to the covering space. E.g., for the left-movers, we have, using (100)

$$L_m = L_{mw}^{cov} = \int_0^{wR} d\tilde{\sigma} \, e^{2\pi i m \tilde{\sigma}/R} \mathcal{H}_L(\tilde{\sigma}), \qquad (132)$$

with $\mathcal{H}_L(\tilde{\sigma} + wR) = \mathcal{H}_L(\tilde{\sigma})$. This integral can be reduced to the integral on an interval of size $R$ of $\sum_{I=0}^{w-1} \mathcal{H}_L(\tilde{\sigma} + RI)$, which is periodic with period $R$. Thus, the integral above becomes a local integral of the periodic current on the base space. The same argument can also be applied to the right-movers, the only difference being that the integrand is now a non-linear function of the fields. We also expect it to extend to the quantum case; note it implies that $L_m, \bar{L}_m$ (whose most appropriate quantum definition may or may not exactly coincide with (85), (86) with $\lambda \to \lambda/w$) are the physical symmetry generators in the full theory.

If $L_m, J_m$ and their right-moving counterparts are the global symmetry generators in single-trace $J\bar{T}$ - deformed CFTs then, given that the relation between them and $\widetilde{L}_m, \widetilde{J}_m$, etc. is that of spectral flow with parameter $\lambda H_R/w$, it follows that the flowed generators are explicitly sector-dependent. This fact is not suprising, given that the flow operator used to define these

generators also depends explicitly on the sector of the theory. It appears possible that requiring at most implicit sector dependence of the symmetry generators in single-trace $T\bar{T}/J\bar{T}$ - deformed CFTs may provide a criterion for selecting the physical basis of generators in these theories.

In single-trace $T\bar{T}$ - deformed CFTs, one may argue, at least classically, that the natural generators of "unrescaled" field-dependent coordinate transformations in the untwisted sector of the symmetric product orbifold should be

$$Q_m = \sum_I Q_m^I = \sum_I R\widetilde{L}_m^I/R_u^I, \quad \text{with} \quad R_u^I = R + 2\mu H_R^I, \tag{133}$$

where the relevant classical expression for $\widetilde{L}_m$ for the $T\bar{T}$ deformation is given in (89); dividing by the field-dependent radius factors yields an expression for $Q_m$ that is the integral of a quasilocal current. Once the appropriate quantum definition of the "unrescaled" generators of the field-dependent symmetries is understood in the seed theory, the result generalizes trivially in the untwisted sector as above.

The algebra of these generators is a sum over copies of the non-linearly-deformed Virasoro algebras obtained in the double-trace case, given up to $\mathcal{O}(\hbar^2)$ by

$$[Q_m, Q_n] = 2\pi\hbar(m-n)\sum_I \frac{Q_{m+n}^I}{R+2\mu H_R^I} + (m-n)\sum_I \frac{8\pi\hbar\mu^2 H_R^I Q_m^I Q_n^I}{RR_u^I R_H^I} + \frac{\pi^2 c\hbar m^3}{3}\sum_I \frac{1}{(R_u^I)^2}\delta_{m+n}. \tag{134}$$

It is interesting to note that *new* operators appear on the right-hand-side in the single-trace case, rather than a product of operators that were already among the generators, as in the double-trace. As we already discussed, it would be good to understand more deeply whether this basis of operators may be preferred for physical reasons and, if so, what is the significance of this non-linear algebra.

In the $w$-twisted sector, one can introduce new fractional generators by performing the division on the covering space of circumference $Rw$, with the result

$$Q_{m/w} = \frac{Rw\widetilde{L}_{m/w}}{R_u^{(w)}}, \tag{135}$$

with $R_u^{(w)} = Rw + 2\mu H_R^{(w)}$. For $m$ multiple of $w$ we obtain the integer versions on the base. These operators may be interpreted as implementing field-dependent coordinate transformations on the covering space, using a Fourier basis. The algebra of these operators is obtained by replacing $Q^I, H_R^I$ by $Q^{(w)}, H_R^{(w)}$ in the above, $m$ by $m/w$, the sums over copies with sums over cycles. Interestingly, if we take the expectation value of this algebra (restricted to its integer modes) in a high-energy state, the result precisely coincides with that of the asymptotic symmetry group analysis of the asymptotically linear dilaton black hole backgrounds performed in [40]. Note this effectively amounts to replacing $\mu \to \mu/N$ in the double-trace algebra (92), which is precisely what was found by the holographic analysis.

**Higher spin currents**

As discussed in the previous section, $T\bar{T}$ and $J\bar{T}$ - deformed CFTs also preserve the KdV charges associated with integrability. In their single-trace version, it is natural to introduce the single-trace analogues of the $\tilde{I}_s$, which in the untwisted sector are given as a sum over copies of the corresponding expressions (93) - (94) with $L_n^I \to \widetilde{L}_n^I$. Their conservation follows trivially from the conservation of the flowed KdV charges in the seed $T\bar{T}/J\bar{T}$ - deformed CFT, given that

the Hamiltonian is a sum over the Hamiltonians in each copy. In the twisted sectors, their conservation follows from that on the covering space.

The symmetric product orbifold of a two-dimensional CFT possesses, however, many other higher spin conserved currents, associated to its much larger symmetry algebra. As explained in e.g. [41], these higher-spin primary currents only exist thanks to multi-trace contributions to an otherwise non-primary field. One such higher-spin current that has been given as an example therein is, for $R = 2\pi$ and specialising to the untwisted sector

$$(W_4)_n = \sum_{I=1}^N \sum_m L_m^I L_{n-m}^I - \frac{3}{10}(n+2)(n+3) \sum_I L_n^I - \frac{\frac{22}{5c}+1}{N-1} \sum_{m,I\neq J} L_m^I L_{n-m}^J. \tag{136}$$

One may wonder whether this current could be transported along the single-trace $T\bar{T}/J\bar{T}$ flow in a similar manner to how the Virasoro and KdV currents were transported, i.e. by requiring it to be covariantly constant along the flow, which simply amonts to replacing the Virasoro modes by their flowed counterparts. Taking $n = 0$ for simplicity, we would like to show that this simple procedure does not lead to a conserved charge.

For this, we compute the commutator of $(\widetilde{W}_4)_0$ with the Hamiltonian. The terms that correspond to single-trace sums are conserved, following the same steps as we used to show the conservation of the KdV charges. However, the multitrace terms lead to a different result, due to the fact that the commutator $[L_m^I, \alpha_n^J]$ only takes the form (88) if $I = J$, and is zero otherwise. Using this, we find, for $I \neq J$

$$[\widetilde{L}_m^I \widetilde{L}_{-m}^J, H] = \widetilde{L}_m^I \alpha_{-m}^J \widetilde{L}_{-m}^J + \alpha_m^I \widetilde{L}_m^I \widetilde{L}_{-m}^J = (\alpha_m^I + \alpha_{-m}^J)\widetilde{L}_m^I \widetilde{L}_{-m}^J. \tag{137}$$

In a CFT, $\alpha_m^I = m\hbar = -\alpha_{-m}^J$ so this term vanishes; however, in $T\bar{T}$ and $J\bar{T}$ - deformed CFTs, this is no longer the case. To leading order in $\mu$, $\alpha_m^I = m\hbar - 2\mu m\hbar(H^I - P^I)$, implying that the leading term breaking the conservation is proportional to $\mu\hbar \sum_m m(H_R^I - H_R^J)\widetilde{L}_m^I \widetilde{L}_{-m}^J$, a combination that does not appear to vanish. Similar results hold in single-trace $J\bar{T}$ - deformed CFTs, on the right-moving side. Thus, the naïve flowed charge associated to this higher-spin symmetry is not conserved in the deformed theory, thus confirming the idea that the conservation of the flowed Virasoro generators and KdV charges is a special feature of these operators, which does not extend to arbitrary currents in the theory. This analysis points towards the conclusion that the associated higher spin symmetries are broken by the single-trace $T\bar{T}/J\bar{T}$ deformations; one should ascertain though that it is not possible to construct corrections to the charges that would restore their conservation.

# 4 Correlation functions

In this section, we would like to discuss correlation functions of operators in symmetric product orbifolds of $T\bar{T}$ and, especially, $J\bar{T}$ - deformed CFTs. Since these theories are neither conformal, nor local, our focus and methods will naturally be quite different from the standard discussion of correlation functions in symmetric product orbifolds of CFTs [72–75], which is centered around computing correlators of twist operators using non-trivial covering maps, and makes essential use of the conformal transformation properties of the operators in question. We will instead concentrate on momentum-space operators, which are natural to consider in a non-local theory, and our main goal will be to understand how to choose a special basis of these operators and compute their correlation functions in terms of the correlators of the undeformed symmetric orbifold CFT.

Our main focus will be $J\bar{T}$ - deformed CFTs, for whose double-trace version [42] has proposed a concrete basis of "primary operator analogues" and computed their correlation functions exactly. The main goal of this section will be to adapt this prescription to the single-trace

$J\bar{T}$ case and use it to compute correlation functions of both untwisted and twisted-sector operators. It is worth mentioning that, quite recently, [44] has also put forth a special basis of operators in $T\bar{T}$ - deformed CFTs and computed their correlation functions, obtaining similar expressions. While a generalisation of these results to the single-trace case would be both interesting and likely possible using their formalism, we do not address this problem here, mainly because it would require a very different method than the one we use for the $J\bar{T}$ case.

The correlation functions in the symmetric product orbifold of $T\bar{T}$ and $J\bar{T}$ - deformed CFTs can then be compared to the correlation functions evaluated using worldsheet techniques in the holographic setups that have been related to these deformations. More precisely, they are expected to match the correlation functions of vertex operators associated to long strings[18] in these backgrounds, which are well described by a symmetric product orbifold. Such worldsheet correlation functions were recently computed in [25] for the case of the asymptotically linear dilaton background, which is related to the single-trace $T\bar{T}$ deformation; their large-momentum behaviour in the untwisted sector agrees with that found by [44]. One may similarly compute correlation functions of long string vertex operators in warped AdS$_3$ by adapting the results of [20] along the lines of [25], and then compare them with our single-trace $J\bar{T}$ result; we find a slight disagreement on the non-local side that we comment upon.

For completeness, we start this section with a review of the correlation functions in double-trace $T\bar{T}$ and $J\bar{T}$ - deformed CFTs, focusing on the explicit proposal of [42] for the latter case.

## 4.1  Review of correlation functions in $T\bar{T}$ and $J\bar{T}$ - deformed CFTs

In local QFTs, there is a special set of operators whose correlation functions are interesting to study, namely local operators. This set is further specialised in CFTs, where much of the focus is on operators that transform as primaries under conformal transformations, as their correlation functions are highly constrained by conformal invariance.

Since $T\bar{T}$ and $J\bar{T}$ - deformed CFTs are non-local, it is not a priori clear which are the natural operators to consider, if a preferred basis exists at all [76]. Indeed, due to the non-locality of these theories, operators are best defined in momentum space. The computation of their correlation functions using e.g. conformal perturbation theory yields divergences, which need to be subtracted via counterterms. Since in a non-local QFT the structure of the allowed counterterms is in general not known, the finite part of the correlator may itself become ambiguous. The situation is under better control in $J\bar{T}$-deformed CFTs, whose locality and $SL(2,\mathbb{R})$ invariance on the left-moving side single out a set of primary operators under these symmetries, for which at least the left-moving part of the correlation function is fixed [43]; however, these questions remain for the right-moving, non-local piece of the correlator.

Despite these concerns and complications, there has been much recent progress in the computation of correlation functions of both $T\bar{T}$ and $J\bar{T}$ - deformed CFTs. Upon making a judicious choice of the operators whose correlation functions one would like to study, it can be shown that the end result is a simple integral transform of the correlation functions of the original CFT. For the case of two-and-three point functions, this integral simply yields the CFT momentum-space correlator, with the conformal dimensions replaced by certain momentum-dependent combinations. Thus, the deformed correlators, though explicitly non-local, are directly and universally determined by the correlation functions in the undeformed CFT, indicating that both $T\bar{T}$ and $J\bar{T}$ - deformed CFTs have a structure that is as rigid as that of two-dimensional CFTs.

---

[18]By contrast, the short string sector is not described by a symmetric product orbifold, and neither their spectrum, nor their correlation functions [21–23] match those in $T\bar{T}/J\bar{T}$ - deformed CFTs.

## $T\bar{T}$ - deformed CFTs

The effect of the $T\bar{T}$ perturbation on correlation functions can be studied at small coupling using conformal perturbation theory, where one can easily note that it induces a momentum-dependent correction to the conformal dimension [77]. The first all-orders analysis of the correlation functions of $T\bar{T}$-deformed CFTs has been performed by [45], who also studied their leading UV divergences and showed they can be absorbed into a non-local renormalization of the operators. More recently, [44] approached this problem using the path integral formulation of the $T\bar{T}$ deformation in terms of coupling the undeformed CFT to JT gravity [78,79]. As it is most clear from its vielbein formulation, this description relates the $T\bar{T}$ - deformed dynamics to that of the original CFT, but seen through a set of "dynamical coordinates" that are related to the $T\bar{T}$ ones in a universal, but field-dependent fashion. These coordinates parametrize a flat target space, identified with the space of the undeformed CFT. The basis of operators considered in [44] corresponds to the original CFT operators on this space or, alternatively, their Fourier transform with respect to the target space coordinates. The correlation function of these momentum-space operators is obtained by carefully performing the JT path integral and absorbing the UV divergences into a renormalization of the operators. The procedure is rather subtle and involved, and the final result takes the form

$$\langle \mathcal{O}_1(p_1)\dots\mathcal{O}_n(p_n)\rangle$$
$$\propto \delta(\sum_{i=1}^{n} p_i) \int \prod_i d^2\sigma_i\, \delta(\sigma_1) e^{i\sum_i p_i\sigma_i} \mathcal{F}(p_i)\langle \mathcal{O}_1(\sigma_1)\dots\mathcal{O}_n(\sigma_n)\rangle_{CFT} \prod_{i<j} \left(\Lambda|\sigma_{ij}|\right)^{\frac{\mu p_i\cdot p_j}{\pi}},$$
(138)

where $\Lambda$ is a renormalization scale and $\mathcal{F}(p_i)$ is a smooth function of the momenta that grows at most polynomially at large $p_i$. If one ignores this term and considers e.g. the two-point function, one finds it corresponds precisely to the CFT momentum-space correlator

$$\langle \mathcal{O}(p,\bar{p})\mathcal{O}(-p,-\bar{p})\rangle = \frac{(2\pi)^2}{2^{2(h+\bar{h})}\sin(\pi(h+\bar{h}))} \frac{p^{2h-1}\bar{p}^{2\bar{h}-1}}{\Gamma(2h)\Gamma(2\bar{h})},$$
(139)

but with the dimensions replaced by the momentum-dependent combinations (in our conventions)

$$h = h(p,\bar{p}) = h_{CFT} + \frac{\mu}{\pi}p\bar{p}, \qquad \bar{h} = \bar{h}(p,\bar{p}) = \bar{h}_{CFT} + \frac{\mu}{\pi}p\bar{p}.$$
(140)

This is precisely the answer obtained in [25] by computing the correlation functions of long string worldsheet vertex operators, as we review in section 4.3. It is interesting to ask whether a different renormalization of the operators in [44] could reproduce this exact formula.

## $J\bar{T}$ - deformed CFTs

As already stated, the main goal of this section is to compute correlation functions of both untwisted and twisted-sector operators in single-trace $J\bar{T}$ - deformed CFTs. For this, we will adapt the prescription of [42] for defining appropriate analogues of primary operators and computing their correlation functions. In order to facilitate the generalisation of these results to the single-trace case, we present the construction of [42] in a slightly different fashion, which we hope also makes its physical interpretation more transparent.

This construction relies on the interplay of physical and flowed Virasoro generators in $J\bar{T}$ - deformed CFTs. Let us start by discussing the left-moving sector, where the meaning of the various operators is very clear. As we already explained, the standard $SL(2,\mathbb{R})_L$ conformal

symmetry of the theory unambigously identifies a set of left primary states whose conformal dimensions become momentum-dependent due to the irrelevant deformation [80]

$$h(\bar{p}) = h^{[0]} + \lambda q^{[0]}\bar{p} + \frac{\lambda^2 k \bar{p}^2}{4}, \qquad q(\bar{p}) = q^{[0]} + \frac{\lambda k \bar{p}}{2}, \tag{141}$$

where $h^{[0]}, q^{[0]}$ are the undeformed left conformal dimension and charge. Note this relation takes precisely the form of a spectral flow by $\lambda \bar{p}$. To simplify the expressions in this section, we have rescaled the deformation parameter $\lambda \to 2\pi\lambda$ with respect to the previous ones. We also set $R = 2\pi$.

The state-operator correspondence, which is still valid [81], then predicts the existence of a set of primary operators with these conformal dimensions, which obey the standard Ward identities with respect to the true conformal generators $L_m, J_m$, and for which the left-moving part of the correlation function is fixed by $SL(2,\mathbb{R})_L$ in the standard way. To construct them, one starts by formally defining the 'flowed' operators [65]

$$\partial_\lambda \widetilde{\mathcal{O}}^\lambda(\zeta,\bar{\zeta}) = [\mathcal{X}_{J\bar{T}}, \widetilde{\mathcal{O}}^\lambda(\zeta,\bar{\zeta})], \tag{142}$$

where $\zeta, \bar{\zeta}$ are the coordinates on the cylinder and $\mathcal{X}_{J\bar{T}}$ is the operator that drives the flow of the energy eigenstates in the $J\bar{T}$ - deformed CFT.[19] By construction, their correlation functions in the flowed vacuum state and their commutation relations with $\widetilde{L}_n, \widetilde{J}_n$ are identical to those in the undeformed CFT

$$[\widetilde{L}_n, \widetilde{\mathcal{O}}(\zeta,\bar{\zeta})] = e^{n\zeta}\left(n h^{[0]} \widetilde{\mathcal{O}}(\zeta,\bar{\zeta}) + \partial_\zeta \widetilde{\mathcal{O}}(\zeta,\bar{\zeta})\right), \qquad [\widetilde{J}_n, \widetilde{\mathcal{O}}(\zeta,\bar{\zeta})] = e^{n\zeta} q^{[0]} \widetilde{\mathcal{O}}(\zeta,\bar{\zeta}), \tag{143}$$

and similarly on the right-moving side. Despite this fact, the $\widetilde{\mathcal{O}}(\zeta,\bar{\zeta})$ do not correspond to physical operators in the theory and are non-local even on the left-moving side; thus, $\zeta, \bar{\zeta}$ should be simply viewed as labels inherited from the undeformed CFT.

The operators we are interested in should instead be primary with respect to the untilded Virasoro generators (85), which in our new conventions read

$$L_m = \widetilde{L}_m + \lambda H_R \widetilde{J}_m + \frac{\lambda^2 k}{4} H_R^2 \delta_{m,0}, \qquad J_m = \widetilde{J}_m + \frac{\lambda k}{2} H_R \delta_{m,0}, \tag{144}$$

with left dimensions given by (141). In order for these dimensions to make sense, they should also be eigenoperators of $H_R$, with eigenvalue $\bar{p}$. We will find it convenient to work in a mixed basis, $(\zeta, \bar{p})$. The operators should satisfy

$$[H_R, \mathcal{O}(\zeta,\bar{p})] = \bar{p}\, \mathcal{O}(\zeta,\bar{p}). \tag{145}$$

Since the relation between the undeformed and deformed dimension is given by spectral flow, the relation between the physical and tilded operator should involve dressing by an appropriate vertex operator. Note, however, that $\widetilde{\mathcal{O}}$ already carries the correct charges with respect to $J_0$ and $L_0$, implying that we should remove by hand the zero mode of the vertex operator involved. Our Ansatz is, therefore

$$\mathcal{O}(\zeta,\bar{p}) = \int d\bar{\zeta}\, e^{-\bar{\zeta}\bar{p}} : \widetilde{\mathcal{V}}\widetilde{\mathcal{O}}(\zeta,\bar{\zeta}) : e^{y_\mathcal{O}\zeta + \bar{y}_\mathcal{O}\bar{\zeta}}, \tag{146}$$

where the normal-ordered dressed operator is given by

$$: \widetilde{\mathcal{V}}\widetilde{\mathcal{O}}(\zeta,\bar{\zeta}) : \equiv \widetilde{\mathcal{V}}_\eta^+(\zeta)\widetilde{\mathcal{V}}_\eta^+(\bar{\zeta})\widetilde{\mathcal{O}}(\zeta,\bar{\zeta})\widetilde{\mathcal{V}}_\eta^-(\bar{\zeta})\widetilde{\mathcal{V}}_\eta^-(\zeta). \tag{147}$$

---

[19]The flow is defined on a fixed time slice, say $\tau = 0$. The operator at $\tau \neq 0$ is simply defined as $\mathcal{O}_{CFT}(\tau,\sigma) = e^{iH_{CFT}\tau}\mathcal{O}_{CFT}(0,\sigma)e^{-iH_{CFT}\tau}$, and the whole expression is then flowed.

Here, $\widetilde{\mathcal{V}}^{\pm}$ represent the positive and, respectively, negative-frequency parts of the dressing operator and the $\tilde{}$ stands for the fact that they are simply the $J\bar{T}$ flow of an identical expression in the undeformed CFT. With foresight, we have included vertex operator dressings for both the left and right-movers, and have left the spectral flow parameter arbitrary in order to be able to reuse this computation in the single-trace case. The argument above yields the following expression for the left dressing operators

$$\widetilde{\mathcal{V}}^{+}_{\eta}(\zeta) = e^{\eta \sum_{n=1}^{\infty} \frac{1}{n} \widetilde{J}_{-n} e^{n\zeta}}, \qquad \widetilde{\mathcal{V}}^{-}_{\eta}(\zeta) = e^{\eta\left(\widetilde{J}_0 \zeta - \sum_{n=1}^{\infty} \frac{1}{n} \widetilde{J}_n e^{-n\zeta}\right)}. \tag{148}$$

Note in (146) we have also allowed for a correction, $\mathcal{Y}_{\mathcal{O}}$, to the zero mode of the current, which commutes with all the left generators and will be fixed by the commutation relations of $\mathcal{O}(\zeta, \bar{p})$ with the physical Virasoro generators.

In order to prepare the ground for generalizing these results to the single-trace case, it will be useful to compute of the Ward identities obeyed by $\mathcal{O}(\zeta, \bar{p})$ in two steps: first, we compute the commutation relations of $:\widetilde{\mathcal{V}}\widetilde{\mathcal{O}}$ with the left-moving flowed generators using the explicit expressions (148) for the dressings; then, we assemble them into commutation relations of the non-linear combinations (144) with $\mathcal{O}(\zeta, \bar{p})$. The advantage of performing the first step separately is that the commutation relations of $:\widetilde{\mathcal{V}}\widetilde{\mathcal{O}}:$ with the flowed generators are *identical* to the corresponding expressions in the undeformed CFT, given that the flow acts by simple conjugation. The result is

$$[\widetilde{J}_m, :\widetilde{\mathcal{V}}\widetilde{\mathcal{O}}(\zeta, \bar{\zeta}):] = e^{m\zeta}\left(q^{[0]} + \frac{\eta k}{2}(1 - \delta_{m,0})\right) :\widetilde{\mathcal{V}}\widetilde{\mathcal{O}}(\zeta, \bar{\zeta}):, \tag{149}$$

$$[\widetilde{L}_m, :\widetilde{\mathcal{V}}\widetilde{\mathcal{O}}(\zeta, \bar{\zeta}):] = e^{m\zeta}\left(m(h^{[0]} + \eta q^{[0]} + \frac{k\eta^2}{4}) - \frac{\eta^2 k}{4}(1 - \delta_{m,0}) + \partial_{\zeta}\right) :\widetilde{\mathcal{V}}\widetilde{\mathcal{O}}(\zeta, \bar{\zeta}): -\eta :\widetilde{\mathcal{V}}\widetilde{\mathcal{O}}(\zeta, \bar{\zeta}): \widetilde{J}_m,$$

and corresponds, as it is clear from (148), to the CFT Ward identities for a normal-ordered left vertex operator of charge $q^{[0]} + \eta k/2$, but with part of its zero mode left out (so that the charge carried with respect to $\widetilde{J}_0$ is just $q^{[0]}$). Note the $\widetilde{\mathcal{V}}_{\eta}$ (constructed from right-moving current modes only) do not affect the left Ward identities.

It is then not hard to check that the commutation relations of $\mathcal{O}(\zeta, \bar{p})$ with the generators (144) are precisely those of a primary of the expected dimension and charge, provided we set $\eta = \lambda \bar{p}$ and

$$\mathcal{Y}_{\mathcal{O}} = \lambda q H_R + \lambda \bar{p} q^{[0]} + \frac{k\lambda^2 \bar{p}^2}{4}. \tag{150}$$

This can also be seen from the fact that the left-moving piece of $\mathcal{O}(\zeta, \bar{p})$ organises into a left vertex operator of charge $q$ in the deformed theory, i.e. constructed from the current modes $J_m$: the dressings (148) shift the charge from $q^{[0]}$ to $q$ for the non-zero modes, whereas the the first term in the expression for $\mathcal{Y}_{\mathcal{O}}$ shifts the $\widetilde{J}_0$ term in (146) (whose total coefficient is $2q/k$) to $J_0$. The last two terms correspond to the difference $h(\bar{p}) - h^{[0]}$ in the operator dimensions, which becomes important when translating the operators defined on the cylinder to those on the plane via the standard map $\mathcal{O}_{(cyl)}(\zeta) = e^{h\zeta} \mathcal{O}_{(pl)}(z)$. Finally, the total charge of the operator, as measured by $J_0$, is $q$. One may check that, upon performing a conformal transformation to the plane, the left-moving piece of (146) corresponds precisely to a local left vertex operator in the deformed theory with charge $q$, with no mismatch between the coefficients of the zero and non-zero modes.

Having understood in detail the construction of the left-moving piece of the operator - which is entirely fixed by conformal symmetry - via the dressing discussed above, the proposal of [42] consists in choosing the dressing operators for the right-movers to have an identical form, but with all quantities barred. Requiring that the resulting operator be an eigenoperator

of $H_R$,[20] we obtain the following expression for $\bar{\mathcal{Y}}_{\mathcal{O}}$

$$\bar{\mathcal{Y}}_{\mathcal{O}} = \lambda\bar{p}(\widetilde{J}_0 - \widetilde{\bar{J}}_0) + \lambda q H_R + \lambda\bar{p}q^{[0]} + \frac{k\lambda^2\bar{p}^2}{4} \,. \tag{151}$$

The right-moving piece of $:\widetilde{\mathcal{V}}\widetilde{\mathcal{O}}:$ can again be interpreted as a vertex operator in the unde-formed CFT, of charge $\bar{q}^{[0]} + k\eta/2$, but with a discrepancy between the coefficient of the zero and non-zero modes, which is then trivially flowed to the deformed theory. A factor of $\lambda\bar{q}H_R$ in the expression for $\bar{\mathcal{Y}}_{\mathcal{O}}$ could again be understood as a correction to the zero mode of the right-moving current. However, since this term does not commute with the right current gen-erators, we no longer have a useful interpretation for the full operator as a vertex operator in the deformed theory. In addition, there are a number of discrepancies involving winding between the operator we obtain and a naïve spectrally-flowed operator on the right-moving side, at least at finite $R$. These discrepancies can presumably be traced back to the fact that $\bar{L}_0$ and $H_R$ differ by winding terms. In any case, given the explicit form (146) of the operator, the the commutation relations with the generators (86) can be shown to take the form of CFT Ward identities in the limit $R \to \infty$, with

$$\bar{h}(\bar{p}) = \bar{h}^{[0]} + \lambda\bar{q}^{[0]}\bar{p} + \frac{k\lambda^2\bar{p}^2}{4} \,, \qquad \bar{q}(\bar{p}) = \bar{q}^{[0]} + \frac{k\lambda\bar{p}}{2} \,, \tag{152}$$

which resembles a spectral flow transformation by $\lambda\bar{p}$ of the right-moving conformal dimen-sions and charges.

   Let us now compute correlation functions of these operators. Since $\mathcal{Y}_{\mathcal{O}}, \bar{\mathcal{Y}}_{\mathcal{O}}$ only involve $H_R$ and the winding and the $\mathcal{O}(\zeta, \bar{p})$ are eigenoperators of both, the correlation function can be simplified to

$$\langle \mathcal{O}_1(\zeta_1, \bar{p}_1) \dots \mathcal{O}_n(\zeta_n, \bar{p}_n) \rangle = \int \prod_{i=1}^n d\bar{\zeta}_i \, e^{-\sum_i \bar{p}_i\bar{\zeta}_i} e^{\sum_i (\lambda\bar{p}_i q_i^{[0]} + \frac{k\lambda^2\bar{p}_i^2}{4})(\zeta_i + \bar{\zeta}_i)} e^{\lambda H_R^{vac} q_i(\zeta_i + \bar{\zeta}_i)}$$

$$\times \, e^{\sum_{i<j}(\lambda\bar{p}_i(q_j^{[0]} - \bar{q}_j^{[0]})\bar{\zeta}_i + \lambda q_i\bar{p}_j(\zeta_i + \bar{\zeta}_i))} \langle :\widetilde{\mathcal{V}}_1\widetilde{\mathcal{O}}_1(\zeta_1, \bar{\zeta}_1): \dots :\widetilde{\mathcal{V}}_n\widetilde{\mathcal{O}}_n(\zeta_n, \bar{\zeta}_n): \rangle, \tag{153}$$

where $H_R^{vac}$ represents the eigenvalue of $H_R$ in the flowed vacuum. The last term in the in-tegrand is almost a correlation function of vertex operators of charges $q_i$ in the undeformed CFT, up to the missing zero modes. Using the explicit expressions for the vertex operators to evaluate the zero mode contribution, we obtain

$$\langle :\widetilde{\mathcal{V}}_1\widetilde{\mathcal{O}}_1(\zeta_1, \bar{\zeta}_1): \dots :\widetilde{\mathcal{V}}_n\widetilde{\mathcal{O}}_n(\zeta_n, \bar{\zeta}_n): \rangle = e^{-\sum_{i<j} \lambda\bar{p}_j(q_i\zeta_i + \bar{q}_i\bar{\zeta}_i)} e^{-\sum_i(\lambda\bar{p}_i(q_i^{[0]}\zeta_i + \bar{q}_i^{[0]}\bar{\zeta}_i) + \frac{k\lambda^2\bar{p}_i^2}{4}(\zeta_i + \bar{\zeta}_i))}$$

$$\times \prod_{i<j} \left( e^{\frac{\zeta_{ij}}{2}} - e^{-\frac{\zeta_{ij}}{2}} \right)^{\frac{2}{k}(q_i q_j - q_i^{[0]} q_j^{[0]})} \left( e^{\frac{\bar{\zeta}_{ij}}{2}} - e^{-\frac{\bar{\zeta}_{ij}}{2}} \right)^{\frac{2}{k}(\bar{q}_i\bar{q}_j - \bar{q}_i^{[0]}\bar{q}_j^{[0]})} \langle \widetilde{\mathcal{O}}_1(\zeta_1, \bar{\zeta}_1) \dots \widetilde{\mathcal{O}}_n(\zeta_n, \bar{\zeta}_n) \rangle, \tag{154}$$

where $\zeta_{ij} = \zeta_i - \zeta_j$, $\bar{\zeta}_{ij} = \bar{\zeta}_i - \bar{\zeta}_j$. The second line simply represents the spectrally-flowed correlation function on the cylinder. The first exponential factor on the first line corresponds to the correction due to the missing zero modes, while the second one accounts for the change in the definition of the cylinder operators due to the shift in their dimensions. The same result can be obtained by commuting the current modes until they annihilate the (flowed) vacuum. Plugging in this expression into (153) and reinstating the factors of the radius, the final result

---

[20]In practice, this is obtained by writing the commutator of $\mathcal{O}$ with $\widetilde{L}_0$ in two different ways, see [42] for details.

for the $n$-point function simplifies to

$$\langle \mathcal{O}_1(\zeta_1, \bar{p}_1) \dots \mathcal{O}_n(\zeta_n, \bar{p}_n) \rangle = \int \prod_{i=1}^{n} d\bar{\zeta}_i \, e^{-\sum_i \bar{p}_i \bar{\zeta}_i} \, e^{\frac{2\pi}{R} \mathcal{A}} \prod_{i<j} \left( e^{\frac{\pi \zeta_{ij}}{R}} - e^{-\frac{\pi \zeta_{ij}}{R}} \right)^{\frac{2}{k}(q_i q_j - q_i^{[0]} q_j^{[0]})}$$

$$\times \left( e^{\frac{\pi \bar{\zeta}_{ij}}{R}} - e^{-\frac{\pi \bar{\zeta}_{ij}}{R}} \right)^{\frac{2}{k}(\bar{q}_i \bar{q}_j - \bar{q}_i^{[0]} \bar{q}_j^{[0]})} \langle \mathcal{O}_1^{CFT}(\zeta_1, \bar{\zeta}_1) \dots \mathcal{O}_n^{CFT}(\zeta_n, \bar{\zeta}_n) \rangle, \quad (155)$$

where we plugged in the fact that the correlation function of the $\widetilde{\mathcal{O}}$ is identical to the corresponding undeformed CFT correlator and the exponent $\mathcal{A}$ can be written, using charge conservation, as

$$\mathcal{A} = \sum_{i=1}^{n} \lambda H_R^{vac} q_i (\zeta_i + \bar{\zeta}_i) + \sum_{i<j} \lambda \bar{p}_j (q_i - \bar{q}_i) \bar{\zeta}_{ij}. \quad (156)$$

If we ignore this exponential factor, then the result of the integral is simply the spectrally-flowed CFT correlation function in the mixed $(\zeta_i, \bar{p}_i)$ basis. Note that the spectral flow only affects the operator dimensions, but not the OPE coefficients of appropriate Virasoro-Kac-Moody primaries. In particular, for the case of two-and three-point function, the spectral flow will simply shift the operator dimensions to the momentum-dependent combinations (141), (152) in the corresponding mixed position/momentum-space correlator. Including the exponential factor will simply shift the coefficients of the $\bar{\zeta}_i$ in the Fourier integral by a factor that, importantly, is proportional to $1/R$. The result will be the spectrally-flowed CFT momentum-space correlator evaluated at momenta that differ from $\bar{p}_i$ by terms that vanish in the decompactification limit.

Note that on the left-moving side, the only term that could potentially upset the conformal invariance of the spectrally-flowed correlator is the shift proportional to $H_R^{vac} \zeta_i$ in the expression for $\mathcal{A}$. As discussed in [42], this discrepancy is related to the lack of standard $SL(2, \mathbb{R})_L$ invariance of the flowed vacuum in finite size, which also disappears in the $R \to \infty$ limit.

## 4.2 Correlation functions in single-trace $J\bar{T}$ - deformed CFTs

Let us, for completeness, start with some generalities. There are several classes of operators one may discuss in a symmetric product orbifold QFT: untwisted or twisted, single-trace or multitrace. Operators that are untwisted take the form

$$\sum_{\sigma \in S^N} \mathcal{O}_{\sigma(1)}^1 \otimes \mathcal{O}_{\sigma(2)}^2 \otimes \dots \otimes \mathcal{O}_{\sigma(N)}^N, \quad (157)$$

where all insertions are at the same position $(\zeta, \bar{\zeta})$. In the particular case where all but one operator are the identity, this reduces to $\sum_I \mathcal{O}_I(\zeta, \bar{\zeta})$, which is a single-trace untwisted operator. When more than one insertions are non-trivial, the operator is called multitrace. The Fourier transform of these operators can be written as a sum over copies, e.g. $\sum_I \mathcal{O}_I(p, \bar{p})$ in the single-trace case, provided one acts on a state from the untwisted sector. The multi-trace untwisted operators take a similar form, but with several sums over copies and an integral over all the possible ways to distribute the momenta among them: e.g., in the double-trace case, the relevant operators take the form $\sum_{I,J} \int d^d k \, \mathcal{O}_I^1(k) \mathcal{O}_J^2(p - k)$. The untwisted-sector correlation functions of such operators can be straightforwardly computed using those of the seed: in the single-trace case, they are simply proportional to seed correlators; for multi-trace operators, one obtains a momentum-space convolution of seed correlation functions, which may be evaluated if the latter are known explicitly.

Thus, all the new correlators with respect to the seed lie in the twisted sector. One may again classify the twisted-sector operators into single-trace and multi-trace. For the single-trace operators, the twist involves only one non-trivial cycle; for the multitrace ones, several

cycles should be considered. The end result should of course be symmetrized with respect to all permutations of the various copies.

The construction of twisted-sector correlation functions in single-trace $T\bar{T}$ and $J\bar{T}$ - deformed CFTs should depend on the particular method used to build the corresponding double-trace correlators. In the $T\bar{T}$ case, the method of [44] uses the JT formulation of the deformation, which involves maps from the space where the theory is defined to a flat target space. Given the similarity between this formalism and one involving the string worldsheet, for example in what regards the computation of the finite-size spectrum [26,27], it appears reasonable to assume that the correlation functions in the twisted sectors will be captured by considering "worldsheets" in the JT formulation that wind around the target space. The string worldsheet computation of [25] suggests the effect of the winding on the correlation functions is extremely straightforward, as it simply amounts to replacing the $T\bar{T}$ coupling $\mu$ by $\mu/w$ in the twisted sectors.

The main goal of this section is to put forth an appropriate basis of twisted-sector operators in single-trace $J\bar{T}$ - deformed CFTs and to compute their correlation functions. We will be following step-by-step the procedure used in the double-trace case. We will concentrate on single-trace operators, which contain a twist corresponding to a single non-trivial cycle of length $w$, inserted at their location. The operators of interest are primary on the left-moving side, with the following momentum-dependent $SL(2,\mathbb{R})_L$ dimensions and $U(1)$ charges

$$h^{(w)}(\bar{p}) = h^{(w)}_{CFT} + \frac{\lambda \bar{p}}{w} q^{[0]} + \frac{k\lambda^2 \bar{p}^2}{4w}, \quad q = q^{[0]} + \frac{\lambda k \bar{p}}{2}, \tag{158}$$

which were computed in section 2.3 via an infinite boost of the twisted-sector deformed energies. $h^{(w)}_{CFT}$ is the dimension of the twisted-sector operator in the undeformed symmetric product orbifold CFT, which is related to a seed dimension by the holomorphic counterpart of (47). Note $\lambda$ has been rescaled by a factor of $2\pi$ with respect to (65). The extension to several non-trivial cycles is straightforward, as the dimensions are additive.

The relation between the deformed and undeformed twisted-sector dimensions corresponds to a transformation known as *fractional* spectral flow [61, 62], defined as the base space transformation that lifts to the standard spectral flow on the covering space. It acts as

$$h^{(w)} \to h^{(w)} + \eta q^{(w)} + \frac{kw\eta^2}{4}, \qquad q^{(w)} \to q^{(w)} + \frac{kw\eta}{2}, \tag{159}$$

on the base. In the standard discussions, $\eta$ is taken to be fractional, hence the name. In our case, $\eta = \lambda \bar{p}/w$ is a continuous parameter, so its fractionality is not very important; what is important is that the level that enters the spectral flow transformation is the effective level in the $w$-twisted sector, $k^{(w)} = kw$.

As argued in section 3.4, the physical generators of conformal symmetries in the $w$ - twisted sectors are the integer modes[21] of (131), which are related to the flowed generators in that sector by a transformation resembling fractional spectral flow with parameter $\lambda H_R/w$

$$L_m = \widetilde{L}_m + \frac{\lambda}{w} H_R \widetilde{J}_m + \frac{\lambda^2 k}{4w} H_R^2 \delta_{m,0}, \qquad J_m = \widetilde{J}_m + \frac{\lambda k}{2} H_R \delta_{m,0}, \tag{160}$$

where we have again set $R = 2\pi$, rescaled $\lambda$ by a $2\pi$ factor and dropped the index indicating the twist sector from $H_R$. Throughout this section, we will work with operators corresponding

---

[21]In principle, one could also consider fractional Virasoro modes around the location of the twist operator. The cylinder fractional modes appearing in (131) are associated to a twist operator inserted at $-\infty$, whereas in this section we will be considering twists inserted at finite distance on the cylinder. The two sets of fractional generators can be related provided no other operator insertion is present - an overly restrictive requirement. On the other hand, the integer-moded Virasoro generators are everywhere well-defined.

to the first $w$ copies of the seed QFT, and symmetrization is assumed only for the final expressions. Our first task is to construct a basis of operators that are primary with respect to the generators above, with conformal dimensions and charges given in (158).

We may again proceed by defining a flowed operator $\widetilde{\mathcal{O}}^{(w)}(\zeta, \bar{\zeta})$ via an equation of the form (142), with the initial condition that at $\lambda = 0$, the operator is a $w$-twisted operator in the undeformed CFT.[22] This operator will have the same conformal dimension and will satisfy the same Ward identities with respect to the flowed Virasoro and Kac-Moody currents as in the undeformed theory. As in the previous subsection, one should add an appropriate dressing to this operator, so it becomes primary with respect to the standard conformal generators, with the expected dimension (65). However, since the relation between the deformed and undeformed dimension is now given by *fractional* spectral flow, we can no longer write a simple, explicit expression for the dressing vertex as in (148); instead, any explicit expression would involve fractional current modes. Since the fractional modes are defined only locally around operators of generically different twists, it does not make sense to commute them as we did in the double-trace case, and thus we need a different way to evaluate the correlation functions of interest.

The approach that we will instead take will be to reformulate, to the largest extent possible, the computations in terms of undeformed CFT ones - where the lift to the appropriate covering space is standard - and a $J\bar{T}$ flow, which does not affect the end result for correlators and commutators. Concretely, we will write, in analogy with (146)

$$\mathcal{O}^{(w)}(\zeta, \bar{p}) = \int d\bar{\zeta} \, e^{-\bar{\zeta}\bar{p}} : \widetilde{\mathcal{V}}_\eta \widetilde{\mathcal{O}}^{(w)}(\zeta, \bar{\zeta}) : e^{\mathcal{Y}_{\mathcal{O}}^{(w)}\zeta + \bar{\mathcal{Y}}_{\mathcal{O}}^{(w)}\bar{\zeta}}, \tag{161}$$

where $: \widetilde{\mathcal{V}}_\eta \widetilde{\mathcal{O}}^{(w)} :$ now represents the single-trace $J\bar{T}$ flow of an operator in the undeformed theory that is almost given by the fractional spectral flow, with parameter $\eta = \lambda \bar{p}/w$, of the original $w$-twisted CFT operator, up to some missing zero modes and some rescaling factors. The fractional spectral flow in the undeformed CFT may be implemented by lifting to the covering space, where it becomes usual spectral flow. Guided by the double-trace case, one should subtract by hand[23] from the result the zero mode of the dressing vertex operator, so that its commutation relations with the undeformed CFT generators - which, upon $J\bar{T}$ flow, will be the same as those of the flowed generators with $: \widetilde{\mathcal{V}}\widetilde{\mathcal{O}} :$ - take precisely the form (149), but with $k \to kw$. Note that, in order to reach this conclusion, we do not absolutely need the explicit form of $\widetilde{\mathcal{V}}$.

Using the fact that, by assumption, $\mathcal{O}^{(w)}(\zeta, \bar{p})$ is an eigenoperator of the right Hamiltonian $H_R$, with eigenvalue $\bar{p}$, and requiring that the commutation relations with the generators (160) are CFT Ward identities for operators with conformal dimension (65), we find $\eta = \lambda \bar{p}/w$, and

$$\mathcal{Y}_{\mathcal{O}}^{(w)} = \frac{1}{w}\left( \lambda q H_R + \lambda \bar{p} q^{[0]} + \frac{k\lambda^2 \bar{p}^2}{4} \right). \tag{162}$$

Assuming that, as in the double-trace case, the commutation relation of $\widetilde{\bar{L}}_0$ with $: \widetilde{\mathcal{V}}\widetilde{\mathcal{O}}(\zeta, \bar{\zeta}) :$ still yields a simple $\bar{\zeta}$ derivative of the operator, we also deduce that

$$\bar{\mathcal{Y}}_{\mathcal{O}}^{(w)} = \frac{1}{w}\left( \lambda q H_R + \lambda \bar{p} q^{[0]} + \frac{k\lambda^2 \bar{p}^2}{4} + \lambda \bar{p}(\tilde{J}_0 - \bar{\tilde{J}}_0) \right). \tag{163}$$

---

[22]Strictly speaking, since $\mathcal{O}^{(w)}$ is only associated with the first $w$ copies of the QFT, it should be flown with the flow operator associated with these copies - the quantum analogue of (123). While this analogue of (142) is a well-defined operator equation, it is difficult to lift it to the covering space due to the presence of the twist operator at the location of $\widetilde{\mathcal{O}}^{(w)}$. In particular, the mapping to the larger cylinder used in section 3.4, which did not rely on conformal invariance, cannot be used anymore.

[23]Note that this procedure is not as innocuous as it may sound. In the CFT, the transformation under the covering map is understood for local operators, but once we remove some zero modes e.g. from the operator on the covering space, it is no longer clear how to map it back to the base.

Thus, choosing $:\widetilde{\mathcal{V}}\widetilde{\mathcal{O}}:$ to be a fractionally spectrally flowed operator on the left (with some zero modes removed) reproduces the correct left Ward identities, even in absence of explicit expressions for the dressing factors. Our proposal for the right-moving piece of this operator - which will fix the basis (161) of operators we consider in single-trace $J\bar{T}$ - deformed CFTs - is that it behaves exactly the same in the undeformed CFT, i.e. it is a fractional spectral flow on the right with parameter $\lambda\bar{p}/w$. The commutation relations with the flowed right-moving generators then follow from the CFT ones, and are given by (149) with all quantities barred and $k \to kw$. The commutation relations of $\mathcal{O}^{(w)}(\zeta,\bar{p})$ with $\bar{L}_m, \bar{J}_m$, whose relationship to the flowed right-moving generators is given by (160) with all quantities barred, then follows straightforwardly from their definitions. It is not hard to show that in the $R \to \infty$ limit , they correspond to standard CFT right-moving Ward identities with

$$\bar{h}^{(w)}(\bar{p}) = \bar{h}^{(w)}_{CFT} + \frac{\lambda\bar{q}^{[0]}\bar{p}}{w} + \frac{k\lambda^2\bar{p}^2}{4w}, \qquad \bar{q}(\bar{p}) = \bar{q}^{[0]} + \frac{k\lambda\bar{p}}{2}, \qquad (164)$$

which resembles a right-moving spectral flow with parameter $\lambda\bar{p}/w$.

Let us now discuss the correlations functions of these operators, which take the form

$$\langle \mathcal{O}^{(w_1)}_1(\zeta_1,\bar{p}_1)\dots\mathcal{O}^{(w_n)}_n(\zeta_n,\bar{p}_n)\rangle, \qquad (165)$$

where each $\mathcal{O}^{(w_i)}_i$ is now a gauge-invariant operator in the $w_i$-twisted sector of the theory, and we assume that the twists are such that the correlation function is non-vanishing. For simplicity, we will consider only correlation functions of single-trace twist operators.

As in the previous section, we can evaluate the correlation function in two steps: first, we use the fact that (161) are eigenoperators of $\mathcal{Y}^{(w_i)}_{\mathcal{O}}, \bar{\mathcal{Y}}^{(w_i)}_{\mathcal{O}}$ to reduce the computation to the evaluation of a correlation function of "almost CFT vertex operators", as in (153)

$$\langle \mathcal{O}^{(w_1)}_1(\zeta_1,\bar{p}_1)\dots\mathcal{O}^{(w_n)}_n(\zeta_n,\bar{p}_n)\rangle = \int \prod_{i=1}^n d\bar{\zeta}_i\, e^{-\sum_i \bar{p}_i\bar{\zeta}_i}\, e^{\sum_i (\lambda\bar{p}_i q_i^{[0]} + \frac{k\lambda^2\bar{p}_i^2}{4})\frac{(\zeta_i+\bar{\zeta}_i)}{w_i}}\, e^{\lambda H_R^{vac} q_i \frac{(\zeta_i+\bar{\zeta}_i)}{w_i}}$$

$$\times\, e^{\sum_{i<j} \frac{1}{w_i}(\lambda\bar{p}_i(q_j^{[0]}-\bar{q}_j^{[0]})\bar{\zeta}_i + \lambda q_i\bar{p}_j(\zeta_i+\bar{\zeta}_i))}\, \langle :\widetilde{\mathcal{V}}_1\widetilde{\mathcal{O}}^{(w_1)}_1(\zeta_1,\bar{\zeta}_1): \dots :\widetilde{\mathcal{V}}_n\widetilde{\mathcal{O}}^{(w_n)}_n(\zeta_n,\bar{\zeta}_n):\rangle. \qquad (166)$$

The computation of the correlator of $:\widetilde{\mathcal{V}}_i\widetilde{\mathcal{O}}_i:$ can then be traced back (by inverting the $J\bar{T}$ flow) to the cylinder correlation function of the original CFT operators, fractionally spectrally flowed by an amount $\lambda\bar{p}_i/w_i$, up to some missing zero modes. Note that, unlike for the standard spectral flow, whose action on a general correlation function is known in closed form (154), for fractional spectral flow involving operators from different twisted sectors no such formula appears to exist. On the other hand, there does exist a well-defined prescription for computing such a correlation function by mapping the CFT correlators to the covering space, so we may consider this part of the problem as being in principle solved.

The additional issue of subtracting the zero modes is complicated by the fact that: i) unlike in the double-trace case, we no longer have a globally defined explicit expression for the dressing vertex operators and ii) the maps to the covering space are for local operators, but locality is spoiled by the removal of the zero modes. Note one constraint on the corrections is that they should be consistent with translation invariance on the cylinder, which the prefactor in (166) currently violates, as did the analogous prefactor in (153). We will not attempt to estimate the zero-mode contribution herein. Instead, we argue, guided by the explicit expressions in the double-trace case, that the corrections from such zero modes will take the form of exponential factors, which are all suppressed in the $R \to \infty$ limit, and will thus not contribute to the final correlation function on the plane. Thus, in this limit we are left with usual

CFT $n$-point function in mixed position/ momentum space, for twisted-sector operators with fractionally spectrally flowed conformal dimensions given by (158), (164)

$$\langle \mathcal{O}_1^{(w_1)}(\zeta_1, \bar{p}_1) \dots \mathcal{O}_n^{(w_n)}(\zeta_n, \bar{p}_n) \rangle_{plane} \tag{167}$$
$$= \int \prod_{i=1}^{n} d\bar{\zeta}_i \, e^{-\sum_i \bar{p}_i \bar{\zeta}_i} \langle :(\mathcal{V}_1 \mathcal{O}_1^{(w_1)})_{CFT}(\zeta_1, \bar{\zeta}_1): \dots :(\mathcal{V}_n \mathcal{O}_n^{(w_n)})_{CFT}(\zeta_n, \bar{\zeta}_n): \rangle_{plane}.$$

Thus, we have succeeded in evaluating correlation functions of a particular basis of of twisted-sector operators in single-trace $J\bar{T}$ - deformed CFTs in terms of correlation functions of the undeformed theory. In order to compare with the holographic computation of the next section, it is useful to fully transform the result to momentum space and directly consider the Fourier transform of the euclidean correlator. For the specific case of the two-point function, the result is

$$\langle \mathcal{O}^{(w)\dagger}(p, \bar{p}) \mathcal{O}^{(w)}(-p, -\bar{p}) \rangle = \frac{(2\pi)^2}{2^{2(h^{(w)}+\bar{h}^{(w)})} \sin(\pi(h^{(w)} + \bar{h}^{(w)}))} \frac{p^{2h^{(w)}-1} \bar{p}^{2\bar{h}^{(w)}-1}}{\Gamma(2h^{(w)})\Gamma(2\bar{h}^{(w)})}, \tag{168}$$

where $h^{(w)}, \bar{h}^{(w)}$ stand for the momentum-dependent combinations (158), (164).

### 4.3  Comparison with holographic results

The single-trace $T\bar{T}$ and $J\bar{T}$ deformations have been linked to holography for certain non-AdS backgrounds, namely an asymptotically linear dilaton spacetime for the case of $T\bar{T}$ [8], and warped AdS$_3$ for $J\bar{T}$ [9, 10]. Both of these backgrounds are supported by pure NS-NS flux and flow to AdS$_3$ in the interior; the full spacetimes can be viewed as non-normalizable deformations thereof. Perturbative worldsheet string theory in these backgrounds is given by the $SL(2, \mathbb{R})$ WZW model (for level $N_5 > 1$), deformed by a certain class of exactly marginal current-current operators. Such deformations are exactly solvable. In certain examples, the deformed string background can be thought of as the near-horizon geometry of a stack of $N_5$ NS5 branes and $N_1 \gg 1$ F1 strings [8, 82].

For $N_5 > 1$, the full CFT dual to the infrared AdS$_3$ is known to *not* be described by a symmetric product orbifold. It is a somewhat singular theory [5], due to the presence of states with a continuous spectrum known as long strings, which wind around the asymptotic AdS$_3$ boundary. Its long string subsector has, on the other hand, been argued to be described by a symmetric orbifold [5, 83]; note, however, that it captures only a small fraction of the system, at least in the regime of interest [83]. Given this structure, it should be clear that the symmetric product orbifold of $T\bar{T}/J\bar{T}$-deformed CFTs cannot be exactly dual to these non-AdS backgrounds for $N_5 > 1$, because the remaining sectors do not possess a symmetric orbifold structure;[24] at the same time, the long string subsector survives in the deformed backgrounds and, moreover, its dynamics are well-described by single-trace $T\bar{T}$ and, respectively, $J\bar{T}$ - deformed CFTs. Concretely, the spectrum of long strings has been shown [8–10, 18] to perfectly match that of single-trace $T\bar{T}/ J\bar{T}$ (as derived in this article, or also in [30]); more recently, correlation functions of long string vertex operators in the asymptotically linear dilaton background have been computed [30] and appear to match well with the recent $T\bar{T}$ results of [44].

The aim of this section is to compare the correlation functions of our proposed set of "primary operator analogues" in single-trace $J\bar{T}$ - deformed CFTs with the those of long string vertex operators in warped AdS$_3$. The latter will be estimated by adapting the short-string

---

[24]Nevertheless, there exists an apparently independent link between asymptotically linear dilaton backgrounds and single-trace $T\bar{T}$, suggested by the fact that black hole entropy [8,82,84] and the infinite asymptotic symmetries of the linear dilaton spacetime [40] precisely agree with those of single-trace $T\bar{T}$. Since only universal quantities match, this suggests the dual could be part of a "universality class" of $T\bar{T}$-deformed CFTs.

computation of [20] for the same background to long strings, along the lines of [25]. The two results turn out to not exactly match. To explain this, we first review some relevant prior work.

Let us start by considering worldsheet vertex operators of type II superstring theory in $AdS_3 \times \mathcal{N}$ in the presence of pure NS-NS flux. Here $\mathcal{N}$ is a 7-dimensional compact manifold. Long strings correspond to worldsheet vertex operators that belong to the continuous series representation of $SL(2, \mathbb{R})$. Their worldsheet dimension is given by

$$
\begin{aligned}
\Delta &= -\frac{j(j+1)}{N_5} - w\left(h + \frac{N_5 w}{4}\right) + \Delta_{\mathcal{N}} + N, \\
\bar{\Delta} &= -\frac{j(j+1)}{N_5} - w\left(\bar{h} + \frac{N_5 w}{4}\right) + \bar{\Delta}_{\mathcal{N}} + \bar{N},
\end{aligned}
\tag{169}
$$

where $j \in -1/2 + i\mathbb{R}$ labels the Casimir of the global $SL(2, \mathbb{R})$ algebra, $w \geq 1$ denotes the integer spectral flow in $AdS_3$ - identified with the winding of the long string around the AdS$_3$ boundary, $(\Delta_{\mathcal{N}}, \bar{\Delta}_{\mathcal{N}})$ are left/right vertex operator dimensions of the worldsheet CFT in $\mathcal{N}$, $(N, \bar{N})$ are the left/right oscillator numbers in $AdS_3$ and, finally, $(h, \bar{h})$ represent the eigenvalues of the $J^3, \bar{J}^3$ zero modes of the worldsheet $SL(2, \mathbb{R})$. Note that for continuous series representations, these are unrelated to the eigenvalue of the Casimir. In global AdS$_3$, $(h, \bar{h})$ are identified with the left/right energies of the state on the cylinder, and thus to the dual operator dimensions via the standard state-operator map. The physical on-shell condition for these superstring vertex operators is $\Delta = \bar{\Delta} = 1/2$. They can be constructed explicitly [85], and their correlation function takes the form[25]

$$
\langle V(z_1, x_1) V(z_2, x_2) \rangle = \frac{1}{z_{12}^{2\Delta} \bar{z}_{12}^{2\bar{\Delta}} x_{12}^{2h} \bar{x}_{12}^{2\bar{h}}},
\tag{170}
$$

where $x, \bar{x}$ are auxiliary coordinates that become identified with the space where the dual CFT lives. Integrating over the worldsheet coordinates, one obtains the standard correlation function of CFT operators on the boundary. One may also perform a Fourier transform of the latter, to obtain the momentum-space boundary correlator. For example, the momentum space two-point function takes the form

$$
\langle V(z_1, p) V(z_2, -p) \rangle = \frac{(2\pi)^2 p^{2h-1} \bar{p}^{2\bar{h}-1}}{2^{2(h+\bar{h})} \sin(\pi(h+\bar{h})) \Gamma(2h) \Gamma(2\bar{h})} \frac{1}{z_{12}^{2\Delta} \bar{z}_{12}^{2\bar{\Delta}}},
\tag{171}
$$

which will be useful in this section.

The asymptotically linear dilaton and warped AdS$_3$ backgrounds can both be obtained from AdS$_3 \times \mathcal{N}$ via a transformation known as TsT: T-duality, shift, T-duality. In both cases, the effect of the TsT transformation can be encoded in a non-local coordinate transformation, which is equivalent to twisting the boundary conditions of the fields on the AdS$_3$ worldsheet theory in a charge-dependent way [86]. This mildly affects the relationship between $\Delta$ and $h$, by adding

$$
\delta_{T\bar{T}}\Delta = \delta_{T\bar{T}}\bar{\Delta} = \frac{\mu}{\pi} p\bar{p}, \qquad \delta_{J\bar{T}}\Delta = \delta_{J\bar{T}}\bar{\Delta} = \lambda \bar{p}\left(q^{[0]} + \frac{\lambda k \bar{p}}{4}\right),
\tag{172}
$$

on the left-hand side of (169). If $(h, \bar{h})$ are interpreted as the left/right global energy, then these shifts yield the correct deformed energy formulae to match to single-trace $T\bar{T}/J\bar{T}$.

The vertex operators in the deformed theory may be expressed in terms of the vertex operators in the undeformed theory, with some appropriate dressing. This has been worked out

---

[25]In this article, we normalize the worldsheet operators such that the two-point function of the dual CFT operators takes the form $x_{12}^{-2h} \bar{x}_{12}^{-2\bar{h}}$. Note this normalization is different from the standard convention in string theory in $AdS_3$.

in [20] for the warped $AdS_3$ background and in [25] for the asymptotically linear dilaton one; see also [24] for relevant prior results. It turns out that the correlation function of the appropriate vertex operators still takes the form (171), but the relationship between $h$ and $\Delta$ is modified by the shifts (172). Since $\Delta$ must equal $1/2$, this translates into a shift of $h, \bar{h}$ that takes the form:

$$
\begin{aligned}
T\bar{T} : \; & h \to h + \frac{\mu}{w\pi} p\bar{p} \,, & \bar{h} &\to \bar{h} + \frac{\mu}{w\pi} p\bar{p} \,, \\
J\bar{T} : \; & h \to h + \frac{\lambda q^{[0]}\bar{p}}{w} + \frac{\lambda^2 k \bar{p}^2}{4w} \,, & \bar{h} &\to \bar{h} + \frac{\lambda q^{[0]}\bar{p}}{w} + \frac{\lambda^2 k \bar{p}^2}{4w} \,,
\end{aligned}
\tag{173}
$$

and can be obtained by combining (169) with the appropriate shift in (172). Thus, we obtain a two-point function that is identical to the momentum-space two-point function in a CFT, but with shifted dimensions.[26] The $T\bar{T}$ shift of $h, \bar{h}$ nicely agrees with [44] for the case $w = 1$. More generally, one obtaines the shifts in the twisted sectors to be the same, but controlled by $\mu/w$.

Note that the shift in the left-moving single-trace $J\bar{T}$ dimension precisely agrees with our previous analysis (65), including the $w$ dependence. However, the right-moving piece of the correlator (171) does not agree with (168), which involves $\bar{q}^{[0]}$ instead of $q^{[0]}$ in the right-moving dimension $\bar{h}^{(w)}(\bar{p})$. It would be interesting to understand the origin of this mismatch. Remember, in particular, that in our field-theory analysis we did encounter shifts of the right-movers that involved $J_0$ instead of $\bar{J}_0$; however, the discrepancies produced by these terms disappeared in the $R \to \infty$ limit.[27] It would be interesting to perform a more careful worldsheet analysis, possibly on the cylinder, in order to track this discrepancy. Of course, it could also be that the string theory vertex operators simply are a different set of operators from those for which we computed correlation functions in field theory. Our criterion for fixing the right-moving piece of the operators in field theory was based on symmetries, i.e. by requiring that they satisfy CFT-like Ward identities with respect to the right-moving generators (86), which were related to the flowed right-moving Virasoro generators by an operator-dependent spectral flow. Since these Virasoro symmetries are not yet understood from the worldsheet perspective, the analogous way to fix the operator basis is not yet available on the string theory side. Conversely, it could be that consistency of the worldsheet vertex operators (e.g., mutual locality) would single out a different set of constraints on the operators that would be natural to impose. In any case, it would be interesting to further explore the properties of the two sets of operators and check whether one may be preferred to the other.

Finally, let us mention that exactly the same method may be used to compute correlation functions of the short string vertex operators [21–23]; one simply needs to reinterpret the relation between $\Delta$ and $h$ for the discrete $SL(2, \mathbb{R})$ representations on the worldsheet. In this case, one most commonly considers $w = 0$ short strings. Since now the representation is lowest-weight, the spacetime dimension is related to the worldsheet Casimir as $h = j + 1$. Taking into account the shift in the relation between $\Delta$ and the worldsheet dimension, one

---

[26]Note that this argument appears to imply that, if the normalization of the undeformed $AdS_3$ vertex operators is rescaled by some function of the dimensions, then all these factors would become momentum-dependent. This is *not* what happens if we perform a similar rescaling in the field-theory analysis of section 4.1, where the only functions of $h$ that acquire a momentum dependence are those that result from the Fourier transform. This suggests that the argument of [25] may be more subtle than it naïvely appears.

[27]The reason was that these discrepancies only appeared in a single exponential factor. Had they appeared in a sinh factor, they would have contributed to the shift in dimension, which would have then agreed with the long string result.

finds

$$h_{ALD} = \frac{1}{2} + \sqrt{\left(h - \frac{1}{2}\right)^2 + \frac{\mu N_5}{2\pi} p\bar{p}}\,, \quad h_{wAdS} = \frac{1}{2} + \sqrt{\left(h - \frac{1}{2}\right)^2 + \lambda N_5 q^{[0]}\bar{p} + \frac{\lambda^2 N_5 k}{4}\bar{p}^2}\,.$$

(174)

Since the information about the $T\bar{T}$ deformed symmetric product is only captured by the long string sector , it is not surprising that the momentum dependence of the conformal dimensions is different from [44]. The same comment applies to $J\bar{T}$, where the short string sector conformal dimensions do not match with our symmetric product orbifold results.

# 5 Conclusions

In this article, we have studied various properties of symmetric product orbifolds of $T\bar{T}$ and $J\bar{T}$ - deformed CFTs - namely the spectrum, the symmetries and the correlation functions - from a purely field-theoretical perspective. Our derivations relied mostly on Hilbert space techniques and made no use of conformal invariance, which is not present in these models.

The first observable we discussed was the torus partition function. We showed that the group-theoretical techniques [33–35] that were previously developed to determine the partition function of a symmetric product orbifold of *C*FTs in terms of that of the seed can be easily generalised to two-dimensional QFTs (not necessarily Lorentz-invariant) by appropriately taking into account the dependence of the partition function on the size of the circle on which the theory is defined. We also showed that the modular properties of the symmetric product orbifold followed from those of the seed theory. We then applied these results to the symmetric product orbifold of $T\bar{T}$ and $J\bar{T}$ - deformed CFTs, reproducing the finite-size spectra that were previously computed using worldsheet techniques. It would be interesting to find other classes of UV-complete QFTs with dimensionful couplings whose symmetric product orbifold can be studied with this method.

Our second result was a proof that the full Virasoro × Virasoro ($\ltimes$ Kac-Moody$^2$) symmetries of the symmetric product orbifold CFT, including their fractional counterparts, survive the single-trace $T\bar{T}$/$J\bar{T}$ deformation. As in the double-trace case [38], the argument was based on transporting the extended symmetry generators of the undeformed CFT along the irrelevant flow, and then showing that they remain conserved. The "physical" symmetry generators may now be singled out by the fact that they correspond to integrals of quasi-local current densities descended from the covering space, whereas the flowed ones explicitly depend on the twist sector. We further exploited these symmetries - following the steps of the double-trace analysis [42] - to single out a special basis of operators in single-trace $J\bar{T}$ - deformed CFTs, both from the untwisted and the twisted sector, and compute arbitrary correlation functions thereof. It seems reasonable to hope that similar techniques may be used in the future to construct the correlation functions of single-trace $T\bar{T}$ - deformed CFTs, provided the construction of correlation functions in the double-trace $T\bar{T}$-deformed CFTs can be recast in the same language as in the $J\bar{T}$ case, namely using an interplay of the symmetries and the flow equation.

Our results show that the 'QFT data' of single-trace $T\bar{T}$ and $J\bar{T}$ - deformed CFTs are rigidly determined by the corresponding observables in the seed double-trace-deformed theories, which in turn are universally determined by those of the undeformed CFT. Thus, from the point of view of the program set forth by [11] - of understanding the space of UV-complete[28] two-dimensional quantum field theories - these theories are not significantly more general than their double-trace counterparts, which themselves can be understood as mostly kinemat-

---

[28]We replaced the integrability requirement of [11] by UV-completeness, because for just the standard $T\bar{T}$/$J\bar{T}$ deformations, the universal modification of the S-matrix does not require integrability of the underlying theory.

ical deformations of the underlying CFT [87]. On the other hand, symmetric product orbifold CFTs do sometimes allow for non-universal exactly marginal deformations that break the symmetric orbifold structure; if such deformations could also be applied to a $T\bar{T}/J\bar{T}$ symmetric product orbifold in such a way that its UV completeness is preserved, then this would have the potential of significantly enlarging the space of known UV-complete, yet non-local QFTs.

Another important motivation for understanding the detailed properties of single-trace $T\bar{T}/J\bar{T}$ - deformed CFTs is their application to non-AdS holography. The exact symmetric orbifolds we studied should be holographically dual to a highly stringy spacetime, corresponding to $N_5 = 1$ in our analysis of section 4.3. The worldsheet theory for a string propagating in such a background is no longer described by the RNS formalism, but could in principle be studied with the methods of [88,89], which would be very interesting to adapt to this non-AdS setting. Alternatively, one could concentrate on the dual description of the weakly-curved spacetimes that are usually of most interest in holography, which involve deformations of these theories that break the symmetric orbifold structure. It would be very interesting to understand which features of the single-trace $T\bar{T}/J\bar{T}$ - deformed CFTs we studied - the entropy, the symmetries, the structure of the correlators - remain universal once one moves off the symmetric orbifold point. The results of [8, 40] suggest that at least the entropy and the extended symmetries should be part of the list.

# Acknowledgments

We are grateful to Luis Apolo, Brando Bellazzini, Alex Maloney, Sylvain Ribault and especially Alex Belin for insightful conversations.

**Funding information** The work of SC received funding under the "Horizon 2020" Framework Program for Research and innovation under the Marie Sklodowska-Curie grant agreement number 945298. The work of SG is supported by the PhD track fellowship of the École Polytechnique. The work of SG and MG was supported in part by the ERC starting grant 679278 Emergent-BH.

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
