# Peer review of "States, symmetries and correlators of $T\bar{T}$ and $ J\bar{T} $ symmetric orbifolds"

_SciPost Physics, doi:SciPost Phys. 16, 011 (2024)_

## Round 1 · Referee Report · Anonymous (Referee 1) · 2023-8-28

# On "States, symmetries and correlators of $T\bar{T}$ and $J\bar{T}$ symmetric orbifolds"

1: The authors studied primarily the following three points in detail in two dimensions:

- The authors studied the torus partition sum of a symmetric product obtained by taking the tensor product of a seed theory and quotienting the product by its maximal permutation symmetry in the case the seed theory is modular invariant.

  The seed theory belongs to a class of special theories. An example is a $T\bar{T}$ deformed CFT. By *requiring* the dimensionless coupling to transform with a particular modular weight the torus partition sum of a $T\bar{T}$ deformed CFT can be made modular invariant.

- The authors also studied the symmetries of a single-trace $T\bar{T}$ and $J\bar{T}$ deformed conformal field theories (CFTs). These are symmetric product theories of $T\bar{T}$ and $J\bar{T}$ deformed CFTs. In particular Virasoro (and its universal covering) and Kac-Moody symmetries.

- Other point studied by the authors is concerning correlation functions of single-trace $T\bar{T}$ and $J\bar{T}$ deformed CFTs. As in item (2), these are symmetric products of $T\bar{T}$ and $J\bar{T}$ deformed CFTs.

2: Below are my questions and/or suggestions to the authors:

- The authors claim that any quantum field theory (QFT) that is UV complete is modular invariant. I believe what they really meant is any theory that can be defined on a tours is modular invariant. Even then it is clear, by definition, why it should be modular invariant. Of course what the authors are asking is concerning the modular property of a symmetric product of modular invariant seed theory which is a different and interesting question. Therefore, it would be nice to clarify this since the wording is a bit confusing and can potentially mislead.

  In general, one is free to consider a theory that has several dimensionless couplings and require that the theory is modular invariant as in $T\bar{T}$ deformation. This requirement of course will constrain the theory since not all mathematically consistent theories, say on a plane or sphere, are modular invariant. That is, one may fail to consistently define (or extend) the theory on (or to) a torus. A good example, I believe, is 2d critical Ising model with only the identity and energy primary fields. I believe there are also other examples with defects

- The authors seem to suggest, or at least that is what it sounds to me, that the seed theory they are considering is generic *i.e.*, is a general 2d QFT. I believe, on the contrary, as I mentioned earlier, the seed theory belongs to a special class of theories that are *required* to be modular invariant. Thus, in this sense the seed theory is very special. I believe what the authors mean is in the sense that the theory can have several couplings. I would suggest the authors clarify this point.

- Check equation (2.36). The correct equation I believe has no a factor of the length of the cycle $|\xi|$ inside the product.

- In equation (2.39) the authors introduced the $n$th Hecke operator $T_n$ and its action on modular forms. The dimensionless coupling $\mu/R^2$ has modular weight (-1, -1) since we are assuming the $T\bar{T}$ deformed theory, *i.e.*, its torus partition sum, is modular invariant. Thus, the operator $T_n$ also acts on $\mu/R^2$ or at least one expects to act.

  Are the authors assuming it does not act? If it does not act or if the authors are assuming this, I would suggest the authors to write this explicitly. If it acts on the coupling, how does the operator acts on product of modular forms? Is it even defined? It could be that the Hecke operator approach might not be well suited here. It would be nice if they clarify these points.

- Check the sentence below equation (2.43). There is a typo.

- A similar result to that given in equation (2.56) is also obtained from worldsheet computation in the long string sector in the TsT picture in 2112.02596 on page 35 for all values of $w$ even before those papers cited. It is also noted there the spectrum agrees with the standard $T\bar{T}$ only for $w = 1$. I thought this might be good to cite since it could be useful for interested readers.

- The following is a comment based on the discussion on page 45, in case the authors are not aware. In the TsT picture correlation functions computation in the long string sector in a general setting was actually discussed earlier in 2112.02596. The correlation analysis was explicitly applied for $w = 1$ using the same exact reasoning in a later work by other authors. In that paper it is also noted earlier correlation computations are for $w = 0$, and thus they are not to be compared with the standard $T\bar{T}$ deformation results since their spectrums, to begin with, do not agree as it is pointed out also here which I think is an important point.

- It would be nice if the authors also comment on the states with $E^{(w)}(R) = P^{(w)}(R)$. They are not affected by the deformation, and thus do not flow.

3: Final comment:

The authors have provided detailed analyses and provided several useful results concerning the three points mentioned in (1). I recommend the paper for publication provided the authors answer the above questions properly.

---

## Round 2 · Referee Report · Anonymous (Referee 1) · 2023-11-29

Report

The authors have improved the manuscript. I recommend it for publication in its present form.

---

## Round 2 · Referee Report · Anonymous (Referee 2) · 2023-12-1

Report

I am happy with the minor revisions made and recommend the paper for publication.

---

## Round 2 · Author Response

minor corrections, references added

---

## Editorial Decision

published